# Generalizable Insights for Graph Transformers in Theory and Practice

**Timo Stoll**[*]   **Luis Müller**[*]   **Christopher Morris**
Department of Computer Science
RWTH Aachen University
Aachen, Germany
`timo.stoll@log.rwth-aachen.de`

## Abstract

Graph transformers (GTs) have shown strong empirical performance, yet current architectures vary widely in their use of attention mechanisms, positional embeddings (PEs), and expressivity. Existing expressivity results are often tied to specific design choices and lack comprehensive empirical validation on large-scale data. This leaves a gap between theory and practice, preventing generalizable insights that exceed particular application domains. Here, we propose the Generalized-Distance Transformer (GDT), a GT architecture based on standard attention that incorporates many recent advancements for GTs, and we develop a fine-grained understanding of the GDT's representation power in terms of attention and PEs. Through extensive experiments, we identify design choices that consistently perform well across various applications, tasks, and model scales, demonstrating strong performance in a few-shot transfer setting without fine-tuning. Our evaluation covers over eight million graphs with roughly 270M tokens across diverse domains, including image-based object detection, molecular property prediction, code summarization, and out-of-distribution algorithmic reasoning. We distill our theoretical and practical findings into several generalizable insights about effective GT design, training, and inference.

## 1   Introduction

Graphs are a fundamental data structure for representing relational data and are prevalent across scientific and industrial domains. They naturally model interactions in chemistry [Gilmer et al., 2017, Jumper et al., 2021], biology [Zitnik et al., 2018, Fout et al., 2017], social and citation networks [Kipf and Welling, 2017, Hamilton et al., 2017], recommendation systems [Ying et al., 2018, Wu et al., 2020], computer vision [Xu et al., 2021], and code analysis [Allamanis et al., 2018, Hellendoorn et al., 2021]. While *graph neural networks* (GNNs) [Zhou et al., 2020, Bronstein et al., 2021], specifically *message-passing neural networks* (MPNNs) [Gilmer et al., 2017], remain the most prominent architectures in graph learning, recently, *graph transformers* (GTs) have emerged [Müller et al., 2024] and have found success in applications such as protein folding [Abramson et al., 2024], weather forecasting [Price et al., 2025], or robotics [Vosylius and Johns, 2025]. Moreover, because graphs are a general modeling language, GTs can be seen as generalizations of traditional transformer architectures [Vaswani et al., 2017, Devlin et al., 2019, Brown et al., 2020]. As such, theoretical and practical insights about GTs can be leveraged to improve our understanding of transformers' reasoning abilities and representation power [Sanford et al., 2024, Cheng et al., 2025]. In addition, LLMs with causal masking can be seen as GTs on special types of directed acyclic graphs, and tools

---

[*]Equal contribution.

39th Conference on Neural Information Processing Systems (NeurIPS 2025).

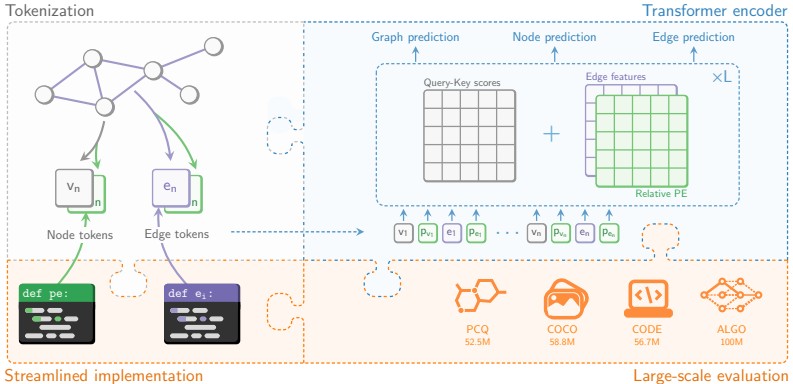

Figure 1: Overview of the GDT and accompanying evaluation. *Top left*: We support node- and edge-level tokenization with corresponding absolute PEs (depicted below each token). *Top right*: We incorporate relative PEs and edge features via the attention bias. *Bottom left*: We provide effective and streamlined implementations for incorporating edge features and PEs. *Bottom right*: All empirical evaluations are done on large-scale datasets spanning various applications.

from graph learning can be used to study and understand their behavior at inference time [Barbero et al., 2024, 2025].

Despite these promises, the progress of GTs is hindered by a lack of standardized methods for obtaining generalizable insights. Specifically, we identify three main obstacles in the current literature: *architecture-tied expressivity*, *limited evaluation*, and *graph-specific attention*. Here, architecture-tied expressivity refers to the shortcoming that current expressivity results are often tied to specific architectural designs, such as special attention mechanisms [Zhang et al., 2023, Ma et al., 2023, Müller et al., 2024, Black et al., 2024] or choices of *positional embeddings* (PEs) [Tsitsulin et al., 2022, Ma et al., 2023, Kim et al., 2022, Müller and Morris, 2024]. In addition, empirically, GTs are often evaluated and compared on small-scale datasets [Rampášek et al., 2022, Ma et al., 2023] where otherwise negligible implementation choices can be more prominent, leading to limited insights. In addition, GTs are rarely evaluated on their representation-learning capabilities, for example, in zero— or few-shot transfer settings. Finally, regarding graph-specific attention, most GTs deviate far from the traditional transformer architecture, making it challenging to derive generalizable insights about transformers beyond particular application domains.

**Present work**   This work aims to overcome the above obstacles and provide a general and powerful graph transformer architecture. Through rigorous theoretical and empirical analysis, we develop the *Generalized-Distance Transformer* (GDT), a general graph transformer architecture whose expressivity can be characterized by the powerful Generalized-Distance Weisfeiler–Leman algorithm [Zhang et al., 2023]. Concretely, the GDT

1. captures MPNNs, most graph transformers, and many other transformer models, e.g., causal and bi-directional transformers;

2. is effective across application domains, as well as on graph-, node-, and edge-level prediction tasks; and

3. is evaluated at a sufficient data scale and can learn transferable representations, allowing for few-shot transfer and extrapolation.

*Our provably expressive GDT architecture, supported by a rigorous empirical evaluation, represents a significant step toward developing highly effective, general-purpose graph models that enable generalizable insights across diverse domains.*

**Related work**   With GT architectures being successful in various domains, several approaches exist for applying transformers to graph learning tasks. Apart from pure transformer architectures such as [Dwivedi and Bresson, 2020, Ying et al., 2021a, Kim et al., 2022, Müller and Morris, 2024],

most GT designs incorporate changes to the attention mechanism [Bo et al., 2023, Kreuzer et al., 2021, Ma et al., 2023], or use attention jointly with MPNNs [Rampášek et al., 2022, Choi et al., 2024]; see Müller et al. [2024] for an overview of GTs. Moreover, Zhang et al. [2023] propose a modified attention mechanism for a GT to simulate the *Generalized Distance Weisfeiler–Leman algorithm* (GD-WL), a variant of the *Weisfeiler–Leman algorithm* (1-WL) [Weisfeiler and Leman, 1968] incorporating distance information. Indeed, there exists an extensive literature on deriving architectures more expressive than the 1-WL test, both for GNNs [Azizian and Lelarge, 2021, Maron et al., 2019a,b, Morris et al., 2020, Puny et al., 2023] as well as GTs [Ma et al., 2024, Zhang et al., 2023, 2024, Kim et al., 2022, Müller and Morris, 2024, Müller et al., 2024]. As noted by Müller et al. [2024], GTs heavily rely on structural and positional information captured by a positional embedding to increase expressiveness. Common choices include absolute PEs such as SAN [Kreuzer et al., 2021], LPE [Müller and Morris, 2024], SPE [Huang et al., 2024], and SignNet/BasisNet [Lim et al., 2023], RWSE [Dwivedi et al., 2021], RRWP [Ma et al., 2023], as well as PEs based on substructure counting [Ying et al., 2021a]. In terms of theoretical and empirical evaluation of GTs, the closest related works on the theoretical side are Zhang et al. [2024], Black et al. [2024], Rampášek et al. [2022] and Li et al. [2024]. However, neither of these works considers standard attention or compares design choices such as PEs on large-scale data.

## 2 Generalized-Distance Transformer

In this section, we derive the GDT by combining multiple methods from the recent graph learning literature, while maintaining standard attention and compatibility with most traditional transformer models. Moreover, we prove that the GDT is powerful enough to simulate the general and expressive GD-WL algorithm [Zhang et al., 2023]. We will first introduce some notation and necessary background, and then develop our theoretical framework.

### 2.1 Expressivity and Weisfeiler–Leman variants

We consider finite graphs $G \coloneqq (V(G), E(G), \ell_V, \ell_E)$ with nodes $V(G)$, edges $E(G)$. Note that for simplicity we assume that the nodes and edges are already embedded via node embeddings $\ell_V \colon V(G) \to \mathbb{R}^d$, and edge embeddings $\ell_E \colon V(G)^2 \to \mathbb{R}^d$, where $d \in \mathbb{N}^+$ is the embedding dimension and $\ell_E(v, w)$ is simply the all-zero vector if there is no edge between nodes $v$ and $w$. We always fix an arbitrary order on the nodes $V(G)$ to be consistent with vectorial representations such as those used in transformers. We study the expressivity of a graph model via its ability to distinguish non-isomorphic graphs, which is common practice for graph neural networks and GTs [Morris et al., 2019, Abboud et al., 2022, Zhang et al., 2023, Black et al., 2024, Müller and Morris, 2024]. Such a notion of expressivity is often studied in the context of the new $k$-dimensional Weisfeiler–Leman algorithm ($k$-WL) [Cai et al., 1992], a hierarchy of graph isomorphism heuristics with increasing expressivity and computational complexity as $k > 0$ grows. MPNNs without PEs typically have 1-WL expressivity. Another important graph isomorphism heuristic in the context of this work is the GD-WL variant [Zhang et al., 2023], which we formally introduce here. Concretely, given a graph $G \coloneqq (V(G), E(G), \ell_V)$, we seek to iteratively update colors for each node $v \in V(G)$, denoted $\chi_G^t(v)$, where $t \geq 0$ denotes the iteration number. We initialize $\chi_G^0(v)$ with the node colors consistent with $\ell_V$, that is $\chi_G^0(v) = \chi_G^0(w)$ if and only if $\ell_V(v) = \ell_V(w)$, for all pairs of nodes $v, w$. Then, the GD-WL updates the color $\chi_G^t(v)$ of node $v \in V(G)$, as

$$\chi_G^{t+1}(v) \coloneqq \mathsf{hash}\big(\{\!\{(d_G(v, w), \chi_G^t(w)) : w \in V(G)\}\!\}\big), \tag{1}$$

where $d_G \colon V(G)^2 \to \mathbb{R}^+$ is a distance between nodes in $G$ and $\mathsf{hash}$ is an injective function, mapping each distinct multiset to a previously unused color. The expressivity of the GD-WL depends on the choice of $d_G$. Setting $d_G(v, w) = 1$ if and only if $v$ and $w$ have an edge in $G$, yields 1-WL expressivity. In practice, the GD-WL is often implemented with a GT, where $d_G$ is incorporated via a modified attention [Ma et al., 2023, Zhang et al., 2023]. In Section 2.3, for the first time, we prove that a GT with standard attention can simulate the GD-WL, with $d_G$ being incorporated as a PE.

### 2.2 Defining the GDT

While many variations of GTs exist, we consider the standard transformer encoder based on Vaswani et al. [2017]. This allows us to use a standard attention layer, without modifications as commonly

seen in other GTs. Concretely, the GDT processes a matrix of initial token embeddings $\boldsymbol{X}^0 \in \mathbb{R}^{L \times d}$, derived from $G$, using scaled dot-product attention and subsequent application of a multi-layer perceptron (MLP). Here $L \in \mathbb{N}^+$ denotes the number of tokens, typically in the order of the number of nodes, and $d$ denotes the embedding dimension. We now describe tokenization and attention, and how we incorporate edge embeddings into the GDT.

**Tokenization** For this study, we will consider two possible tokenizations: (a) node-level tokenization, where each token corresponds to a node in $G$ and the initial token embeddings are constructed from node embeddings $\ell_V$; and (b) edge-level tokenization, where each token corresponds to either a node or an edge in $G$ and the initial token embeddings are constructed from the node embeddings $\ell_V$ and the edge embeddings $\ell_E$ for node- and edge-tokens, respectively. In practice, we use the fact that edge-level tokenization is equivalent to node-level tokenization on a transformation $G'$ of $G$, with $V(G') := \{(v, v) \mid v \in V(G)\} \cup E(G)$ and $E(G') := \{((u, v), (w, z)) \mid u = w \lor u = z \lor v = w \lor v = z\}$.

**Special tokens** As a convention, and following many prior works on transformer encoders, we use a special [cls] token to read out graph-level representations from the GT. For simplicity, we treat the [cls] token as a virtual node connected to all other nodes. This virtual node is also equipped with a unique node embedding $\ell_V([\texttt{cls}])$ and unique edge embeddings $\ell_E([\texttt{cls}], v) = \ell_E([\texttt{cls}], w)$ and $\ell_E(v, [\texttt{cls}]) = \ell_E(w, [\texttt{cls}])$, for all pairs of nodes $v, w \in V(G)$.

**Attention** For the attention, let $\boldsymbol{Q}, \boldsymbol{K}, \boldsymbol{V} \in \mathbb{R}^{L \times d}$ and $\boldsymbol{B} \in \mathbb{R}^{L \times L}$ be the *attention bias*. We define biased attention as

$$\mathsf{Attention}(\boldsymbol{Q}, \boldsymbol{K}, \boldsymbol{V}, \boldsymbol{B}) := \mathsf{softmax}\big(d^{-\frac{1}{2}} \cdot \boldsymbol{Q}\boldsymbol{K}^T + \boldsymbol{B}\big)\boldsymbol{V},$$

where $\mathsf{softmax}$ is applied row-wise. While many variations of the standard transformer layer exist, it generally takes the form

$$\boldsymbol{X}^{t+1} := \mathsf{MLP}\big(\mathsf{Attention}(\boldsymbol{X}^t\boldsymbol{W}_Q, \boldsymbol{X}^t\boldsymbol{W}_K, \boldsymbol{X}^t\boldsymbol{W}_V, \boldsymbol{B})\big), \tag{2}$$

where $\boldsymbol{W}_Q, \boldsymbol{W}_K, \boldsymbol{W}_V \in \mathbb{R}^{d \times d}$ are learnable linear transformations, and we use a two-layer MLP commonly found in transformer encoder layers; see Appendix B for a formal definition. In practice, transformers typically have additional normalizations and residual connections. They are implemented using multi-head attention with attention bias tensor $\mathbf{B} \in \mathbb{R}^{L \times L \times h}$ where $h$ is the number of attention heads; see Appendix B for a formal definition. Note that Equation (2) is a general formulation whose special cases include the local GT [Dwivedi and Bresson, 2020], an attention-based variant of MPNNs with $\mathbf{B}_{ij} = 0$ if nodes $i$ and $j$ share an edge and $\mathbf{B}_{ij} = -\infty$ else; attention with causal masking [Vaswani et al., 2017] with $\mathbf{B}_{ij} = 0$ if $i < j$ and $\mathbf{B}_{ij} = -\infty$ else; as well as many relative PEs [Shaw et al., 2018, Beltagy et al., 2020, Press et al., 2022].

**Edge embeddings** To incorporate edge embeddings into the GDT, we distinguish between node-level and edge-level GT. For the edge-level GT, edge embeddings are explicitly incorporated via edge tokens. For the node-level case, we adapt the strategy from Bechler-Speicher et al. [2025], which is itself adapted from Graphormer [Ying et al., 2021a], to incorporate edge embeddings into the attention bias via an additional projection to the number of attention heads. Formally, for all pairs of nodes $i, j \in V(G) \cup \{[\texttt{cls}]\}$,

$$\boldsymbol{B}_{ij} := \rho(\ell_E(i, j)),$$

where $\rho \colon \mathbb{R}^d \to \mathbb{R}^h$ is a neural network, such as a linear transformation or an MLP.

**Making predictions** For supervised learning with the GDT, we can make graph-, node-, and edge-level predictions by applying an MLP head to the [cls] token embedding, the node token embeddings, and the edge token embeddings. Note that we can leverage edge-level tokenization for edge-level tasks, which provides explicit edge token embeddings. We apply $k$-nearest-neighbors ($k$-NN) to the token embeddings after the last layer for few-shot transfer without additional fine-tuning.

**Absolute and relative PEs** We can incorporate two classes of PEs, *absolute* PEs such as RWSE [Dwivedi et al., 2021], LPE [Kreuzer et al., 2021, Müller and Morris, 2024], and SPE [Huang et al., 2024], which are added at the token-level, and *relative* PEs such as RRWP [Ma et al., 2023], which describe relational information between two tokens. Concretely, an absolute PE takes the form

$\mathbf{P} \in \mathbb{R}^{L \times d}$ where the row $\mathbf{P}_i$ is the embedded PE vector corresponding to token $i$. We then project and add $\mathbf{P}_i$ to the node embedding of token $i$ to obtain the initial token embeddings, or formally,

$$\boldsymbol{X}_i := \ell_V(i) + \boldsymbol{P}_i \boldsymbol{W}_P,$$

where $\boldsymbol{W}_P \in \mathbb{R}^{d \times d}$ is a learnable weight matrix. Moreover, a relative PE takes the form $\mathbf{U} \in \mathbb{R}^{L \times L \times d}$, which we project and add to the edge embeddings to construct the attention bias $\mathbf{B}$. Note that we only consider relative PEs in node-level tokenization. Concretely, for all pairs of nodes $i, j \in V(G) \cup \{[\texttt{cls}]\}$,

$$\boldsymbol{B}_{ij} := \rho(\ell_E(i,j)) + \mathbf{U}_{ij} \boldsymbol{W}_U,$$

where $\boldsymbol{W}_U \in \mathbb{R}^{d \times h}$ is a learnable weight matrix. We note that the GDT is permutation-equivariant if and only if the absolute and relative PEs used are permutation-equivariant. Since we always assume the presence of, potentially trivial, edge embeddings, the GDT has, at the very least, an embedding of the adjacency matrix as its attention bias, forming a kind of default relative PE. We will refer to this PE as NoPE.

## 2.3 The expressive power of the GDT

We now have the necessary definitions to formally state our theoretical result for the expressivity of the GDT. Importantly, our result allows us to study GT expressivity solely through PE choice, effectively decoupling model expressivity from attention selection. Concretely, we show that the GDT, with absolute and relative PEs, is sufficient to simulate the GD-WL, as well as that the GD-WL provides an upper bound on the expressivity of the GDT; see Appendix C for the proof.

**Theorem 1** (informal). *The following holds:*

1. *For every choice of distance function, there exists a selection of PEs and a parameterization of the GDT sufficient to simulate the GD-WL.*

2. *For every choice of PEs and parameterization of the GDT, there exists a distance function and an initial coloring of the GD-WL sufficient to simulate the GDT.*

The central problem we face when proving the first statement is how to injectively encode the multisets in Equation (1) with softmax-attention. This is because softmax-attention computes a weighted mean, whereas existing results for encoding multisets use sums [Xu et al., 2019, Morris et al., 2019, Zhang et al., 2023]. To overcome this limitation, we first note that the weighted mean of softmax-attention is essentially a normalized sum of exponential numbers. We then leverage a classical result from number theory, namely that sums of distinct exponential numbers are linearly independent over the algebraic numbers, known as the Lindemann–Weierstrass theorem [Baker, 1990]. In the proof of Theorem 1, we show that this linear independence property is sufficient for injectivity, provided that there are at least two distinct token embeddings, a property always satisfied in the presence of the [CLS] token. We note that our expressivity result explicitly uses a property of softmax-attention instead of leveraging idealized versions of softmax, such as saturated softmax or hardmax, commonly used for expressivity results for graph transformers [Zhang et al., 2023, Müller and Morris, 2024] and transformers in general [Pérez et al., 2019, Hahn, 2020, Merrill et al., 2022, Merrill and Sabharwal, 2024].

> **Insight 1:** The expressivity of biased attention can be characterized by the GD-WL.

A consequence of Theorem 1 is that the GDT with NoPE is equivalent to 1-WL; see Appendix C for a formal discussion of this fact. With Theorem 1, we have characterized the expressivity of the GDT in terms of the GD-WL. In the next section, we show that this expressivity can be enhanced using PEs. To this end, we present a range of new PE expressivity results, yielding the most fine-grained picture of GT expressivity.

## 3 The expressive power of positional embeddings

This section provides a comprehensive theoretical expressiveness hierarchy of PEs based on the works of Black et al. [2024] and Zhang et al. [2024], including novel results on PE expressiveness. Based on the theoretical results from Section 2.3, we expand on GDT expressiveness by introducing

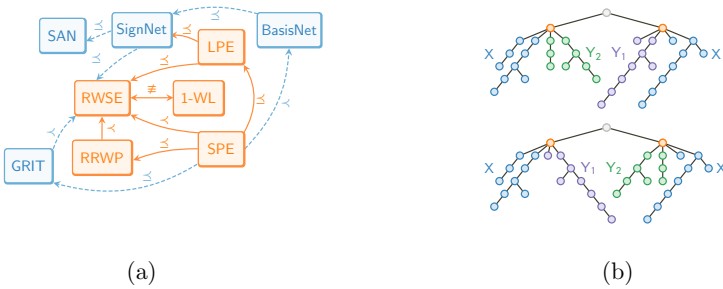

(a)                      (b)

Figure 2: (a): Overview of our theoretical PE results in the context of existing results for PE expressivity. $A \prec B$ ($A \preceq B$, $A \not\equiv B$): algorithm A is strictly more powerful (at least as powerful, incomparable) than/to B (b): Trees proposed by Cvetković [1988] used in the proof of Theorem 2.

PE expressiveness, establishing a pathway between our transformer architecture and incorporating graph structure information. Furthermore, leveraging the PE expressiveness results, we obtain initial guidelines for empirically evaluating PE design choices. Here, we introduce the four PEs central to our theoretical and empirical study; see Appendix D for results for additional PEs.

**PEs** We consider random-walk-based PEs and those based on the graph Laplacian's eigenvalues. Random-walk-based PEs are embeddings of the random-walk probabilities obtained from multiple powers of the degree-normalized adjacency matrix of the graph. We consider RWSE [Dwivedi et al., 2021], an absolute PE which uses only the return probabilities of random walks for each node and has linear-time complexity, and RRWP [Ma et al., 2023], a relative PE, which uses all random walk probabilities between two nodes and has quadratic runtime complexity.

Laplacian PEs are embeddings of the eigenvectors and eigenvalues of the graph Laplacian; see Appendix B for a definition. Here, we consider LPE [Kreuzer et al., 2021, Müller and Morris, 2024], an absolute PE which uses a linear-time embedding method but suffers from a lack of basis-invariance, making the PE non-equivariant to the permutation of nodes [Lim et al., 2023]. In addition, we consider SPE [Huang et al., 2024], an absolute PE which is permutation-equivariant but has quadratic runtime complexity. We restrict ourselves to this selection as other common PEs, such as the shortest path distance or resistance distance, can be approximated by RWSE or RRWP [Black et al., 2024].Further details and formal definitions are presented in Appendix B.

**1-WL and random-walk PEs** Tönshoff et al. [2023] already show that there are pairs of graphs with $n$ nodes, distinguishable by the 1-WL, requiring random walks with at least $\mathcal{O}(n)$ steps to be distinguished. Here, we show that RWSE is incomparable to the 1-WL. This holds independent of the number of random walk steps, and as a result, we can consider RWSE to provide additional information as a PE to a pure transformer architecture by differentiating 1-WL indistinguishable graphs; see Appendix D for the proof.

**Theorem 2.** *The RWSE embedding is incomparable to the* 1*-WL test.*

We briefly highlight the most essential proof idea. Concretely, the selected trees introduced by Cvetković [1988], shown in Figure 2, are known not to be distinguishable using their eigenvalues and graph angles. However, all trees can be distinguished by the 1-WL test [Cai et al., 1992]. At the same time, it is well-known that RWSE can distinguish indistinguishable CSL graphs by the 1-WL [Dwivedi et al., 2021]. Further, we note that Theorem 2 provides not only a single pair of graphs but rather an infinite number of trees indistinguishable by RWSE. Finally, we show the following result relating RRWP to RWSE.

**Proposition 3.** *RRWP is strictly more expressive than RWSE, given the same random walk length.*

**Random-walk PEs and eigen PEs** We provide results for RWSE, RRWP, LPE, and SPE. In contrast to previous works [Ma et al., 2023, Zhang et al., 2024], we analyze the expressivity of RRWP directly, without using the GRIT architecture. We find that RRWP is approximated by SPE, which in turn is strictly weaker than the 3-WL test [Zhang et al., 2024]. Further, we expand on proofs by Lim

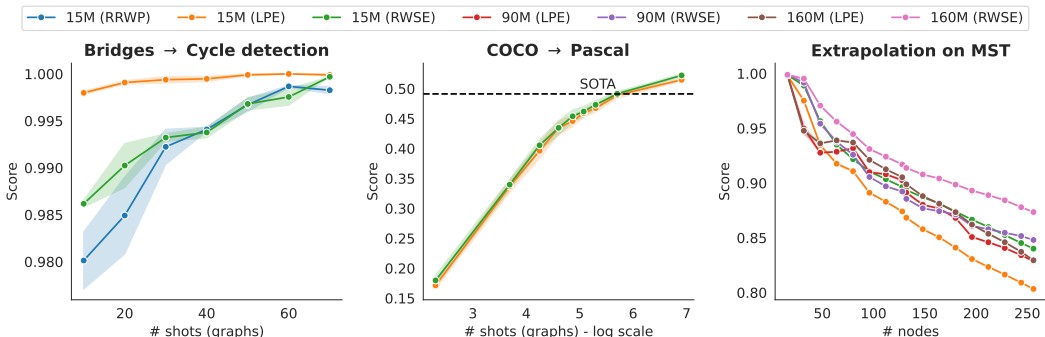

Figure 3: Results of inference-time experiments. From left to right: Few-shot transfer from BRIDGES to CYCLES with 3-NN over 3 random seeds; few-shot transfer from COCO to PASCAL with 5-NN over 10 random seeds; extrapolation beyond training data on MST over 3 random seeds.

et al. [2023] to approximate RWSE using LPE. Taken together, we obtain a fine-grained hierarchy of PEs, which we summarize in the following result.

**Proposition 4.** *SPE is at least as expressive as LPE and RRWP, LPE is at least as expressive as RWSE.*

We obtain an even more fine-grained hierarchy of Eigen-vector-based PEs by including additional embeddings as discussed in Appendix D. As a consequence of the hierarchy, all PEs considered in this section can distinguish graphs not distinguishable by the 1-WL or, equivalently, the GDT with NoPE.

> **Insight 2:** Any PE in {RWSE, RRWP, LPE, SPE} enhances the expressivity of the GDT.

We present a summary of our results in Figure 2 and highlight our contributions to a completed theoretical expressiveness hierarchy of PEs, complementing the work of Zhang et al. [2024] and Black et al. [2024]. Together with the theoretical insights obtained in Section 2 we arrive at a detailed understanding of the expressivity of the GDT with four popular PEs.

## 4   Experiments

In this section, we empirically evaluate our GDT model on various real-world and synthetic datasets, at graph-, node-, edge-level, and in- and out-of-distribution. Concretely, we compare the performance and efficiency of the PEs in Section 3 and study few-shot transfer, parameter scaling, and size generalization capabilities of the GDT. In the process, we hope to gain a deeper understanding of the relationship between empirical performance and expressivity, based on the results presented in Section 3, and derive generalizable insights for GTs. Next, we describe our implementation design, dataset selection, and experimental schedule, and present our empirical results.

**Implementation**   We base our implementation on the `torch.nn.TransformerEncoderLayer` proposed in PyTorch [Paszke et al., 2019]. This allows us to use memory and runtime-efficient attention implementations such as FlashAttention [Dao et al., 2022] and Memory Efficient Attention [Rabe and Staats, 2021]. In addition, we seek to harmonize implementation differences across PEs to reduce the impact of implementation-specific advantages as much as possible by using the same number and width of MLP layers and the same activation functions across all PEs. A complete overview of our implemented model architecture is given in Appendix A.

**Real-world datasets**   For consistency across graph-, node-, and edge-level tasks, we measure dataset size by the number of input tokens. For the real-world tasks, we evaluate our models on PCQM4Mv2 (PCQ) [Hu et al., 2021], a molecular property prediction dataset with 52.5M tokens, COCO [Dwivedi et al., 2022], an image-based object detection dataset with 58.8M tokens, and OGB-Code2 (CODE) [Hu et al., 2020], a code summarization dataset with 56.7M tokens. We focus

exclusively on large-scale datasets from diverse domains to derive generalizable insights for GT performance. For few-shot transfer, we select PASCAL [Dwivedi et al., 2022], which has the same image domain as COCO but uses different object categories. Bechler-Speicher et al. [2025] already demonstrate strong transfer from COCO to PASCAL through fine-tuning. We give an overview of state-of-the-art performance on these datasets in Appendix A.

**Algorithmic reasoning datasets** In addition to real-world tasks, we add synthetic algorithmic reasoning tasks for graph algorithms inspired by the CLRS benchmark [Velickovic et al., 2022]. Our selection includes the minimum spanning tree problem (MST), detecting bridges in a graph (BRIDGES), and calculating the maximum flow in an undirected graph (FLOW). Here, BRIDGES and MST are edge-level tasks, and FLOW is a graph-level task. We further consider the task of detecting whether a node lies on a cycle (CYCLES), a node-level complement to BRIDGES, to evaluate transfer learning capabilities. Following the literature in algorithmic reasoning for transformer architectures [Zhou et al., 2022, 2024a,b] and in particular, graph algorithmic reasoning [Diao and Loynd, 2023, Velickovic et al., 2022, Markeeva et al., 2024, Müller et al., 2024], we evaluate in the size generalization setting where test-time graph instances are up to 16 times larger than those seen during training. Size generalization has been recently identified as a key challenge for graph learning [Morris et al., 2024]. An expanded introduction to each task is available in Appendix A.

## 4.1 Experimental design

In the first step, we evaluate different PE choices from Section 3 for the GDT on all six upstream tasks. We also consider NoPE, which uses only edge embeddings to infer the graph structure. We fix the parameters to 15M; see Appendix A for the choice of hyperparameters. Additionally, we compute the runtime and memory efficiency observed for each PE and task. Due to the high memory requirements of storing full random-walk matrices for RRWP on large datasets, we compute RRWP matrices at runtime. To allow a fair comparison between PEs that accounts for computational efficiency, we set a compute budget of 5 GPU days for the 16M models. Furthermore, we evaluate the GDT in comparision to Graphormer-GD [Zhang et al., 2023] on BREC [Wang and Zhang, 2024], a benchmark evaluating theoretical expressiveness on graph samples. For this, we use RWSE and LPE as PEs for both architectures.

In the second step, we select the best models from the first step and further evaluate them using few-shot transfer, scaling model size, and extrapolating the graph size. In particular, we assess few-shot transfer from COCO to PASCAL, as well as few-shot transfer from BRIDGES to CYCLES. Note that even though BRIDGES is an edge-level task and CYCLES is a node-level task, we should expect strong transfer, as a node lies on a cycle if and only if at least one of its incident edges is not a bridge. For scaling, we train additional models with 90M and 160M parameters for PCQ and MST. Finally, we provide extrapolation results for up to 256 nodes (16× the size of the training graphs) on MST.

## 4.2 Discussion of base models

We present our task results in Table 1, as well as runtime and memory requirements in Figure 4. RRWP performs best of all selected PEs on 4 out of 6 tasks. However, due to the need to compute RRWP matrices at runtime, for COCO and CODE we use additional resources to provide results. Most notably, RWSE and LPE perform significantly better than NoPE and SPE for all tasks except FLOW, but do not face any efficiency issues. Furthermore, LPE and RWSE perform similarly across tasks, placing second and third, respectively, and are often competitive with the less efficient RRWP. Despite theoretical results observed in Section 3, we note the differences in experimental results to be less pronounced. This aligns with previous work on PEs, indicating discrepancies between theory and empirical evaluation. However, we extend this observation by providing an evaluation at scale and a direct comparison of PEs on a single architecture.

> **Insight 3:** PE efficiency can vary greatly while predictive performance differences are less pronounced.

Due to their favorable efficiency and competitive predictive performance, we selected LPE and RWSE for our scaling and extrapolation experiments and few-shot transfer. For few-shot transfer from

Table 1: 16M parameter results for different PEs over 3 random seeds. PCQ MAE is in micro electron volt (meV) for clarity of presentation. The mean rank is computed by sorting the models' scores for each task.

| PE | Mean Rank | PCQ MAE ↓ | COCO F1 ↑ | CODE F1 ↑ | FLOW MAE ↓ | MST F1 ↑ | BRIDGES F1 ↑ |
|---|---|---|---|---|---|---|---|
| NoPE | 3.50 | $93.6_{\pm 0.5}$ | $43.12_{\pm 00.85}$ | $19.27_{\pm 00.20}$ | $1.73_{\pm 0.09}$ | $93.29_{\pm 00.88}$ | $55.36_{\pm 24.94}$ |
| LPE | 2.50 | $92.7_{\pm 0.9}$ | $44.83_{\pm 00.71}$ | $19.48_{\pm 00.21}$ | $1.75_{\pm 0.12}$ | $91.08_{\pm 00.95}$ | $91.76_{\pm 07.66}$ |
| SPE | 4.00 | $94.1_{\pm 0.6}$ | $43.87_{\pm 00.54}$ | $19.35_{\pm 00.21}$ | $1.98_{\pm 0.14}$ | $92.52_{\pm 00.12}$ | $54.81_{\pm 21.20}$ |
| RWSE | 2.67 | $92.9_{\pm 0.6}$ | $43.82_{\pm 01.01}$ | $19.39_{\pm 00.47}$ | $1.49_{\pm 0.02}$ | $93.26_{\pm 00.45}$ | $87.34_{\pm 03.97}$ |
| RRWP | **2.33** | $90.4_{\pm 0.3}$ | $39.91_{\pm 01.07}$ | $19.42_{\pm 00.10}$ | $1.45_{\pm 0.06}$ | $96.04_{\pm 00.91}$ | $99.21_{\pm 00.09}$ |

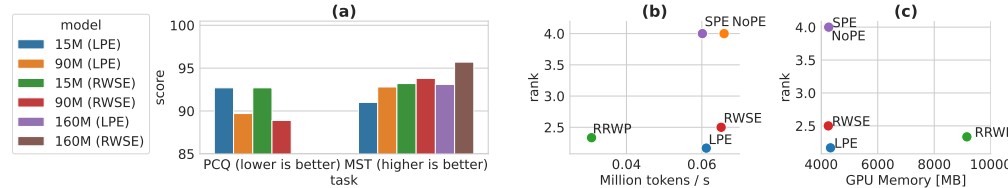

Figure 4: (a): Results on 90M and 160M models for PCQ and MST evaluated on LPE and RWSE. (b): Number of tokens evaluated per second during training for each PE. Results are obtained by averaging runtimes per token across tasks. (c): Average GPU memory requirement for each PE.

BRIDGES to CYCLES, we additionally select RRWP due to its significantly better OOD performance on BRIDGES.

With results for BREC shown in Table 2, we observe the performance of GDT and Graphormer-GD models to be comparable and differences to be negligible, aligning experimental results to theoretical observations made in Theorem 1 and showcasing non-trivial performance. Further, the difference in expressiveness of RWSE and LPE can be observed for both models. However, we note that BREC results depend on specific hyperparameter choices and are often unstable during training. An overview of the hyperparameters selected for BREC can be found in Appendix A.7.

### 4.3 Extended evaluation

We present the scaling results in Figure 4 (a), few-shot transfer in Figure 3 (a) and (b), and extrapolation results in Figure 3 (c). For scaling, we observe that the relative performance between PEs is relatively robust to model scale. In three out of four cases, in- and out-of-distribution performance improves consistently with increasing model scale. The only exception is the 160M model with LPE on MST, which drops off slightly compared to its 90M counterpart but still outperforms both 15M models on this task.

> **Insight 4:** Scaling the GDT generally improves in- and out-of-distribution performance.

For few-shot transfer, we find that all three evaluated models can demonstrate strong performance when transferring from BRIDGES to CYCLES with just a few shots. In particular, the 15M model with LPE already achieves near-perfect performance with 10 shots and is significantly better than RWSE and RRWP up to 60 shots. When transferring from COCO to PASCAL, we observe performance increases even for 1000 shots where both RWSE and LPE surpass the current SOTA on PASCAL, despite seeing less than 10% of the available training samples at inference-time; see Appendix A for an overview of state-of-the-art performance on PASCAL.

> **Insight 5:** Representations learned by the GDT allow for effective few-shot transfer.

Finally, we find all PEs and model scales to extrapolate well on MST. In particular, we still observe an F1 score of around 85 at 256 nodes or $16\times$ the graph sizes seen during training.

Table 2: Results of GDT and Graphormer-GD models on the BREC benchmark. Each column indicates the number of samples correctly distinguished by each model.

| Model | PE | Basic | Regular | Extension | CFI | Total |
|---|---|---|---|---|---|---|
| Graphormer-GD | RWSE | 57 | 50 | 96 | 0 | 203 |
| Graphormer-GD | LPE | 55 | 40 | 85 | 3 | 183 |
| GDT | RWSE | 57 | 50 | 96 | 0 | 203 |
| GDT | LPE | 54 | 39 | 84 | 3 | 180 |

## 5 Limitations

Currently, the GDT can only make use of memory-efficient attention at inference time, due to the use of a learnable attention bias. For example, the learnable attention bias is not compatible out-of-the-box with FlashAttention2 [Dao, 2024] or FlexAttention [Dong et al., 2025]. Moreover, many more PE variants exist that could be included in our study; see, for example, PEs listed in Section 1. Finally, we note that our current implementation does not take the sparsity of the attention bias into account, which can lead to prohibitive memory requirements for very large graphs.

## 6 Conclusion

We establish the GDT, a generalizable, expressive graph transformer based on the standard transformer implementation. We show the GDT to be equivalent to the GD-WL in terms of theoretical expressiveness, augmented in expressivity by using PEs and their respective expressiveness. Further, we demonstrate strong empirical performance across multiple domains and large-scale datasets, determining an empirical hierarchy of PEs. We also show the GDT to be able to learn transferable representations, extrapolate on graph size for synthetic tasks, and results being robust concerning model scale. Thereby, we provide generalizable theoretical and empirical insights for graph transformers.

## Acknowledgments and Disclosure of Funding

TS, LM, and CM are partially funded by a DFG Emmy Noether grant (468502433) and RWTH Junior Principal Investigator Fellowship under Germany's Excellence Strategy. We thank Erik Müller for crafting the figures.

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

Table 3: Hyperparameters for our 16M models.

| Hyperparameter | PCQ | COCO | CODE | FLOW | MST | BRIDGES |
|---|---|---|---|---|---|---|
| Learning rate | 1e-4 | 4e-4/3.5e-4 | 1e-4/1.5e-4 | 1e-4/2e-4 | 3e-4 | 2e-4/1e-4 |
| Batch size | 256 | 32 | 32 | 256 | 256 | 256 |
| Optimizer | AdamW | AdamW | AdamW | AdamW | AdamW | AdamW |
| Grad. clip norm | 1 | 1 | 1 | 1 | 1 | 1 |
| Num. layers | 16 | 16 | 16 | 16 | 16 | 16 |
| Hidden dim. | 384 | 384 | 384 | 384 | 384 | 384 |
| Num. heads | 16 | 16 | 16 | 16 | 16 | 16 |
| Activation | ReLU | ReLU | ReLU | ReLU | ReLU | ReLU |
| RWSE/RRWP Steps | 32 | 32 | 32 | 16 | 16 | 16 |
| Num eigvals/eigvecs | 32 | 32 | 32 | 16 | 16 | 16 |
| Hidden dim. RWSE/RRWP | 768 | 768 | 768 | 768 | 768 | 768 |
| Hidden dim. LPE/SPE | 384 | 384 | 384 | 384 | 384 | 384 |
| Num. layers $\phi$ | 2 | 2 | 2 | 2 | 2 | 2 |
| Num. layers $\rho$ | 2 | 2 | 2 | 2 | 2 | 2 |
| GNN type $\rho$ (SPE) | GIN | GIN | GIN | GIN | GIN | GIN |
| Edge encoder | MLP | MLP | MLP | MLP | MLP | MLP |
| Weight decay | 0.1 | 0.1 | 0.1 | 0.1 | 0.1 | 0.1 |
| Dropout | 0.1 | 0.1 | 0.1 | 0.1 | 0.1 | 0.1 |
| Attention dropout | 0.1 | 0.1 | 0.1 | 0.1 | 0.1 | 0.1 |
| #Steps | 2M | 1M | 200k | 11718 | 11718 | 11718 |
| #Warmup steps | 20k | 10k | 2k | 118 | 118 | 118 |

# A Experimental details

Here, we present hyperparameter choices, architecture design, and dataset selections for the empirical evaluation of our GT architecture.

## A.1 Data sources and licenses

PCQM4MV2 is available at `https://ogb.stanford.edu/docs/lsc/pcqm4mv2/` under a CC BY 4.0 license. OGB-Code2 is available at `https://ogb.stanford.edu/docs/graphprop/#ogbg-code2` under a MIT license. The COCO-SP and PASCAL-VOC-SP datasets as part of the LRGB benchmark [Dwivedi et al., 2022] are available at `https://github.com/vijaydwivedi75/lrgb` under a CC BY 4.0 license. BREC is available at `https://github.com/GraphPKU/BREC` under a MIT license. Statistics for all datasets, including the algorithmic reasoning datasets, are available in Table 5

## A.2 Hyperparameters

Table 3 and Table 4 give an overview of the hyperparameters used for models highlighted in our work. Given the large number of hyperparameters and the scale of the tasks, we did not perform a grid search or any other large-scale hyperparameter optimization. Nonetheless, we swept the learning rate for each task and model size. Across the experiments, we select the hyperparameters based on the best validation score and then evaluate on the test set. We search for suitable learning rates on the 16M models to determine which models to scale. Due to the increased computational demand, we then reduce the learning rate for the 100M models.

For PCQ, we set the learning rate to 1e-4 after sweeping the learning rate over the set {7e-5, 1e-4, 3e-4}. Further, we set the learning rate for COCO to 4e-4 for eigen-information-based embeddings and 3.5e-4 for RWSE after sweeping over {7e-5, 1e-4, 3e-4, 4e-4}. For CODE, we reduce the learning rate to 1.5e-4 (RWSE) with the same sweep as with PCQ, and across all MST runs, we keep the learning rate at 3e-4 for the 16M models, reducing the learning rate to 1e-4 (LPE) and 7e-5 (RWSE) for the 90M model and 7e-5 for the 160M model at MST with the same initial sweep as for COCO. Furthermore, for FLOW, we set the learning rate to 1e-4 (LPE, SPE, NoPE) and 2e-4 (RWSE, RRWP), respectively. We obtain similar results for BRIDGES with 1e-4 (RWSE, LPE) and 2e-4

Table 4: Hyperparameters for our 90M and 160M models.

| Hyperparameter | PCQ(90M) | MST(90M) | MST(160M) |
|---|---|---|---|
| Learning rate | 1e-4 | 1e-4/7e-5 | 7e-5 |
| Batch size | 256 | 256 | 256 |
| Optimizer | AdamW | AdamW | AdamW |
| Grad. clip norm | 1 | 1 | 1 |
| Num. layers | 24 | 24 | 24 |
| Hidden dim. | 768 | 768 | 1024 |
| Num. heads | 16 | 16 | 16 |
| Activation | ReLU | ReLU | ReLU |
| RWSE/RRWP steps | 32 | 32 | 32 |
| Num. eigvals/eigvecs | 32 | 32 | 32 |
| Hidden dim. RWSE/RRWP | 768 | 768 | 768 |
| Hidden dim. LPE/SPE | 384 | 384 | 384 |
| Num. layers $\phi$ | 2 | 2 | 2 |
| Num. layers $\rho$ | 2 | 2 | 2 |
| GNN type $\rho$ (SPE) | GIN | GIN | GIN |
| Edge encoder | MLP | MLP | MLP |
| Weight decay | 0.1 | 0.1 | 0.1 |
| Dropout | 0.1 | 0.1 | 0.1 |
| Attention dropout | 0.1 | 0.1 | 0.1 |
| #Steps | 2M | 11718 | 11718 |
| #Warmup steps | 20k | 118 | 118 |

Table 5: Dataset statistics.

| Statistic | PCQ | COCO | CODE | FLOW | MST | BRIDGES |
|---|---|---|---|---|---|---|
| # Graphs | 3.746M | 123,286 | 452,741 | 1M | 1M | 1M |
| # Avg. Nodes | 14.13 | 476.88 | 125.2 | 16/64 | 16/64 | 16/64 |
| # Avg. Edges | 14.56 | 3,815.08 | 124.2 | 48.11/213.586 | 31.66/209.34 | 48.46/ 395.02 |
| Prediction level | graph | node | graph | graph | edge | edge |
| Metric | MAE | F1 | F1 | MAE | F1 | F1 |

(RRWP, SPE, NoPE) as the learning rate, using the initial sweep from PCQ. In addition, we evaluated each PE with {8,32} random walk steps or eigenvectors and {4,16} for algorithmic reasoning tasks. Regarding PE encoder design, we selected a simple MLP architecture with two layers where applicable. However, we use the same number of layers, heads, and embedding dimensions across datasets in our transformer architecture, thereby keeping the architecture unchanged. Otherwise, we follow previous literature for initial hyperparameter choices, namely the GraphGPS [Rampášek et al., 2022], GRIT [Ma et al., 2023], and Graphormer [Ying et al., 2021a] papers. We used an AdamW optimizer for each experiment with $\beta_1 = 0.9$ and $\beta_2 = 0.999$. Further, the learning rate scheduler uses a cosine annealing schedule with a 1% warm-up over the total number of steps. Additionally, we use an L1 loss for regression targets and a cross-entropy loss for classification targets, except CODE, where we use the proposed loss function. In Table 6 and Table 7, we further report runtimes and memory usage of all models evaluated in our work.

## A.3 Architecture

In the following, we showcase the implementation of our GT architecture and the injection of PE information into the attention mechanism. We consider an Encoder, Processor, Decoder architecture with additional preprocessing for the PEs and graph-specific features.

**Preprocessing**  First, we preprocess each dataset to include the respective eigenvalues and eigenvectors of the graph Laplacian and the powers of the random walk matrices. These are then applied to the respective PE encoder, which is an MLP with two layers that casts the PE features to the embedding dimension. We consider transformed graphs for edge-level tasks, such as BRIDGES and MST, as a special case. In this case, the graph is converted to constitute the edge-level graph corresponding to the original graph. Then, node features and PE features are computed on this transformed graph. For the GDT, we provide a maximum context size per dataset. Tokens exceeding the context size are then removed.

**Encoder**  The node and edge features of each dataset's graphs are then fed into a linear layer, mapping them to the embedding dimension. These feature embeddings are specific to each dataset and embed graph-specific features. For CODE we consider additional preprocessing steps, as described by [Hu et al., 2021] to derive the respective graph structure. Further, we add a [cls] token as it is a standard practice to read out graph-level representations [Ying et al., 2021b].

**Processor**  Following our description of the GDT architecture as shown in Section 2, a single layer in the GDT architecture computes the expression shown in Definition 8 using GELU as a nonlinearity. In the case of absolute PEs, they are added to the node embeddings before the initial layer; for relative PEs, they are added to the attention bias $B$. The GT layer then computes full multi-head scaled-dot-product attention over node-level tokens, adding $B$ to the unnormalized attention matrix before applying softmax. We refer to Appendix B.1 for a detailed discussion. From this representation, including node and edge features, relative and absolute PEs, and the embedded graph structure, the processor computes representations of node- and graph-level features. We then stack multiple GT layers together: 12 for the 16M model and 24 for the 100M model.

**Decoder**  After the last layer, an MLP decoder with two layers is applied to provide the prediction head of the model. Since each dataset has its prediction target, we provide a decoder for each dataset mapping the last layer output to the prediction target, where $\boldsymbol{W}_1 \in \mathbb{R}^{d \times d}$ and $\boldsymbol{W}_2 \in \mathbb{R}^{d \times o}$ are learnable weight matrices and $o$ is the respective output dimension for each task, i.e.,

$$\boldsymbol{W}_2 \text{LayerNorm}(\text{GELU}(\boldsymbol{W}_1 x)).$$

For clarity, we omit bias terms throughout this section. Each result is then passed to the respective loss function to compute the gradient step.

### A.4  Algorithmic reasoning data

For our synthetic experiments, we evaluate on three out-of-distribution algorithmic reasoning tasks derived from the CLRS benchmark [Velickovic et al., 2022], totaling 100M tokens. These tasks assess size generalization in a controlled synthetic setting with randomly generated graphs. Unlike CLRS, we do not train models with intermediate algorithmic steps. Here, we describe graph generation and each algorithmic reasoning task in detail.

**Graph generation**  We develop a heuristic graph-generation method that produces graphs with desirable problem-specific properties, such as a reasonable distribution of shortest-path lengths or the number of bridges. Concretely, we begin by sampling an Erdos-Renyi graph $G$ with $n$ nodes and edge probability $p$ and denote the connected components of $G$ with $C_1, \ldots, C_m$. For each $i \in [m]$, we randomly choose a component $C_j$ with $j \neq i$. Then, we select random nodes $v \in C_i$ and $w \in C_j$ and augment $G$ with the edge $(i, j)$. We repeat this process $K$ times. We select parameters $p$ and $K$ for each task based on problem-specific characteristics. We detail these choices in the task descriptions, which we provide next.

**Maximum flow**  In FLOW, the task is to predict the maximum flow value in an edge-weighted directed graph. The task uses discrete node features indicating whether a node is the source, the sink, or neither. The task uses the flow capacity between two nodes as continuous scalar-valued edge features. FLOW is a graph-level regression task.

**Minimum spanning tree**  In MST, the task is to predict the set of edges that forms the minimum spanning tree (MST) in an edge-weighted graph with mutually distinct edge weights, ensuring the

Table 6: 16M/90M/160M models runtime results of a single step, averaged across 1 000 steps. Each value is given in seconds/step.

| PE | #Param. | PCQ | COCO | CODE | FLOW | MST | BRIDGES |
|----|---------|-----|------|------|------|-----|---------|
| NoPE | 16M | 0.079 | 0.105 | 0.0921 | 0.071 | 0.096 | 0.072 |
| LPE | 16M | 0.084 | 0.109 | 0.104 | 0.071 | 0.106 | 0.088 |
| SPE | 16M | 0.091 | 0.123 | 0.105 | 0.059 | 0.106 | 0.085 |
| RWSE | 16M | 0.072 | 0.101 | 0.102 | 0.071 | 0.091 | 0.088 |
| RRWP | 16M | 0.137 | 0.135 | 0.219 | 0.066 | 0.1452 | 0.101 |
| LPE | 90M | 0.167 | - | - | - | 0.184 | - |
| RWSE | 90M | 0.159 | - | - | - | 0.186 | - |
| LPE | 160M | 0.219 | - | - | - | 0.246 | - |
| RWSE | 160M | 0.219 | - | - | - | 0.237 | - |

uniqueness of the MST. The task uses the weight of each edge as continuous scalar-valued edge features. MST is a binary edge classification task where the class label indicates whether an edge is contained within the MST.

**Bridges**   In BRIDGES, the task is to predict the set of edges that are bridges in an undirected graph. The task does not use any node or edge features. BRIDGES is a binary edge classification task where the class label indicates whether an edge is a bridge in the graph.

### A.5   Runtime and memory

Here we provide additional information on the runtime and memory requirements of our GDT. We sample the runtime of each experiment by running multiple steps and averaging their runtime. For memory consumption, we consider the model's complete forward pass and estimate the allocated memory using PyTorch's memory profiling. All computations were made using bfloat16 precision during computation. We run the experiments on a single node consisting of one L40 GPU with 40GB VRAM, 12 CPU cores, and 120GB RAM for all runtime and memory computations. In case of COCO and CODE with RRWP as a PE, we used 2 l40 GPUs. We further note that the presented runtimes are the final runtimes obtained from the selected experiments, and significantly more runtime was used to obtain the chosen hyperparameter choices. We note that the automatic compilation is performed automatically by torch.compile, improves the runtime and memory scaling significantly across all tasks.

Table 6 shows the runtime for a single step, averaged across 1 000 training steps obtained for each model. Timings were obtained using torch functionality. Further Table 7 shows the memory requirement for 1 000 steps of each model. We further note the runtime speed improvements during inference experiments from Section 4 while using FlashAttention [Dao et al., 2022].

**Hardware optimizations**   Efficient neural network compilation is already available via CUDA implementations in PyTorch and other programming languages such as Triton. We use torch.compile throughout all our experiments. In addition, we want to highlight FlashAttention [Dao et al., 2022], available for the standard transformer, and used in the GDT as an example of architecture-specific hardware optimizations that can reduce runtime and memory requirements.

### A.6   Comparison with state-of-the-art

While our study focuses exclusively on the GDT, we provide SOTA performance numbers for our real-world tasks to understand whether the GDT performance is competitive with the best models in the literature. Concretely, for PCQ without 3D positions, the best models typically achieve between 0.0809 and 0.0859 MAE [Chen et al., 2023, Müller et al., 2024, Ma et al., 2023, Rampášek et al., 2022]. For COCO and PASCAL, we find models are generally evaluated on a 500K parameter budget and achieve up to 43.98 F1 and 49.12 F1, respectively [Chen et al., 2025]. Note that we do not adhere to this budget when training on COCO as we find it overly restrictive given the considerable size

Table 7: 16M/90M/160M models memory requirements in MB for 1 000 steps of each model during training.

| PE | #Param. | PCQ | COCO | CODE | FLOW | MST | BRIDGES |
|---|---|---|---|---|---|---|---|
| NoPE | 16M | 3120.63 | 5117.57 | 9749.29 | 1702.69 | 3255.37 | 2456.55 |
| LPE | 16M | 3221.13 | 5239.89 | 9852.71 | 1763.87 | 3401.60 | 2499.54 |
| SPE | 16M | 3161.32 | 5157.65 | 9763.58 | 1730.19 | 3290.09 | 2490.0 |
| RWSE | 16M | 3131.57 | 5147.34 | 9766.30 | 1713.27 | 3276.84 | 2474.63 |
| RRWP | 16M | 5419.56 | 5223.77 | 19221.97 | 2253.85 | 5522.04 | 3723.31 |
| LPE | 90M | 9844.48 | - | - | - | 10197.77 | - |
| RWSE | 90M | 9659.54 | - | - | - | 9947.07 | - |
| LPE | 160M | 13513.76 | - | - | - | 13986.39 | - |
| RWSE | 160M | 13266.44 | - | - | - | 13647.66 | - |

of this dataset. Consequently, we also use the pre-trained 15M model when performing few-shot transfer from COCO to PASCAL. Finally, on CODE, the best models score somewhere between 19.37 [Chen et al., 2022] and 22.22 F1 [Geisler et al., 2023].

## A.7 BREC Evaluation

We selected RWSE and LPE as two representative PEs for the random-walk and eigen PE, respectively. We further evaluated RRWP PEs but observed training instability in both models, which we attribute to the small dataset size in BREC (only 64 samples per task). The small-scale dataset size seems to affect the relative PE RRWP more than the node-level PE RWSE and LPE. Our implementation aligns with GDT and Graphormer-GD, differing only in the attention style. Moreover, we choose 4 layers, an embedding dimension of 64, 4 attention heads, and 8 random walks and eigenvalues, respectively. We generally observed training instabilities caused by the multiplicative $\phi_1$ of Graphormer-GD [Zhang et al., 2023]. To achieve the performance seen in Table 2, we manually initialize the weights of $\phi_1$ and $\phi_2$ and such that Graphormer-GD initially performs standard attention (akin to the GDT).

## A.8 Scaling Results

Here, we provide additional results for scaling the GDT to 90M and 160M parameters. These results correspond to the results seen in Figure 4.

Table 8: 90M and 160M parameter results for different PEs over 2 random seeds. PCQ MAE is in micro electron volt (meV) for clarity of presentation.

| PE | PCQ (90M) MAE $\downarrow$ | MST (90M) F1 $\uparrow$ | MST (160M) F1 $\uparrow$ |
|---|---|---|---|
| LPE | 89.7 $_{\pm 0.4}$ | 92.86 $_{\pm 00.17}$ | 93.11 $_{\pm 01.01}$ |
| RWSE | 88.9 $_{\pm 0.7}$ | 94.29 $_{\pm 00.68}$ | 95.80 $_{\pm 00.18}$ |
| RRWP | 86.5 $_{\pm 0.3}$ | - | - |

## A.9 Note on Architecture Selection

By using standard quadratic attention, as seen in transformers for other domains, we can leverage existing theoretical insights and empirical improvements observed in the literature. In addition, these assumptions make our proposed architecture, in theory, compatible with existing frameworks for improving training and inference resources, such as FlashAttention [Dao et al., 2022] and FlexAttention [Dong et al., 2025]. While GTs with linear time scaling exist, we note that quadratic attention has been successfully applied in GTs for real-world tasks, such as the works of [Abramson et al., 2024, Wang et al., 2025] on protein folding and [Price et al., 2025] on weather forecasting. Moreover, linear attention variants can often be understood as approximations of quadratic attention, thereby leveraging theoretical results obtained for quadratic attention GTs [Xing et al., 2024, Wu et al., 2022, Choromanski et al., 2021].

# B  Background

Here, we provide background material on various concepts and definitions used in our work.

**Basic notations**  Let $\mathbb{N} := \{1, 2, \ldots\}$ and $\mathbb{N}_0 := \mathbb{N} \cup \{0\}$. The set $\mathbb{R}^+$ denotes the set of non-negative real numbers. For a set $X$, $A \subset X$ denotes the *strict* subset and $A \subseteq X$ denotes the subset. For $n \in \mathbb{N}$, let $[n] := \{1, \ldots, n\} \subset \mathbb{N}$. We use $\{\!\{\ldots\}\!\}$ to denote multisets, i.e., the generalization of sets allowing for multiple, finitely many instances for each of its elements. For two non-empty sets $X$ and $Y$, let $Y^X$ denote the set of functions from $X$ to $Y$. Given a set $X$ and a subset $A \subset X$, we define the indicator function $1_A \colon X \to \{0, 1\}$ such that $1_A(x) = 1$ if $x \in A$, and $1_A(x) = 0$ otherwise. Let $\boldsymbol{M}$ be an $n \times m$ matrix, $n > 0$ and $m > 0$, over $\mathbb{R}$, then $\boldsymbol{M}_{i,\cdot}$, $\boldsymbol{M}_{\cdot,j}$, $i \in [n]$, $j \in [m]$, are the $i$th row and $j$th column, respectively, of the matrix $\boldsymbol{M}$. We denote with $\mathsf{set}(\boldsymbol{M})$ the set of rows of $\boldsymbol{M}$. Let $\boldsymbol{N}$ be an $n \times n$ matrix, $n > 0$, then the *trace* $\mathrm{Tr}(\boldsymbol{N}) := \sum_{i \in [n]} N_{ii}$. In what follows, $\boldsymbol{0}$ denotes an all-zero vector with an appropriate number of components.

**Graphs**  An *(undirected) graph* $G$ is a pair $(V(G), E(G))$ with *finite* sets of *vertices* $V(G)$ and *edges* $E(G) \subseteq \{\{u, v\} \subseteq V(G) \mid u \neq v\}$. *vertices* or *nodes* $V(G)$ and *edges* $E(G) \subseteq \{\{u, v\} \subseteq V(G) \mid u \neq v\}$. The *order* of a graph $G$ is its number $|V(G)|$ of vertices. If not stated otherwise, we set $n := |V(G)|$ and call $G$ an *$n$-order graph*. We denote the set of all $n$-order (undirected) graphs by $\mathcal{G}_n$ and the set of all (undirected) graphs up to $n$ vertices by $\mathcal{G}_{\leq n}$. In a *directed graph*, we define $E(G) \subseteq V(G)^2$, where each edge $(u, v)$ has a direction from $u$ to $v$. Given a directed graph $G$ and vertices $u, v \in V(G)$, we say that $v$ is a *child* of $u$ if $(u, v) \in E(G)$. A (directed) graph $G$ is called *connected* if, for any $u, v \in V(G)$, there exist $r \in \mathbb{N}$ and $\{u_1, \ldots, u_r\} \subseteq V(G)$, such that $(u, u_1), (u_1, u_2), \ldots, (u_r, v) \in E(G)$, and analogously for undirected graphs by replacing directed edges with undirected ones. We say that a graph $G$ is *disconnected* if it is not connected. For a graph $G$ and an edge $e \in E(G)$, we denote by $G \setminus e$ the *graph induced by removing* edge $e$ from $G$. For an $n$-order graph $G \in \mathcal{G}_n$, assuming $V(G) = [n]$, we denote its *adjacency matrix* by $\boldsymbol{A}(G) \in \{0, 1\}^{n \times n}$, where $\boldsymbol{A}(G)_{vw} = 1$ if and only if $\{v, w\} \in E(G)$. The *neighborhood* of a vertex $v \in V(G)$ is denoted by $N_G(v) := \{u \in V(G) \mid \{v, u\} \in E(G)\}$, where we usually omit the subscript for ease of notation, and the *degree* of a vertex $v$ is $|N_G(v)|$. A graph $G$ is a *tree* if connected, but $G \setminus e$ is disconnected for any $e \in E(G)$. A tree or a disjoint collection of trees is known as a *forest*.

A *rooted tree* $(G, r)$ is a tree where a specific vertex $r$ is marked as the *root*. For a rooted (undirected) tree, we can define an implicit direction on all edges as pointing away from the root; thus, when we refer to the *children* of a vertex $u$ in a rooted tree, we implicitly consider this directed structure. For $S \subseteq V(G)$, the graph $G[S] := (S, E_S)$ is the *subgraph induced by $S$*, where $E_S := \{(u, v) \in E(G) \mid u, v \in S\}$. A *(vertex-)labeled graph* is a pair $(G, \ell_G)$ with a graph $G = (V(G), E(G))$ and a (vertex-)label function $\ell_G \colon V(G) \to \Sigma$, where $\Sigma$ is an arbitrary countable label set. For a vertex $v \in V(G)$, $\ell_G(v)$ denotes its *label*. A Boolean *(vertex-)d-labeled graph* is a pair $(G, \ell_G)$ with a graph $G = (V(G), E(G))$ and a label function $\ell_G \colon V(G) \to \{0, 1\}^d$. We denote the set of all $n$-order Boolean $d$-labeled graphs as $\mathcal{G}_{n,d}^{\mathbb{B}}$. An *attributed graph* is a pair $(G, a_G)$ with a graph $G = (V(G), E(G))$ and an (vertex-)attribute function $a_G \colon V(G) \to \mathbb{R}^{1 \times d}$, for $d > 0$. That is, unlike labeled graphs, vertex annotations may come from an uncountable set. The *attribute* or *feature* of $v \in V(G)$ is $a_G(v)$. We denote the class of all $n$-order graphs with $d$-dimensional, real-valued vertex features by $\mathcal{G}_{n,d}^{\mathbb{R}}$.

Two graphs $G$ and $H$ are *isomorphic* if there exists a bijection $\varphi \colon V(G) \to V(H)$ that preserves adjacency, i.e., $(u, v) \in E(G)$ if and only if $(\varphi(u), \varphi(v)) \in E(H)$. In the case of labeled graphs, we additionally require that $\ell_G(v) = \ell_H(\varphi(v))$ for $v \in V(G)$. Moreover, we call the equivalence classes induced by $\simeq$ *isomorphism types* and denote the isomorphism type of $G$ by $\tau(G)$. A *graph class* is a set of graphs closed under isomorphism. Given two graphs $G$ and $H$ with disjoint vertex sets, we denote their disjoint union by $G \dot{\cup} H$.

## B.1  Transformers

Here, we will introduce attention with an additive attention bias and the transformer architecture.

**Definition 5** (Attention (with bias)). Let $\boldsymbol{Q}, \boldsymbol{K}, \boldsymbol{V} \in \mathbb{R}^{n \times d}$ and $\boldsymbol{B} \in \mathbb{R}^{n \times n}$, with $n, d \in \mathbb{N}^+$. We define biased attention as

$$\mathsf{Attention}(\boldsymbol{Q}, \boldsymbol{K}, \boldsymbol{V}, \boldsymbol{B}) := \mathsf{softmax}\big(d^{-\frac{1}{2}} \cdot \boldsymbol{Q}\boldsymbol{K}^T + \boldsymbol{B}\big)\boldsymbol{V},$$

where $\mathsf{softmax}$ is applied row-wise and defined, for a vector $\boldsymbol{x} \in \mathbb{R}^{1 \times n}$, as

$$\mathsf{softmax}(\boldsymbol{x}) := \left[ \frac{\exp(\boldsymbol{x}_1)}{\sum_{i \in [n]} \exp(\boldsymbol{x}_i)} \quad \cdots \quad \frac{\exp(\boldsymbol{x}_n)}{\sum_{i \in [n]} \exp(\boldsymbol{x}_i)} \right].$$

**Definition 6** (Multi-head attention (with bias)). Let $\boldsymbol{X} \in \mathbb{R}^{n \times d}$, $\boldsymbol{B} \in \mathbb{R}^{n \times n \times h}$, and let $\boldsymbol{W}_Q, \boldsymbol{W}_K, \boldsymbol{W}_V \in \mathbb{R}^{d \times d}, \boldsymbol{W}_O \in \mathbb{R}^{d \times d}$ be learnable parameters, with $n, d \in \mathbb{N}^+$. Let $h \in \mathbb{N}^+$ be the number of heads, such that a $d_h \in \mathbb{N}^+$ for which $d = h \cdot d_h$. We call $d_h$ the *head dimension* and define $h$-head attention over $\boldsymbol{X}$ as

$$\mathsf{MHA}(\boldsymbol{X}, \boldsymbol{B}) := \begin{bmatrix} \tilde{\boldsymbol{X}}_1 & \dots & \tilde{\boldsymbol{X}}_h \end{bmatrix} \boldsymbol{W}_O,$$

where, for all $i \in [h]$,

$$\tilde{\boldsymbol{X}}_i := \mathsf{Attention}(\boldsymbol{X}\boldsymbol{W}_Q^{(i)}, \boldsymbol{X}\boldsymbol{W}_K^{(i)}, \boldsymbol{X}\boldsymbol{W}_V^{(i)}, \boldsymbol{B}_i),$$

with $\boldsymbol{B}_i \in \mathbb{Q}^{n \times n}$ denoting the attention bias for the $i$-head, indexed along the third dimension of $\boldsymbol{B}$, $\boldsymbol{W}_Q^{(i)}, \boldsymbol{W}_K^{(i)}, \boldsymbol{W}_V^{(i)} \in \mathbb{R}^{d \times d_h}$, and

$$\boldsymbol{W}_Q := \left[ \boldsymbol{W}_Q^{(1)}, \dots, \boldsymbol{W}_Q^{(h)}m \right]$$

$$\boldsymbol{W}_K := \left[ , \boldsymbol{W}_K^{(1)}, \dots, \boldsymbol{W}_K^{(h)} \right],$$

$$\boldsymbol{W}_V := \left[ \boldsymbol{W}_V^{(1)}, \dots, \boldsymbol{W}_V^{(h)} \right].$$

**Definition 7** (Two-layer MLP). Let $\boldsymbol{X} \in \mathbb{R}^{n \times d}$ with $n, d, d_f \in \mathbb{N}^+$, where $d_f$ is the hidden dimension. We define a two-layer MLP as

$$\mathsf{MLP}(\boldsymbol{x}) := \sigma(\boldsymbol{x}\boldsymbol{W}_1)\boldsymbol{W}_2,$$

where $\mathsf{MLP}$ is applied independently to each row $\boldsymbol{x} \in \mathbb{R}^{1 \times d}$ in $\boldsymbol{X}$. Here, $\boldsymbol{W}_1 \in \mathbb{R}^{d \times d_f}$ is the in-projection matrix, $\boldsymbol{W}_2 \in \mathbb{R}^{d_f \times d}$ is the out-projection matrix, and $\sigma : \mathbb{R} \to \mathbb{R}$ is an element-wise activation function such as GELU [Hendrycks and Gimpel, 2016].

**Definition 8** (Transformer architecture). Let $\boldsymbol{X} \in \mathbb{R}^{n \times d}$ be a token matrix and $\boldsymbol{B} \in \mathbb{R}^{n \times n}$ be an attention bias, with $n, d \in \mathbb{N}^+$. The $t + 1$-th transformer layer updates token representations $\boldsymbol{X}^t \in \mathbb{R}^{n \times d}$ as

$$\boldsymbol{X}' \leftarrow \boldsymbol{X}^t + \mathsf{MHA}(\mathsf{LayerNorm}(\boldsymbol{X}^t), \boldsymbol{B}),$$

$$\boldsymbol{X}^{t+1} \leftarrow \boldsymbol{X}' + \mathsf{MLP}(\mathsf{LayerNorm}(\boldsymbol{X}')),$$

where $\mathsf{MLP}$ is defined in Definition 7.

## B.2 Extended notation for the theoretical analysis of the GDT

Here, we introduce some notation for the GDT that we will use in our theoretical analysis.

**Learnable parameters**   We give an overview of all learnable parameters of the GDT in Table 9. In practice, node and edge features are typically present as integers or continuous feature vectors, and we embed them using learnable MLPs. We refer to such parameters as *embedding parameters*. Note that in Table 9, we exclude embedding parameters, as for simplicity, we assume in our framework that node and edge features are already embedded.

Let $d, d_f, T, h \in \mathbb{N}^+$ denote the number of embedding dimensions, the number of hidden dimensions, the number of layers, and the number of attention heads, respectively. Then, the number of learnable parameters (excluding embedding parameters) is given by

$$\textbf{\# params} = 3d + d^2 + dd_f + d_f h + h + 3Td^2 + 2Tdd_f$$
$$= (3T + 1)d^2 + (2T + 1)dd_f + 3d + (d_f + 1)h.$$

We denote the complete set of learnable parameters in Table 9 with $\Theta(d, d_f, T, h)$.

Table 9: Overview of learnable parameters in the GDT, excluding embedding parameters. Here, $d \in \mathbb{N}^+$ is the embedding dimension, $d_f \in \mathbb{N}^+$ is the hidden dimension, and $T \in \mathbb{N}^+$ is the number of layers. The suffix $\times T$ indicates that the parameters occur in each of the $T$ layers.

| Params. | Dims. | Module | Description |
|---|---|---|---|
| $\ell_V(\texttt{[cls]})$ | $1 \times d$ | | Learnable embedding for the [cls] token |
| $\ell_E(\texttt{[cls]}, \cdot)$ | $1 \times d$ | | Learnable embedding for the out-going edges from the [cls] token |
| $\ell_E(\cdot, \texttt{[cls]})$ | $1 \times d$ | Token embeddings | Learnable embedding for the in-coming edges to the [cls] token |
| $\boldsymbol{W}_P$ | $d \times d$ | | Weight matrix for the node-level PEs |
| $\rho$ | $(d \times d_f, d_f \times h)$ | Attention bias | MLP applied to the edge embeddings |
| $\boldsymbol{W}_U$ | $1 \times h$ | | Weight matrix for the relative PEs in the attention bias |
| $\boldsymbol{W}_Q \times T$ | $d \times d$ | | Query weight matrix in multi-head attention |
| $\boldsymbol{W}_K \times T$ | $d \times d$ | MHA $_{\text{Definition 6}}$ | Key weight matrix in multi-head attention |
| $\boldsymbol{W}_V \times T$ | $d \times d$ | | Value weight matrix in multi-head attention |
| $\boldsymbol{W}_1 \times T$ | $d \times d_f$ | MLP $_{\text{Definition 7}}$ | Input projection of the MLP |
| $\boldsymbol{W}_2 \times T$ | $d_f \times d$ | | Output projection of the MLP |

**Graph transformer representations** Here, we introduce some short-hand notation for graph transformer representations. Given token matrix $\boldsymbol{X}$ and attention bias $\boldsymbol{B}$, we write $\hat{\boldsymbol{X}}^t(v)$ to denote the representation of node $v \in V(G)$ after $t$ transformer layers with $\boldsymbol{X}$ and $\boldsymbol{B}$ as input. Note that, since we fix an arbitrary order of the nodes, if $v$ is the $i$-th node in this order, $\hat{\boldsymbol{X}}^t(v) = \hat{\boldsymbol{X}}_i^t$.

### B.3 Extended notation for the theoretical analysis of PEs

Here, we introduce the notations used by Zhang et al. [2024] in their paper on PE expressiveness. We adapt this notation to fit LPE [Müller and Morris, 2024], SAN [Kreuzer et al., 2021], and SignNet [Lim et al., 2023] and use it in our proofs in Appendix D. We introduce the notation for the color refinement algorithm and propose one for each PE. The respective algorithms for BasisNet [Lim et al., 2023] and SPE [Huang et al., 2024] are given by Zhang et al. [2024].

**Definition 9.** [Zhang et al., 2024] We call any graph invariant a $k$-*dim color mapping*. The family of $k$-dim color mappings is denoted by $M_k$. Each color mapping defines an equivalence relation $\sim_\chi$ between rooted graphs $G_u, H_v$ marking $k$ vertices and $G^u \sim_\chi H^v$ iff $\chi_G(u) = \chi_H(v)$. Further, we denote the family of $k$-dim spectral color mappings by $M_k^\Lambda$. Similar to $M_k$ the family of spectral color mappings is obtained from the color mappings acting on $\{(G_u, \lambda) : G_u \in G, \lambda \in \Lambda^M(G)\}$ where $\Lambda^M(G)$ denotes the eigenvalues of a matrix $\boldsymbol{M}$.

**Definition 10.** [Zhang et al., 2024] A function $T$ mapping from $M_{k_1}$ to $M_{k_2}$ is called a color transform. We assume that all color transforms are order-preserving with respect to color mappings. Given $T(\chi) \preceq \chi$, a color transform is also called color refinement, and $T^t$ denotes the $t$ times composition of $T$. In addition, $T^\infty$ is the stable color refinement obtained from $T^{t'}$ with $t'$ the smallest integer where further iterations do not induce a different partition of the underlying nodes in a graph, resulting in $T \circ T^\infty \equiv T^\infty$. Following Zhang et al. [2024] $T^\infty$ is well defined.

A coloring algorithm is then formed by concatenating a stable color transform $T^\infty : M_k \to M_k$ and a pooling function $U : M_k \to M_0$.

**Definition 11.** We say that color mappings $\chi_1, \chi_2$ are equivalent given that $G^u \sim_{\chi_1} H^v$ iff $G^u \sim_{\chi_2} H^v$. Furthermore, we say that a color mapping $\chi_1$ is finer/more expressive than $\chi_2$ if $G^u \sim_{\chi_1} H^v \Rightarrow G^u \sim_{\chi_2} H^v$, noted by $\preceq$.

**Lemma 12.** *[Zhang et al., 2024] Let* $T_1, T_2 : M_{k_1} \to M_{k_2}$ *and* $U_1, U_2 : M_{k_2} \to M_{k_3}$ *be color refinements. If* $T_1 \preceq T_2$ *and* $U_1 \preceq U_2$ *then* $U_1 \circ T_1 \preceq U_2 \circ T_2$.

**Lemma 13.** *[Zhang et al., 2024] Let* $T_1 : M_{k_1} \to M_{k_1}$ *and* $T_2 : M_{k_2} \to M_{k_2}$ *be color refinements and* $T^\infty : M_k \to M_k$ *be the stable refinement of* $M_k$. *Further let* $U_1 : M_{k_0} \to M_{k_1}$ *and* $U_2 : M_{k_1} \to M_{k_2}$ *be color refinements. Then it follows. If* $T_2 \circ U_2 \circ T_1^\infty \circ U_1 \equiv U_2 \circ T_1^\infty \circ U_1$ *then* $U_2 \circ T_1^\infty \circ U_1 \preceq T_2^\infty \circ U_2 \circ U_1$.

The two lemmas above provide a straightforward approach to determining whether architecture $A_1$ is more expressive than architecture $A_2$ [Zhang et al., 2024]. To prove that $A_1$ is more expressive than $A_2$, we show that $T_2 \circ T_1^\infty \equiv T_1^\infty$ holds, with $T_i$ being the color refinement of $A_i$ respectively.

**Definition 14.** [Zhang et al., 2024] We define the following color refinements corresponding to the induced refinements of each algorithm. We define global pooling as providing an injective coloring of a multiset using a hash function. In this case, we consider the multiset over nodes in a graph. Other pooling operations are defined below.

**Global pooling**: Define $T_{\text{GP}} \colon M_1 \to M_0$ and $\chi \in M_1$ such that for a graph $G$ and a color mapping $\chi \in M_1$ it holds:
$$[T_{\text{GP}}(\chi)](G) = \mathsf{hash}(\{\!\!\{\chi_G(u) \colon u \in V(G)\}\!\!\}).$$

The 1-WL refinement gives us the 1-WL coloring update, generalized to all nodes in the graph rather than just neighboring graphs.

**1-WL refinement**: Given $T_{\text{WL}} \colon M_1 \to M_1$ such that for any choice of $\chi \in M_1$:
$$[T_{\text{WL}}(\chi)]_G(u) = \mathsf{hash}(\chi_G(u), \{\!\!\{(\chi_G(v), \mathsf{atp}_G(u,v)) \colon v \in V(G)\}\!\!\}).$$

We then define one- and two-dimensional spectral pooling, which allows for pooling over distinct eigenvalues similar to the global pooling refinement.

**Spectral Pooling**: Define $T_{\text{SP2}} \colon M_2^\Lambda \to M_1$ and $T_{\text{SP1}} \colon M_1^\Lambda \to M_1$ such that for $\chi \in M_2^\Lambda$ and $\chi' \in M_1^\Lambda$:
$$[T_{\text{SP1}}(\chi')]_G(u) = \mathsf{hash}(\{\!\!\{\chi'_G(\lambda, u) \colon \lambda \in \Lambda^M(G)\}\!\!\}).$$
$$[T_{\text{SP2}}(\chi)]_G(u,v) = \mathsf{hash}(\{\!\!\{\chi_G(\lambda, u, v) \colon \lambda \in \Lambda^M(G)\}\!\!\}).$$

A pooling variant without the spectrum is denoted by $T_{P2}$ and $T_{P1}$.

To allow for an examination of BasisNet and SPE, we consider the 2-IGN refinement. This refinement is obtained from evaluating the expressiveness of a 2-IGN and its basis functions as defined by Maron et al. [2019a].

**2-IGN refinement**: With $T_{\text{IGN}} \colon M_2 \to M_2$, $\chi \in M_2$ as any color mapping and $\delta_{uv}(c) = c$ given $u = v$, otherwise 0:
$$\begin{aligned}
[T_{\text{IGN}}(\chi)]_G(u,v) = \mathsf{hash}(&\chi_G(u,v), \chi_G(u,u), \chi_G(v,v), \chi_G(v,u), \delta_{uv}(\chi_G(u,u)), \\
&\{\!\!\{\chi_G(u,w) \colon w \in V(G)\}\!\!\}, \{\!\!\{\chi_G(w,u) \colon w \in V(G)\}\!\!\}, \\
&\{\!\!\{\chi_G(v,w) \colon w \in V(G)\}\!\!\}, \{\!\!\{\chi_G(w,v) \colon w \in V(G)\}\!\!\}, \\
&\{\!\!\{\chi_G(w,w) \colon w \in V(G)\}\!\!\}, \{\!\!\{\chi_G(w,x) \colon w, x \in V(G)\}\!\!\}, \\
&\delta_{uv}(\{\!\!\{\chi_G(u,w) \colon w \in V(G)\}\!\!\}), \delta_{uv}(\{\!\!\{\chi_G(w,u) \colon w \in V(G)\}\!\!\}), \\
&\delta_{uv}(\{\!\!\{\chi_G(w,w) \colon w \in V(G)\}\!\!\}), \delta_{uv}(\{\!\!\{\chi_G(w,x) \colon w, x \in V(G)\}\!\!\})).
\end{aligned}$$

Further, we use BasisNet pooling refinement and Siamese IGN refinement to describe the BasisNet computation process.

**BasisNet Pooling**: Given $T_{\text{BP}} \colon M_2^\Lambda \to M_1^\Lambda$ and $\chi \in M_2^\Lambda$:
$$\begin{aligned}
[T_{\text{BP}}(\chi)]_G(\lambda, u) = \mathsf{hash}(&\chi_G(\lambda, u, v), \{\!\!\{\chi_G(\lambda, u, v) \colon v \in V(G)\}\!\!\}, \\
\{\!\!\{\chi_G(\lambda, v, u) \colon v \in V(G)\}\!\!\}, &\{\!\!\{\chi_G(\lambda, v, v) \colon v \in V(G)\}\!\!\}, \{\!\!\{\chi_G(\lambda, v, w) \colon v, w \in V(G)\}\!\!\}).
\end{aligned}$$

**Siamese IGN refinement**: Given $T_{\text{SIAM}} \colon M_2^\Lambda \to M_2^\Lambda$ and $\chi \in M_2^\Lambda$:
$$[T_{\text{SIAM}}(\chi)]_G(\lambda, u, v) = [T_{\text{IGN}}(\chi(\lambda, \cdot, \cdot))]_G(u,v).$$

We further provide additional color refinement algorithms based on the encodings introduced throughout this work: Given the initial color refinement of SAN as $\chi_{\text{SAN}}(\lambda, u) = (\lambda_1, \ldots, \lambda_m, v_{1:m}^u)$ where $\lambda_i$ denote the eigenvalues and $v^u$ the eigenvector of the graph Laplacian associated to the node $u$. Then we define the SAN color refinement alongside the existing refinements as follows:
$$[T_{\text{SAN}}(\chi)]_G = T_{\text{GP}} \circ T_{\text{ENC}} \circ T_L(\chi_{\text{SAN}}).$$

with $T_{\text{ENC}}$ denoting the transformer encoder layer. The complete BasisNet refinement is given by the concatenation of refinements given in Definition Definition 14:
$$[T_{\text{BasisNet}}]_G = T_{\text{GP}} \circ T_{\text{WL}} \circ T_{\text{SP1}} \circ T_{\text{BP}} \circ T_{\text{SIAM}}(\chi_{\text{Basis}}).$$

Following the definition of BasisNet as a color refinement in Definition 14 and assuming a message passing GNN for $\rho$, the color refinement of SignNet using the initial SignNet color refinement $\chi\text{Sign}(\lambda, u, w) = (\lambda, \boldsymbol{V}^u, \boldsymbol{V}^w)$ and $T_\phi \colon M_2^\Lambda \to M_2^\Lambda$ is given by:

$$[T_{\text{Sign}(\chi)}]_G = T_{\text{GP}} \circ T_{\text{WL}}^\infty \circ T_{\text{SP2}} \circ T_\phi(\chi\text{Sign}),$$

where $\boldsymbol{V}^u$ denotes the eigenvector associated to the eigenvalue $\lambda$ and the node $u$ and $T_\phi$ be the refinement depending on the choice of $\phi$. However, we note $T_{\text{IGN}} \preceq T_\phi$, by definition of SignNet and BasisNet.

The refinement for the LPE encoding is given similarly to the BasisNet and SPE refinement by replacing $\rho$ with a color refinement that is 1-WL expressive. Furthermore, we assume $\phi$ to be a spectral color refinement with expressiveness up to a 2-IGN. Common choices for $\phi$ include MLPs or GNNs known to be less expressive than a 2-IGN. With initial colorings $\chi_{\text{LPE}}(\lambda, u, w) = (\lambda, \boldsymbol{V}^u, \boldsymbol{V}^w)$ and $T_\phi^{\text{LPE}} \colon M_2^\Lambda \to M_2^\Lambda$ the refinement follows:

$$[T_{LPE(\chi)}]_G = T_{\text{GP}} \circ T_{\text{WL}}^\infty \circ T_{\text{SP2}} \circ T_\phi^{\text{LPE}}(\chi_{\text{LPE}}),$$

where $\boldsymbol{V}^u$ denotes the eigenvector associated to node $u$.

## B.4 Positional encodings

In the following, we define positional encodings.

### RWSE

**Definition 15.** Let $\boldsymbol{R} := \boldsymbol{D}^{-1}\boldsymbol{A}$ be the random walk matrix, with $\boldsymbol{D}$ denoting the degree matrix and $\boldsymbol{A}$ the adjacency matrix of a graph $G$. The random walk structural encodings (RWSE) are given by:

$$\boldsymbol{P}_{i,i} = [\boldsymbol{I}, \boldsymbol{R}, \boldsymbol{R}^2, \ldots, \boldsymbol{R}^{k-1}]_{i,i}.$$

**Definition 16.** Let $\boldsymbol{P}_{i,i}$ be the RWSE encoding vector and $\boldsymbol{F} \colon \mathbb{R}^k \to \mathbb{R}^d$ a MLP with two layers, and $d$ denoting the encoding dimension. Then the RWSE encoding is computed by $\boldsymbol{F}(\boldsymbol{P}_{i,i})$ and denoted by $\mathbf{P}_k^{\text{RW}}(G)$ for a graph $G$ with random walk length $k$.

### RRWP

**Definition 17.** Let $\boldsymbol{R} := \boldsymbol{D}^{-1}\boldsymbol{A}$ with $\boldsymbol{D}$ as the diagonal degree matrix and $\boldsymbol{A}$ as the adjacency matrix, be the random walk operator, and $k$ the maximum length of the random walk. Then the relative random walk probabilities (RRWP) are defined as:

$$\boldsymbol{P}_{i,j} = [\boldsymbol{I}, \boldsymbol{R}, \boldsymbol{R}^2, \ldots, \boldsymbol{R}^{k-1}]_{i,j}.$$

The initial node encoding $p_0$ is then defined as $\boldsymbol{R}_{ii}$ for each node $i$ in the graph.

**Definition 18** (RRWP Encoding Computation)**.** Let $\boldsymbol{P}$ be the RRWP encoding tensor and MLP : $\mathbb{R}^k \to \mathbb{R}^d$, where $d$ denotes the encoding dimension, be a multi-layer neural network. Then the encoding $\text{MLP}(\boldsymbol{P}_{i,j,:})$ is computed element-wise by the multi-layer neural network. RRWP is then denoted by $\mathbf{P}_k^{\text{RR}}(G)$ for a graph $G$ with random walk length $k$

**Spectral Attention Networks (SAN)** Kreuzer et al. [2021] propose incorporating eigenvalues and eigenvectors into a positional-encoding neural network. SAN encoding can be computed using row-wise neural networks by selecting the $k$ lowest eigenvalues and their associated eigenvectors.

**Definition 19.** Let $\phi \colon \mathbb{R} \to \mathbb{R}$ be a linear layer and $\rho \colon \mathbb{R} \to \mathbb{R}$ be a transformer encoder layer with sum aggregation. Further $\boldsymbol{V}_i$ denotes the $i$-th column of the eigenvector matrix $\boldsymbol{V}$. Then the SAN encoding is defined as follows:

$$\text{SAN}(\boldsymbol{V}, \lambda) = \rho([\phi(\boldsymbol{V}_1, \lambda) \ldots \phi(\boldsymbol{V}_k, \lambda)]).$$

A generalization of the SAN encoding is given by the LPE encoding of Müller and Morris [2024] in Definition 23.

**SignNet**   Since the computation of eigenvectors using the eigenvector decomposition is not sign invariant, and both $\boldsymbol{V}_i$ and $-\boldsymbol{V}_i$ are valid eigenvectors of the graph Laplacian Lim et al. [2023] propose the construction of a sign-invariant encoding using eigenvector information. Considering the $k$ smallest eigenvalues and associated eigenvectors from the eigenvalue decomposition, SignNet computes the corresponding encoding using a neural network architecture.

**Definition 20.** Let $\phi_1, \ldots \phi_k \colon \mathbb{R} \to \mathbb{R}$ and $\rho \colon \mathbb{R} \to \mathbb{R}$ be permutation equivariant neural networks from vectors to vectors. Then the SignNet encodings are computed using:

$$\text{SignNet}(\boldsymbol{V}) = \rho([\phi_1(\boldsymbol{V}_1) + \phi_1(-\boldsymbol{V}_1) \ldots \phi_k(\boldsymbol{V}_k) + \phi_k(-\boldsymbol{V}_k)]).$$

Commonly, $\phi_1, \ldots \phi_k$ are selected as element-wise MLPs or DeepSets [Lim et al., 2023] and $\rho$ as a GIN with sum aggregation and the adjacency matrix of the original graph.

**BasisNet**   Proposed as an extension of SignNet by Lim et al. [2023], BasisNet encodings provide an encoding invariant to the basis of eigenspaces obtained from the graph Laplacian. Since the orthogonal group $O(1)$ denotes sign invariance, BasisNet also incorporates sign invariance.

**Definition 21.** Let $\boldsymbol{V}_i$ denote the orthonormal basis of an $d_i$-dimensional eigenspace of the graph Laplacian. Further, $l$ denotes the number of eigenspaces. Given unrestricted neural networks $\phi_{d_1}, \ldots \phi_{d_l} \colon \mathbb{R} \to \mathbb{R}$, shared across the subspaces with the same dimension $d_i$, and a permutation equivariant neural network $\rho \colon \mathbb{R} \to \mathbb{R}$ BasisNet encodings are computed the following:

$$\text{BasisNet}(\boldsymbol{V}) = \rho([\phi_{d_1}(\boldsymbol{V}_1 \boldsymbol{V}_1^T) \ldots \phi_{d_l}(\boldsymbol{V}_l \boldsymbol{V}_l^T)]).$$

Implementation wise Lim et al. [2023] propose 2-IGNs [Maron et al., 2019a] for $\phi_{d_i}$ and a FFN with sum aggregation for $\rho$. They note that all neural networks could be replaced with $k$-IGNs; however, they deemed it infeasible for efficient computation. This reduces the computation to the following with $\rho \colon \mathbb{R} \to \mathbb{R}$ and $\text{IGN}_{d_i} \colon \mathbb{R}^{n^2} \to \mathbb{R}^n$ denoting an IGN from matrices to vectors:

$$\text{BasisNet}(\boldsymbol{V}) = \text{FFN}([\text{IGN}_{d_1}(\boldsymbol{V}_1 \boldsymbol{V}_1^T) \ldots \text{IGN}_{d_l}(\boldsymbol{V}_l \boldsymbol{V}_l^T)]).$$

**SPE**   Following notation from Huang et al. [2024], the SPE encoding is computed using the $k$ smallest eigenvalues and associated eigenvectors obtained from an eigenvalue decomposition. With sufficient conditions for neural networks $\phi_1, \ldots, \phi_k$ and $\rho$, SPE is stable with respect to the graph Laplacian.

**Definition 22.** Given $\phi_1 \ldots \phi_k \colon \mathbb{R} \to \mathbb{R}$ as Lipschitz continuous, equivariant FFNs and $\rho \colon \mathbb{R} \to \mathbb{R}$ a Lipschitz continuous, permutation equivariant neural network. Then the SPE encoding is computed by:

$$\text{SPE}(\boldsymbol{V}, \lambda) = \rho([\boldsymbol{V} \text{diag}(\phi_1(\lambda))\boldsymbol{V}^T \ldots \boldsymbol{V} \text{diag}\phi_k(\lambda))\boldsymbol{V}^T]),$$

with $\phi_1, \ldots \phi_k$ and $\rho$ applied row wise. Further, we denote the SPE embedding on a graph $G$ by $\mathbf{P}_k^{\text{SPE}}$.

Commonly, $\phi_i$ is considered an element-wise MLP, and $\rho$ is a GIN using the adjacency matrix of the original graph. Huang et al. [2024] propose to split tensor

$$\boldsymbol{Q} = [\boldsymbol{V} \text{diag}(\phi_1(\lambda))\boldsymbol{V}^T \ldots \boldsymbol{V} \text{diag}\phi_k(\lambda))\boldsymbol{V}^T] \in \mathbb{R}^{n \times n \times l}$$

into $n$ matrices of shape $n \times l$ which are then passed into the GIN $\rho$ and aggregated using sum aggregation into a single $n \times d$ matrix.

**LPE**   Initially introduced by Kreuzer et al. [2021] and generalized by Müller and Morris [2024], the LPE encodings are computed similarly to the previously introduced SPE encodings. Instead of using the eigenvector matrix $\boldsymbol{V} \in \mathbb{R}^{n \times l}$, each $i$-th column consisting of one eigenvector denoted by $\boldsymbol{V}_i \in \mathbb{R}^l$ is used.

**Definition 23.** Let $\phi \colon \mathbb{R}^2 \to \mathbb{R}^k$ be a row-wise applied FFN and $\rho \colon \mathbb{R} \to \mathbb{R}$ a permutation equivariant network. Furthermore, $\epsilon \in \mathbb{R}$ denotes a learnable parameter. Then the LPE are given by:

$$\text{LPE}(\boldsymbol{V}, \lambda) = \rho([\phi(\boldsymbol{V}_1^T, \lambda + \epsilon), \ldots \phi(\boldsymbol{V}_k^T, \lambda + \epsilon)]).$$

Setting $\epsilon = 0$ reduces the LPE to the encoding provided by Kreuzer et al. [2021]. As proposed by Müller and Morris [2024] $\rho$ sums the input tensor concerning its first dimension and applies an FFN. In contrast to the SAN encoding, no transformer encoder is used to compute the encoding. Similar to previous embeddings, we denote LPE embeddings for a graph $G$ by $\mathbf{P}_k^{\text{LPE}}$ with $k$ as the number of eigenvalues used.

# C Proving that the GDT can simulate the GD-WL

Here, we prove Theorem 1. Concretely, we formally state and prove both statements in Theorem 1 in Appendix C.1 and Appendix C.2, respectively.

## C.1 Lower-bound on the expressivity of the GDT

Here, we prove that we can compute the GD-WL [Zhang et al., 2023] with the GDT, our GT defined in Section 2. Concretely, given a graph $G := (V(G), E(G))$, recall from Section 2.3 the GD-WL as updating the color $\chi_G^t(v)$ of node $v \in V(G)$, as

$$\chi_G^{t+1}(v) := \mathsf{hash}\big(\{\!\!\{(d_G(v,w), \chi_G^t(w)) : w \in V(G)\}\!\!\}\big),$$

where $d_G$ is a distance between nodes in $G$ and $\mathsf{hash}$ is an injective map. In the transformer, we will represent node colors as one-hot vectors of some arbitrary but fixed dimension $d$. Furthermore, we will incorporate pairwise distances through the attention bias. We then show that a single GT layer can compute the color update in Equation (1). We demonstrate this result by leveraging specific properties of softmax attention. For notational convenience, we will denote with $\boldsymbol{X}^t \in \{0,1\}^{L \times d}$ the one-hot color matrix of the GD-WL after $t$ iterations.

We begin by stating our main result. Afterwards, we develop our proof techniques and prove the result.

**Stating the main result**    We will formally state our theorem, showing that our GT can simulate the GD-WL. Afterwards, we give an overview of the proof, including the key challenges and ideas.

**Theorem 24.** *Let $G := (V(G), E(G))$ be a graph with $n \in \mathbb{N}^+$ nodes and node distance function $d_G : V(G)^2 \to \mathbb{Q}$. Let $d, d_f, T, h \in \mathbb{N}^+$ denote the number of embedding dimensions, the number of hidden dimensions, the number of layers, and the number of attention heads, respectively. Let $L := n + 1$ and let $\hat{\boldsymbol{X}}^0 \in \mathbb{R}^{L \times d}$ and $\boldsymbol{B} \in \mathbb{R}^{L \times L \times h}$ be initial token embeddings and attention bias constructed according to Section 2 using node distance $d_G$. Then, there exist weights for the parameters in $\Theta(d, d_f, T, h)$ such that $\hat{\boldsymbol{X}}^t = \boldsymbol{X}^t$, for all $t \geq 0$, an arbitrary but fixed $\mathsf{hash}$, and using $d_G$ as the distance function.*

**Proof overview**    The central problem we face when proving the above theorem is how to injectively encode the multisets in Equation (1) with softmax-attention. This is because softmax-attention computes a weighted mean, whereas existing results for encoding multisets use sums [Xu et al., 2019, Morris et al., 2019, Zhang et al., 2023]. Because these multisets are at the core of our proof, we formally introduce them here.

**Definition 25** (Distance-paired multisets). Given a graph $G := (V(G), E(G))$ with $n \in \mathbb{N}^+$ nodes and let $L := n + 1$, for each token $v \in V(G) \cup \{\texttt{[cls]}\}$, we construct a vector $\boldsymbol{v} \in \mathbb{Q}^{1 \times L}$ from the distances of $v$ to tokens in $V(G) \cup \{\texttt{[cls]}\}$, such that

$$\boldsymbol{v}_i := d_G(v, w_i),$$

where $w_i \in V(G)$, for $i \in [n]$, is the $i$-th token in an arbitrary but fixed ordering of nodes in $V(G)$ and $w_L$ is the $\texttt{[cls]}$ token. We fix the distance of $\texttt{[cls]}$ to all tokens as $\max_{v,w \in V(G)} d_G(v, w) + 1$. We represent node colors as one-hot vectors and stack them into a matrix $\boldsymbol{X} \in \{0,1\}^{L \times d}$ with $d \in \mathbb{N}^+$ and where $\boldsymbol{X}_L$, representing the color of the $\texttt{[cls]}$ token, receives a special color, not used by any node. We then write the distance-paired multiset in Equation (1) as

$$[\boldsymbol{v}]_{\boldsymbol{X}} := \{\!\!\{(\boldsymbol{v}_i, \boldsymbol{X}_i)\}\!\!\}_{i \in [L]}.$$

We can then restate the update of token $v \in V(G) \cup \{\texttt{[cls]}\}$ by the GD-WL as

$$\chi_G^{t+1}(v) := \mathsf{hash}([\boldsymbol{v}]_{\boldsymbol{X}^t}), \tag{3}$$

For notational convenience, for every $\boldsymbol{x} \in \mathsf{set}(\boldsymbol{X})$, we define $A(\boldsymbol{x}) := \{i \in [L] \mid \boldsymbol{X}_i = \boldsymbol{x}\}$ as the set of token indices with token representation $\boldsymbol{x}$. Further, we write

$$[\boldsymbol{v}]_{\boldsymbol{x}} := \{\!\!\{\boldsymbol{v}_i \mid i \in A(\boldsymbol{x})\}\!\!\}$$

and

$$[\boldsymbol{v}]_{\boldsymbol{x}, \boldsymbol{w}} := \{\!\!\{v + \boldsymbol{w}_j \mid v \in [\boldsymbol{v}]_{\boldsymbol{x}}, j \in [L]\}\!\!\},$$

again for notational convenience, where $\mathbf{w} \in \mathbb{Q}^{1 \times L}$ is the distance vector corresponding to another node $w \in V(G) \cup \{\texttt{[cls]}\}$.

Recall that we introduced the distance function $d_G$ into the attention via the attention bias $\boldsymbol{B}$. Now, to injectively encode distance-paired multisets, we want to prove that there exists weights $\boldsymbol{W}_Q, \boldsymbol{W}_K, \boldsymbol{W}_V$ such that for two tokens $v, w \in V(G) \cup \{\texttt{[cls]}\}$ with corresponding distance vectors $\boldsymbol{v}, \boldsymbol{w}$,

$$\mathsf{softmax}(\boldsymbol{X}(v)\boldsymbol{W}_Q(\boldsymbol{X}\boldsymbol{W}_K)^T + \boldsymbol{v})\boldsymbol{X}\boldsymbol{W}_V = \mathsf{softmax}(\boldsymbol{X}(w)\boldsymbol{W}_Q(\boldsymbol{X}\boldsymbol{W}_K)^T + \boldsymbol{w})\boldsymbol{X}\boldsymbol{W}_V,$$

if and only if $[\boldsymbol{v}]_{\boldsymbol{X}} = [\boldsymbol{w}]_{\boldsymbol{X}}$. Note that for simplicity, we omit the scaling factor in the attention and that we wrote $\boldsymbol{v}$ and $\boldsymbol{w}$ to indicate the corresponding row of $\boldsymbol{B}$ for tokens $v$ and $w$, respectively. We will now simplify, by setting $\boldsymbol{W}_Q = \boldsymbol{W}_K = \boldsymbol{0}$ and $\boldsymbol{W}_V = \boldsymbol{I}$ and arrive at the condition

$$\mathsf{softmax}(\boldsymbol{v})\boldsymbol{X} = \mathsf{softmax}(\boldsymbol{w})\boldsymbol{X} \iff [\boldsymbol{v}]_{\boldsymbol{X}} = [\boldsymbol{w}]_{\boldsymbol{X}}.$$

Here, we prove the above holds under mild conditions in the following lemma. Note that we split up the forward and backward directions of the lemma, as we will use different proof strategies for each direction.

**Lemma 26.** *Let $\boldsymbol{v}, \boldsymbol{w} \in \mathbb{Q}^{1 \times L}$ with $\max_i \boldsymbol{v}_i = \max_i \boldsymbol{w}_i$ and let $\boldsymbol{X} \in \{0,1\}^{L \times d}$ be a matrix whose rows are one-hot vectors, for some $L, d \in \mathbb{N}^+$. Further, we require $\boldsymbol{X}$ to have at least two distinct rows. Then,*

$$\mathsf{softmax}(\boldsymbol{v})\boldsymbol{X} = \mathsf{softmax}(\boldsymbol{w})\boldsymbol{X} \implies [\boldsymbol{v}]_{\boldsymbol{X}} = [\boldsymbol{w}]_{\boldsymbol{X}} \qquad (4)$$

*and*

$$\mathsf{softmax}(\boldsymbol{v})\boldsymbol{X} = \mathsf{softmax}(\boldsymbol{w})\boldsymbol{X} \impliedby [\boldsymbol{v}]_{\boldsymbol{X}} = [\boldsymbol{w}]_{\boldsymbol{X}}. \qquad (5)$$

As mentioned above, we will treat the forward and backward directions differently. The backward direction is fairly straightforward, seeing that $\mathsf{softmax}(\boldsymbol{v})\boldsymbol{X}$ is a function over $[\boldsymbol{v}]_{\boldsymbol{X}}$. For the forward direction, the idea is first to notice that the condition $[\boldsymbol{v}]_{\boldsymbol{X}} = [\boldsymbol{w}]_{\boldsymbol{X}}$, on the right side of Equation (4), is equivalent to comparing the multiset of distances paired with each distinct one-hot vector in $\boldsymbol{X}$ independently, as distinct one-hot vectors do not have common non-zero channels; see the following lemma for a precise statement of this property and see Appendix E for the proof.

**Lemma 27.** *Let $\boldsymbol{v}, \boldsymbol{w} \in \mathbb{Q}^{1 \times L}$ and let $\boldsymbol{X} \in \{0,1\}^{L \times d}$ be a matrix whose rows are one-hot vectors, for some $L, d \in \mathbb{N}^+$. Then, $[\boldsymbol{v}]_{\boldsymbol{X}} = [\boldsymbol{w}]_{\boldsymbol{X}}$, if, and only if, for every $\boldsymbol{x} \in \mathsf{set}(\boldsymbol{X})$, $[\boldsymbol{v}]_{\boldsymbol{x}} = [\boldsymbol{w}]_{\boldsymbol{x}}$.*

To understand the implication of this result in the context of proving Lemma 26, let us first rearrange the left side of Equation (4) as follows.

**Lemma 28.** *Let $\boldsymbol{v}, \boldsymbol{w} \in \mathbb{Q}^{1 \times L}$ and let $\boldsymbol{X} \in \{0,1\}^{L \times d}$ be a matrix whose rows are one-hot vectors, for some $L, d \in \mathbb{N}^+$. Then, $\mathsf{softmax}(\boldsymbol{v})\boldsymbol{X} = \mathsf{softmax}(\boldsymbol{w})\boldsymbol{X}$, if, and only if, for every $\boldsymbol{x} \in \mathsf{set}(\boldsymbol{X})$,*

$$\sum_{i \in A(\boldsymbol{x})} (\alpha_i - \beta_i) = 0,$$

*where $\alpha_i \coloneqq \mathsf{softmax}(\boldsymbol{v})_i$ and $\beta_i \coloneqq \mathsf{softmax}(\boldsymbol{w})_i$.*

Lemma 28 and Lemma 27 can be seen as complementary decompositions of the left and right side of Equation (4) for each unique one-hot vector in $\boldsymbol{X}$. As a result, we can restate Lemma 26 as follows.

**Lemma 29** (Decomposed Lemma 26). *Let $\boldsymbol{v}, \boldsymbol{w} \in \mathbb{Q}^{1 \times L}$ with $\max_i \boldsymbol{v}_i = \max_i \boldsymbol{w}_i$ and let $\boldsymbol{X} \in \{0,1\}^{L \times d}$ be a matrix whose rows are one-hot vectors, for some $L, d \in \mathbb{N}^+$. Further, we require $\boldsymbol{X}$ to have at least two distinct rows. Then,*

$$\sum_{i \in A(\boldsymbol{x})} (\alpha_i - \beta_i) = 0 \implies [\boldsymbol{v}]_{\boldsymbol{x}} = [\boldsymbol{w}]_{\boldsymbol{x}}, \qquad (6)$$

*for all $\boldsymbol{x} \in \mathsf{set}(\boldsymbol{X})$, where $\alpha_i \coloneqq \mathsf{softmax}(\boldsymbol{v})_i$ and $\beta_i \coloneqq \mathsf{softmax}(\boldsymbol{w})_i$, and*

$$\mathsf{softmax}(\boldsymbol{v})\boldsymbol{X} = \mathsf{softmax}(\boldsymbol{w})\boldsymbol{X} \impliedby [\boldsymbol{v}]_{\boldsymbol{X}} = [\boldsymbol{w}]_{\boldsymbol{X}}. \qquad (7)$$

To prove Lemma 29, we leverage a known result about exponential numbers (as used within softmax) from transcendental number theory, namely that a set of exponential numbers with distinct rational coefficients is linearly independent, also known as the *Lindemann–Weierstrass theorem* [Baker, 1990]. To understand intuitively how this theorem is used, let us assume for simplicity that the softmax is unnormalized, meaning that we can write the left side of Equation (6) as

$$\sum_{i \in A(\boldsymbol{x})} (\exp(\boldsymbol{v}_i) - \exp(\boldsymbol{w}_i)) = 0.$$

With the help of the Lindemann–Weierstrass theorem, we obtain the following claim, which we prove in Appendix E.

**Claim 30.** Let $A, B \subset \mathbb{Q}$ be finite multisets with $|A| = |B|$. Then, the sum

$$\sum_{a \in A} \exp(a) - \sum_{b \in B} \exp(b) = 0,$$

if, and only if, $A = B$.

Hence, with the unnormalized softmax, the left side of Equation (6) holds if and only if $[\boldsymbol{v}]_{\boldsymbol{x}} = [\boldsymbol{w}]_{\boldsymbol{x}}$. However, the full softmax also introduces normalization, which we denote with $Z_\alpha := \sum_{k=1}^n \exp(\boldsymbol{v}_k)$ and $Z_\beta := \sum_{k=1}^n \exp(\boldsymbol{w}_k)$, respectively. As a result, we have the condition

$$\sum_{i \in A(\boldsymbol{x})} (\alpha_i - \beta_i) = 0 \tag{8}$$

$$\Leftrightarrow \sum_{i \in A(\boldsymbol{x})} \frac{1}{Z_\alpha} \exp(\boldsymbol{v}_i) - \frac{1}{Z_\beta} \exp(\boldsymbol{w}_i) = 0 \tag{9}$$

$$\Leftrightarrow \sum_{i \in A(\boldsymbol{x})} \frac{\exp(\boldsymbol{v}_i) \cdot Z_\beta - \exp(\boldsymbol{w}_i) \cdot Z_\alpha}{Z_\alpha Z_\beta} = 0 \tag{10}$$

$$\Leftrightarrow \sum_{i \in A(\boldsymbol{x})} \exp(\boldsymbol{v}_i) \cdot Z_\beta - \exp(\boldsymbol{w}_i) \cdot Z_\alpha = 0 \tag{11}$$

$$\Leftrightarrow \sum_{i \in A(\boldsymbol{x})} \sum_{k=1}^L \exp(\boldsymbol{v}_i + \boldsymbol{w}_k) - \exp(\boldsymbol{w}_i + \boldsymbol{v}_k) = 0. \tag{12}$$

$$\Leftrightarrow \sum_{i \in A(\boldsymbol{x})} \sum_{k=1}^L \exp(\boldsymbol{v}_i + \boldsymbol{w}_k) - \sum_{j \in A(\boldsymbol{x})} \sum_{l=1}^L \exp(\boldsymbol{w}_j + \boldsymbol{v}_l) = 0. \tag{13}$$

Note that the multiset of exponents in the positive exponentials is $[\boldsymbol{v}]_{\boldsymbol{x},\boldsymbol{w}}$ and the set of exponents in the negative exponentials is $[\boldsymbol{w}]_{\boldsymbol{x},\boldsymbol{v}}$. Using Claim 30, we can now restate Lemma 29 once more as follows.

**Lemma 31** (Multi-set only version of Lemma 29). *Let $\boldsymbol{v}, \boldsymbol{w} \in \mathbb{Q}^{1 \times L}$ with $\max_i \boldsymbol{v}_i = \max_i \boldsymbol{w}_i$ and let $\boldsymbol{X} \in \{0,1\}^{L \times d}$ be a matrix whose rows are one-hot vectors, for some $L, d \in \mathbb{N}^+$. Further, we require $\boldsymbol{X}$ to have at least two distinct rows. Then,*

$$[\boldsymbol{v}]_{\boldsymbol{x},\boldsymbol{w}} = [\boldsymbol{w}]_{\boldsymbol{x},\boldsymbol{v}} \implies [\boldsymbol{v}]_{\boldsymbol{x}} = [\boldsymbol{w}]_{\boldsymbol{x}}, \tag{14}$$

*for all $\boldsymbol{x} \in \mathsf{set}(\boldsymbol{X})$ and*

$$\mathsf{softmax}(\boldsymbol{v})\boldsymbol{X} = \mathsf{softmax}(\boldsymbol{w})\boldsymbol{X} \impliedby [\boldsymbol{v}]_{\boldsymbol{X}} = [\boldsymbol{w}]_{\boldsymbol{X}}. \tag{15}$$

We will give the full proof with all details in this section.

First, we will review some number theory background, formally state the Lindemann–Weierstrass theorem and its implications and then give the proof of Lemma 31. Using Lemma 31 and in particular, the equivalent Lemma 26, we finally prove Theorem 24.

**Number theory** We will formally introduce the necessary background on number theory and the Lindemann–Weierstrass theorem. A number is **algebraic** if it is the root of a non-zero single-variable polynomial with finite degree and rational coefficients. For example, all rational numbers $\frac{a}{b}$ with $a, b \in \mathbb{N}^+$ are algebraic, as they are the roots of the polynomial $ax - b$ with integer coefficients. On the other hand, a number is **transcendental** if and only if it is not algebraic. For example, it is known that $\exp(a)$ is transcendental if $a$ is algebraic and non-zero. This last fact follows from the Lindemann–Weierstrass theorem, which we state next [Baker, 1990].

**Theorem 32** (Baker [1990], Theorem 1.4). *Let $a_1, \ldots, a_n$ be distinct **algebraic** numbers. Then, $\exp(a_1), \ldots, \exp(a_n)$ are linearly independent with algebraic rational coefficients.*

Here, we will use the fact that attention uses the $\exp$ function in the softmax and use Theorem 32 to compute injective representations of the GD-WL multisets by expressing them as sums of exponential numbers.

**Proving Lemma 26**  We now prove Lemma 31, equivalent to Lemma 26.

**Lemma 33** (Proof of Lemma 31). *Let $v, w \in \mathbb{Q}^{1 \times L}$ with $\max_i v_i = \max_i w_i$ and let $X \in \{0,1\}^{L \times d}$ be a matrix whose rows are one-hot vectors, for some $L, d \in \mathbb{N}^+$. Further, we require $X$ to have at least two distinct rows. Then,*

$$[v]_{x,w} = [w]_{x,v} \implies [v]_x = [w]_x,$$

*for all $x \in \mathsf{set}(X)$ and*

$$\mathsf{softmax}(v)X = \mathsf{softmax}(w)X \impliedby [v]_X = [w]_X.$$

*Proof.* Note that by assumption $X$ has at least two distinct rows and hence, $A(x) \subset [L]$. As a result, the forward implication follows from the following claim.

**Claim 34.** For all $x \in \mathsf{set}(X)$, if $\max_i v_i = \max_i w_i$ and $A(x) \subset [n]$, then, $[v]_{x,w} = [w]_{x,v} \Rightarrow [v]_x = [w]_x$.

*Proof.* Let $K := \max_i v_i = \max_i w_i$. We begin by sorting the entries in $[v]_x$ and $[w]_x$ in descending order, obtaining sorted vectors $v^*$ and $w^*$. By assumption, we have that $v_1^* = w_1^* = K$. Now, let $i \in [\|[v]_x\|]$ be the smallest number for which $v_i^* \neq w_i^*$. If no such $i$ exists, then $[v]_x = [w]_x$. Otherwise, without loss of generality, we assume that $v_i^* > w_i^*$. We now show that then, the sum $v_i^* + K$ appears at least once more in $[v]_{x,w}$ than in $[w]_{x,v}$.

First, note that there cannot exist some $j > i$ for which $w_j^* + K = v_i^* + K$. Second, for each $j < i$, $v_j^* = w_j^*$, meaning that for each such $j$ where $v_j^* + K = v_i^* + K$ appears in $[v]_{x,w}$, $w_j^* + K = v_i^* + K$ appears in $[w]_{x,v}$.

Hence, $v_i^* + K$ appears at least once more in $[v]_{x,w}$ than in $[w]_{x,v}$, implying $[v]_{x,w} \neq [w]_{x,v}$. As a result, we have that $[v]_x = [w]_x \vee [v]_{x,w} \neq [w]_{x,v}$ which is logically equivalent to $[v]_{x,w} = [w]_{x,v} \Rightarrow [v]_x = [w]_x$. This shows the statement. $\square$

To see why in Claim 34 it is important that $A(x)$ is a strict subset of $[n]$, we note that $A(x) = [L]$ implies $[v]_{x,w} = [w]_{x,v}$, irrespective of whether $[v]_x = [w]_x$. Notably, the proof holds if there exists at least one $i \in [L] \setminus A(x)$, irrespective of whether $v_i = w_i$.

The backward direction follows directly from the fact that $\mathsf{softmax}(v)X$ and $\mathsf{softmax}(w)X$ are functions over $[v]_X$ and $[w]_X$, respectively.

Together with Claim 34, this shows the statement. $\square$

**Proving the GD-WL simulation result**  Now that Lemma 26 has been proven, we will prove the main result, Theorem 24, next.

**Theorem 35** (Proof of Theorem 24). *Let $G := (V(G), E(G))$ be a graph with $n \in \mathbb{N}^+$ nodes and node distance function $d_G : V(G)^2 \to \mathbb{Q}$. Let $d, d_f, T, h \in \mathbb{N}^+$ denote the number of embedding dimensions, the number of hidden dimensions, the number of layers, and the number of attention heads, respectively. Let $L := n + 1$ and let $\hat{X}^0 \in \mathbb{R}^{L \times d}$ and $B \in \mathbb{R}^{L \times L \times h}$ be initial token embeddings and attention bias constructed according to Section 2 using node distance $d_G$. Then, there exist weights for the parameters in $\Theta(d, d_f, T, h)$ such that $\hat{X}^t = X^t$, for all $t \geq 0$, an arbitrary but fixed $\mathsf{hash}$, and using $d_G$ as the distance function.*

*Proof.* Note that the GD-WL produces a finite number of colors at each iteration, and $B$ is constructed from $d_G$ whose co-domain is compact for graphs with finite size. Hence, since each transformer layer is a composition of continuous functions, the domain of each transformer layer is compact. Recall that we want to show that the $t$-th transformer layer can simulate

$$\chi_G^{t+1}(v) := \mathsf{hash}([v]_{X^t}),$$

for all $v \in V(G) \cup \{[\mathtt{cls}]\}$, where $X^t \in \{0,1\}^{L \times d}$ is a one-hot color matrix of the GD-WL colors at iteration $t$. Let $v$ be arbitrary but fixed. We say that $v$ is the $i$-th node in an arbitrary but fixed ordering of $V(G)$.

We restate the transformer layer definition in a simplified form, omitting multiple heads, residual streams, and LayerNorm. In particular, we state that the layer updates only the $i$-th row of the token matrix.

$$\hat{\boldsymbol{X}}(v)^{t+1} = \mathsf{MLP}(\mathsf{softmax}(\hat{\boldsymbol{X}}(v)^t \boldsymbol{W}_Q (\hat{\boldsymbol{X}}^t \boldsymbol{W}_K)^T + v) \hat{\boldsymbol{X}}^t \boldsymbol{W}_V),$$

where we recall that the $i$-th row of $\boldsymbol{B}$ is $v$. We now set $\boldsymbol{W}_Q = \boldsymbol{W}_K$ to all-zeros and $\boldsymbol{W}_V$ to the identity matrix and obtain

$$\hat{\boldsymbol{X}}(v)^{t+1} = \mathsf{MLP}(\mathsf{softmax}(v)\hat{\boldsymbol{X}}^t).$$

We prove the statement by induction over $t$. For the base case at $t = 0$, the token matrix $\boldsymbol{X}$ contains the one-hot colors of the nodes in $V(G)$ as well as the special one-hot color of the [cls] token. Setting $\hat{\boldsymbol{X}}^0 := \boldsymbol{X}$, we have that $\hat{\boldsymbol{X}}^0 \in \{0,1\}^{L \times d}$ and $\hat{\boldsymbol{X}}^0 = \boldsymbol{X}^0$. Further, due to the [cls] token, we know that $\hat{\boldsymbol{X}}^0$ has at least two distinct rows.

Finally, note that, by construction, for each pair of distance vectors $\max_i \boldsymbol{v}_i = \max_i \boldsymbol{w}_i = \max_{v,w \in V(G)} d_G(v,w) + 1$ and that every distance vector $\boldsymbol{v} \in \mathbb{Q}^{1 \times L}$. These two conditions hold throughout the induction and we will use them in the induction step to apply Lemma 26.

In the induction step for $t > 0$, we assume that

1. $\hat{\boldsymbol{X}}^t \in \{0,1\}^{L \times d}$

2. $\hat{\boldsymbol{X}}^t = \boldsymbol{X}^t$

3. $\hat{\boldsymbol{X}}^t$ has at least two distinct rows

We want to prove that the same holds for $t+1$. Let $v, w \in V(G) \cup \{\texttt{[cls]}\}$ be arbitrary but fixed. Note that $\chi^{t+1}(v) = \chi^{t+1}(w)$ if and only if $[\boldsymbol{v}]_{\boldsymbol{X}^t} = [\boldsymbol{w}]_{\boldsymbol{X}^t}$. By the induction hypothesis, we have that $[\boldsymbol{v}]_{\boldsymbol{X}^t} = [\boldsymbol{w}]_{\boldsymbol{X}^t}$ if and only if $[\boldsymbol{v}]_{\hat{\boldsymbol{X}}^t} = [\boldsymbol{w}]_{\hat{\boldsymbol{X}}^t}$. Further, by Lemma 26, we have that

$$\mathsf{softmax}(\boldsymbol{v})\hat{\boldsymbol{X}}^t = \mathsf{softmax}(\boldsymbol{w})\hat{\boldsymbol{X}}^t \iff [\boldsymbol{v}]_{\hat{\boldsymbol{X}}^t} = [\boldsymbol{w}]_{\hat{\boldsymbol{X}}^t},$$

and as a consequence,

$$\mathsf{softmax}(\boldsymbol{v})\hat{\boldsymbol{X}}^t = \mathsf{softmax}(\boldsymbol{w})\hat{\boldsymbol{X}}^t \iff \chi^{t+1}(v) = \chi^{t+1}(w).$$

Hence, there exists an injective function $f$ that maps, for each token $v \in V(G) \cup \{\texttt{[cls]}\}$ with distance vector $\boldsymbol{v}$, the vector $\mathsf{softmax}(\boldsymbol{v})\hat{\boldsymbol{X}}^t$ to a one-hot vector of $\chi^{t+1}(v)$ with $d$ dimensions. Since the domain of the $t$-th transformer layer is compact, $f$ is continuous. Hence, by universal function approximation, there exist weights of the $\mathsf{MLP}$ such that, for each $v \in V(G) \cup \{\texttt{[cls]}\}$, $\hat{\boldsymbol{X}}^{t+1}(v)$ is a one-hot vector of $\chi^{t+1}(v)$. As a result,

1. $\hat{\boldsymbol{X}}^{t+1} \in \{0,1\}^{L \times d}$

2. $\hat{\boldsymbol{X}}^{t+1} = \boldsymbol{X}^{t+1}$

3. $\hat{\boldsymbol{X}}^{t+1}$ has at least two distinct rows.

This completes the induction and proves the statement. $\qquad\square$

## C.2 Upper-bound on expressivity of the GDT

Moreover, we can prove an upper bound on the expressivity of the GDT using a technique adapted from Müller et al. [2024]. We begin by showing the following result.

**Lemma 36.** *Let $G := (V(G), E(G), \ell_V)$ be a graph with $n$ nodes and without edge embeddings, let $\mathbf{W}_Q, \mathbf{W}_K, \mathbf{W}_V \in \mathbb{R}^{d \times d}$ be arbitrary but fixed weight matrices with $d \in \mathbb{N}^+$, and let $\mathbf{B} \in \mathbb{Q}^{n \times n}$ be an attention bias. Let*

$$\alpha(\mathbf{X}, \mathbf{U}) := \mathit{Attention}(\mathbf{X}\mathbf{W}_Q, \mathbf{X}\mathbf{W}_K, \mathbf{X}\mathbf{W}_V, \mathbf{B}).$$

*There exists a distance function $d_G$ over $V(G) \cup \{\texttt{[cls]}\}$ and functions $f, h$ with*

$$f(\mathbf{X}_i) := h(\{\!\{(d_G(i,j), \mathbf{X}_j)\}\!\}),$$

*such that for all $\mathbf{X}$ and all $i$, $\alpha(\mathbf{X}, \mathbf{U})_i = f(\mathbf{X}_i)$.*

*Proof.* We define $d_G$ with have co-domain $\mathbb{Q}^2$ such that

$$d_G(i, j) = [\mathbf{B}_{ij}, I(i = j)],$$

for all $i, j \in V(G) \cup \{[\texttt{cls}]\}$, where $[\cdot]$ is the concatenation operation and $I(i = j)$ is the indicator function. We denote with $d_G(i, j)_k$ the $k$-th element in $d_G(i, j)$, for $k \in \{1, 2\}$. Let

$$g(\mathbf{X}_i, \mathbf{X}_j) := \exp(\mathbf{X}_i \mathbf{W}_Q (\mathbf{X}_j \mathbf{W}_K)^T).$$

We choose $h$ as follows. We note that by definition, $1 = d_G(i, i)_2 > d_G(i, j)_2$ for all $i \neq j$. Hence, $h$ can decompose its input into three arguments:

1. $\mathbf{X}_i$, identified from the tuple $(d_G(i, j), \mathbf{X}_j)$ where $d(i, j)_2 = 1$, i.e., $i = j$,

2. the multiset of distances $\{\!\{d_G(i, j)\}\!\}$,

3. the multiset of vectors $\{\!\{\mathbf{X}_j\}\!\}$.

Then, $h$ computes

$$w_{ij} := \exp(g(\mathbf{X}_i, \mathbf{X}_j) + d_G(i, j)_1)$$

and

$$\tilde{w}_{ij} := \frac{w_{ij}}{\sum_k w_{ik}},$$

for all $i, j$. Finally, $h$ computes

$$\sum_j \tilde{w}_{ij} \mathbf{X_j} \mathbf{W}_V,$$

for all $i$, to obtain $\alpha(\mathbf{X}, \mathbf{U})_i$. $\qquad\square$

Intuitively, the above lemma shows that biased attention can be written as a function over the multiset in the GD-WL if the distance function is a metric. We use this result to show that the GD-WL is at least as expressive as a GDT with relative PEs.

**Proposition 37.** *Let $G := (V(G), E(G), \ell_V)$ be a graph without edge embeddings and let $\mathbf{B} \in \mathbb{Q}^{n \times n}$ be an attention bias. Let $d, d_f, T, h \in \mathbb{N}^+$ denote the number of embedding dimensions, the number of hidden dimensions, the number of layers, and the number of attention heads, respectively. For any choice of parameters $\Theta(d, d_f, T, h)$ for the GDT, there exists a distance function $d_G$ over $V(G) \cup \{[\texttt{cls}]\}$ and a hash function $\texttt{hash}$ for the GD-WL such that for all $t \geq 0$ and all pairs of nodes $i, j \in V(G)$, $\chi^t(i) = \chi^t(j)$ if and only if $\mathbf{X}_i^t = \mathbf{X}_j^t$.*

*Proof.* We prove the statement by induction over $t$. For $t = 0$, the statement holds by definition, as the initial token embeddings $\mathbf{X}^0$ without any absolute PE are simply the node embeddings $\ell_V$ and the initial colors of the GD-WL are chosen to be consistent with the node embeddings $\ell_V$. For $t > 0$, we assume by the induction hypothesis that for all pairs of nodes $i, j \in V(G)$, $\chi^{t-1}(i) = \chi^{t-1}(j)$ if and only if $\mathbf{X}_i^{t-1} = \mathbf{X}_j^{t-1}$. By definition, $\mathbf{X}^t$ is computed via Equation (2), namely

$$\boldsymbol{X}^t := \mathsf{MLP}\big(\mathsf{Attention}(\boldsymbol{X}^{t-1}\boldsymbol{W}_Q, \boldsymbol{X}^{t-1}\boldsymbol{W}_K, \boldsymbol{X}^{t-1}\boldsymbol{W}_V, \boldsymbol{B})\big),$$

where the MLP is applied row-wise. Let

1. $f$ denote the function in Lemma 36 consistent with projections $\mathbf{W}_Q, \mathbf{W}_K, \mathbf{W}_V$ and attention bias $\mathbf{B}$,

2. $\mathsf{onehot} \colon [n] \to \{0, 1\}^n$ denote the function that maps numbers $1, \ldots, n$ to their corresponding $n$-dimensional one-hot vector.

Then, for all $i, j$, we have that

$$\mathsf{MLP} \circ f \circ \mathsf{onehot}(\chi^{t-1}(i)) = \mathsf{MLP} \circ f \circ \mathsf{onehot}(\chi^{t-1}(j))$$

if and only if

$$\boldsymbol{X}_i^t = \boldsymbol{X}_j^t.$$

Finally, there are at most $n$ distinct rows in $\mathbf{X}^t$. Let $h$ be a function that injectively maps each unique row in $\mathbf{X}^t$ to a color in $[n]$. We choose $\texttt{hash} := h \circ \mathsf{MLP} \circ f \circ \mathsf{onehot}$, and have that, for all pairs of nodes $i, j \in V(G)$, $\chi^t(i) = \chi^t(j)$ if and only if $\mathbf{X}_i^t = \mathbf{X}_j^t$. This completes the induction and hence concludes the proof. $\qquad\square$

### C.3 Expressivity of the GDT

Together, Theorem 24 and Proposition 37 correspond to the first and second statements in Theorem 1, respectively. A consequence of Theorem 24 is the fact that the GDT with NoPE is equivalent to the 1-WL. In particular, let $G := (V(G), E(G), \ell_V, \ell_E)$ be a graph with $n$ nodes. Let $\ell_E(v, w) := \mathbf{1}$, for all $v, w \in V(G)$, where $\mathbf{1}$ is the vector containing 1 in every element. Further, let $\ell_E(v, v) := \mathbf{2}$, for all $v \in V(G)$, where $\mathbf{2}$ is the vector containing 2 in every element. Then, the GDT with NoPE is equivalent, according to Theorem 24, to the following update of the GD-WL:

$$\chi_G^{t+1}(v) := \mathsf{hash}\big(\{\!\!\{(d_G(v, w), \chi_G^t(w)) : w \in V(G)\}\!\!\}\big),$$

where $d_G(v, v) = 2$, $d_G(v, w) = 1$ if $(v, w) \in E(G)$, and $d_G(v, w) = 0$, else, for all $v, w \in V(G)$. This can be equivalently written as

$$\chi_G^{t+1}(v) := \mathsf{hash}\big((\chi_G^t(v), \{\!\!\{\chi_G^t(w) : w \in N_G(v)\}\!\!\})\big),$$

giving the 1-WL update rule.

## D  Proofs of Section 3

To guide our proofs of Section 3, we introduce CSL-graphs, obtaining the fact that RWSE cannot distinguish all of these graphs. These results are then expanded to provide an introduction to Theorem 2. We first present the result and proof of Theorem 2, using a simplified version that minimizes the number of random walk steps, leveraging results from Tönshoff et al. [2023]. In addition, the proofs of Proposition 3 and Proposition 47 are then given with additional details on the expressiveness hierarchy obtained from the definition of each PE. We provide additional incremental results for our selection of PEs, complementing those from Section 3.

**Warm-up: CSL graphs**  We begin by introducing a class of simple and intuitive, yet not 1-WL distinguishable graphs, so-called CSL graphs. These graphs consist of an $n$-node cycle with skip connections of length $k, l$ originating from each node. CSL graphs are a canonical example of a graph class requiring a distance measure, motivating additional PEs [Rampášek et al., 2022, Müller et al., 2024]. Here, we show that they cannot be fully distinguished by RWSE and provide guidance on how to find pairs of CSL graphs indistinguishable by RWSE. We first introduce CSL graphs $G_{(n,k)}$ and their properties to prove the following results.

**Definition 38.** Let $n, k$ be natural numbers and $k < n - 1$. $G_{(n,k)}$ defines an undirected graph which is 4-regular. The set of nodes is given by $V(G_{(n,k)}) = \{0, \ldots, n-1\}$. A two-step process gives the edges. First, to construct a cycle in the CSL graph, every edge $\{i, i+1\} \in E(G_{(n,k)})$ for $j \in \{0, \ldots, n-2\}$. Additionally $\{n-1, 0\} \in E(G_{(n,k)})$ holds. Furthermore, the skip links are introduced by defining the sequence $s_1 = 0$ and $s_{i+1} = (s_i + k) \bmod n$ and deriving the edges with $\{s_i, s_{i+1}\} \in E(G_{(n,k)})$.

In addition, we introduce the notation used throughout the following proofs. Considering the skip links introduced in the CSL graphs we denote such a skip link by the mapping $s^k : V(G_{(n,k)}) \to V(G_{(n,k)})$ with $s(v_i) := v_{(i+k) \bmod n}$ for nodes $\{v_1, \ldots, v_n\}$. A traversal to the next node $v_{i+1}$ or $v_{i-1}$ from node $v_i$ is denoted by $s^1$ and $s^{-1}$ respectively. We further provide specific random walks using a tuple of the visited nodes in a graph.

**Proposition 39.** *Two CSL graphs $G_{(n_1,k_1)}$ and $H_{(n_2,k_2)}$ with $n_1 = n_2$ are non isomorphic if $k_1, k_2$ are co-prime natural numbers.*

Given the definition of CSL graphs, it is possible to derive isomorphism results for them. Furthermore, we note that CSL graphs are 1-WL indistinguishable but can be distinguished by various WL variants, such as GD-WL [Zhang et al., 2023].

**Proposition 40.** *There exists at least one pair of CSL graphs that RWSE cannot distinguish for any choice of random walk length.*

Nonetheless, we note that the expressive power of RWSE is sufficient to distinguish many 1-WL indistinguishable graphs; for example, most CSL graphs can already be characterized by RWSE. Furthermore, a minimum step number is given depending on the skip length of each CSL graph.

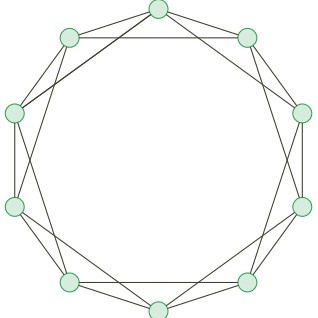 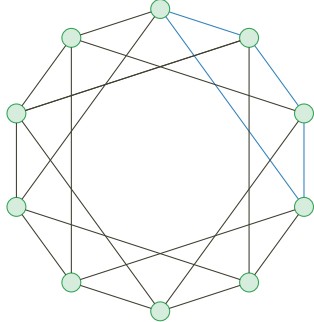

Figure 5: A pair of CSL graphs $G_{10,2}, H_{10,3}$. We note that the path marked in blue does not exist in graph $G$ or has no replacement path.

**Proposition 41.** *Let $n, k \in \mathbb{N}^+$ with $n > k(k+1) + 1$. RWSE can distinguish any pair of CSL graphs with $n$ nodes and skip length $k, k + 1$, with a random walk length of $k + 1$.*

With Definition 38 we derive the proofs for each lemma individually.

**Proof of Proposition 41** For Proposition 41 we consider a subclass of CSL graphs. The minimum node number is specifically chosen to prevent a random walk of $k$ steps from completing a cycle in the graph, even when using $s^{k+1}$ for each step. We follow a two-step process for the proof: First, we gather paths existing in one graph but not the other. Then, in a second step, we show that all different paths of length at most $k$ exist in both graphs. Further, we highlight that for random walks of length less than $k$, the paths are equal in both graphs.

*Proof.* Let $G_{(n,k)}$ and $H_{(n,k+1)}$ be two CSL graphs with skip link mappings $s^k$ and $s^{k+1}$. Further let $n > k(k+1) + 1$. To distinguish them, we denote the same node in both graphs with $v_0$ and $w_0$. Then, for $k$ random walk steps, we first examine whether there exist paths in $G_{(n,k)}$ which are not present in $H_{(n,k+1)}$. These include $(v_0, \ldots, v_{k-1}, v_0)$ and $(v_0, \ldots, v_{n-k+1}, v_0)$ as valid paths in $G_{(n,k)}$, which provide two $k$ step walks with one skip link each. However, we note that such paths are not possible in $H_{(n,k+1)}$ as the skip link $s^{k+1}$ has a length $k + 1$ and therefore no corresponding walk exists. In addition, we show that there exists no other walk in $H_{(n,k+1)}$, which is not present in $G_{(n,k)}$. For this, we must consider the cases of $k$ even or odd.

Case 1: $k$ is even: In this case, we must consider all combinations of skip links and $s^1, s^{-1}$ functions. Since we assume an even number of steps, we know that all return walks with an even number of skip links or no skip links exist in both graphs. However, walks with an uneven number of skip links cannot return to $v_0$ or $w_0$, as with the skip link size of $k$ or $k + 1$, no return walks with at most $k$ steps exist. As can be seen, the skip length does not influence the existence of any of the proposed walks; therefore, they exist both in $G_{(n,k)}$ and $H_{(n,k+1)}$.

Case 2: $k$ is uneven: Given an uneven $k$, we conclude that no return walks with a length of $k$ exist since steps can return neither a skip link nor an uneven number of any combination of skip links and steps can return to the origin node.

For step numbers lower than $k$ the same cases apply in permuted order, depending on the step number. Also, due to the minimal $n$ chosen, no instances exist where a circumference of the graph circle occurs in any combination of steps. As for both cases, there exist no paths which are not present in $G_{(n,k)}$, it follows that there exist additional walks for $G_{(n,k)}$ which do not occur in $H_{(n,k+1)}$. Due to the structure of CSL graphs, the number of random walks, including both returning and non-returning random walks, is the same across all nodes in both graphs, resulting in the statement of Proposition 41.

Since the RWPE encoding is different if a single step returns a different return walk probability, RWPE can distinguish $G_{(n,k)}, H_{(n,k+1)}$ with the given random walk steps. $\qquad \square$

For the following pair of CSL graphs, we consider the computation of the number of returning walks. This enables us to compute the random walk probabilities for a single node directly. Due to the design

of CSL graphs, each node in a graph has the same random walk return probability, thereby allowing us to derive the proof of Proposition 40.

**Proof of Proposition 40**    We consider two CSL graphs $G_{(11,3)}$ and $H_{(11,4)}$. With the computation of the number of returning walks of length $r$, $W^r$ given by $W^r = \sum_{i=1}^{n} \lambda_i^r$, where $\lambda_i$ denotes the eigenvalues of the adjacency matrix, we can compute the number of returning walks for each node in both graphs, since due to the graph structure the number of returning walks is equal for each node. Furthermore, we know that the number of total walks is equal in both graphs, given the graph structure as seen in Figure 5. From this, we compute the fraction of returning walks of varying length $r$ for each node, resulting in the computation of the RWSE embedding. Since both the number of returning walks and the total number of walks are equal for each node in both graphs, as seen in the proof of Proposition 41, we receive the same RWSE embedding for $G_{(11,3)}$ and $H_{(11,4)}$.

In addition to the indistinguishability results obtained for RWSE, we propose a result that initially distinguishes RWSE from RRWP. We then refine this result in Proposition 42, showcasing RRWP to be strictly more expressive than RWSE.

**Proposition 42.** *RRWP can distinguish all pairs of non-isomorphic CSL graphs.*

**Proof of Proposition 42**    In contrast to RWSE, the RRWP embedding uses the full random walk matrix and information about random walks between any two nodes. Because of this, RRWP can capture information that is not visible to RWSE due to the restriction to the diagonal of the random walk matrix.

*Proof.* We consider two arbitrary CSL graphs $G_{(n,i)}$ and $H_{(n,j)}$ with $i, j$ co-prime and $i \neq j$. Then we can compute the RRWP encoding for each node by computing the random walk matrix. We saw from previous CSL graphs that the random walk matrix diagonal can be equal for both graphs, depending on the choice of $i, j$, and the number of random walk steps. However, for RRWP, we also consider non-diagonal elements. Due to $i \neq j$, the RRWP tensor elements differ for each random walk of length 1, since different nodes are connected. Given any injective MLP layer, different RRWP tensor elements result in different RRWP embeddings for the nodes of both graphs, allowing the CSL graphs to be distinguished by RRWP. □

Using the above results for CSL graphs, we derive a first bound for RWSE that depends on the graph structure of the CSL graphs. However, CSL graphs are not distinguishable by 1-WL, leaving open the comparison between RWSE and 1-WL. In the following, we want to further improve our understanding of RWSE and its expressiveness. We first provide an introduction and intermediate result given by Lemma 43 to limit the expressiveness to random walks of sufficient length. Afterwards, we provide the proof of Theorem 2, concluding our examination of RWSE.

**Introduction to Theorem 2**    To prove Theorem 2, we first provide an intermediate result from leveraging the graphs introduced by Tönshoff et al. [2023] for their work. This allows us to derive graphs that need a certain number of random walk steps, depending on their graph structure. We then expand on this concept in our proof of Theorem 2 by giving an example of graphs not distinguishable by RWSE.

**Lemma 43.** *There exists at least one pair of non-isomorphic graphs with order $3n - 1$ that can be distinguished by RWSE only with a random walk length of at least $\mathcal{O}(n)$.*

Since RWSE requires returning random walks to construct the respective embedding, graphs exist that are only distinguishable by random walks of specific length. However, we want to provide a class of graphs requiring at least $\mathcal{O}(n)$ steps. Furthermore, we want these graphs to be 1-WL distinguishable, providing a first step to Theorem 2, where we show the indistinguishability of RWSE and 1-WL. Following Tönshoff et al. [2023] and their evaluation of another random walk-based GNN architecture (CraWL), we adapt their counterexample to our evaluation of RWSE.

*Proof.* We provide a proof by giving a constructed graph only distinguishable by random walks of length greater than $n - 1$, fulfilling the necessary condition of $\mathcal{O}(n)$ for the walk length. Following Tönshoff et al. [2023] with their counterexample for the CraWL algorithm, we adapt the corresponding graphs for the RWSE embedding. Since we only consider the return probabilities of random walks

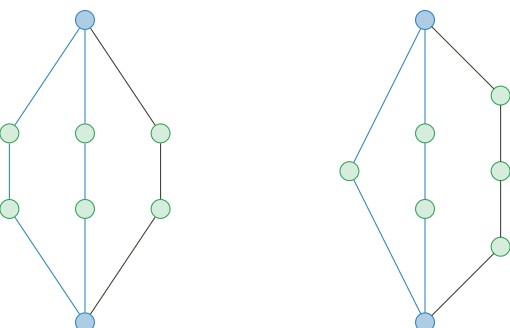

Figure 6: A pair of graphs from the construction method provided by Tönshoff et al. [2023]. Note that these graphs can only be distinguished by a returning random walk of length $\mathcal{O}(n)$ while being 1-WL distinguishable.

in the RWSE embedding, we disregard information from other random walks. Given the graphs in Figure 6 with $n$ nodes, we consider the blue marked nodes. Since the returning walks are the same for both nodes for any walk of length $r \leq n - 1$, the graphs cannot be distinguished for any RWSE embedding with a walk length of $r$. Due to the graph construction, the corresponding walks for all other nodes are the same in both graphs. However, due to the cycle colored in blue, the return walk probabilities differ in both graphs for walks with a length $r' \geq n$. This results in a pair of graphs only distinguishable by random walks of length at least $n$. Results from Tönshoff et al. [2023] allow for constructing further examples with $n$ nodes and order $3n - 1$. By construction, these graphs are distinguishable by 1-WL [Tönshoff et al., 2023], resulting in the stated lemma. □

With the results from Lemma 43, we can now expand the set of graphs not distinguishable by RWSE, while 1-WL is distinguishable. Combining both results, we can determine a set of graphs limiting the expressiveness of RWSE and further investigate the expressive power of random walks.

We provide an example pair of trees not distinguishable by the RWSE encoding. In contrast, all trees are known to be distinguishable by the 1-WL algorithm [Cai et al., 1992]. Combining the findings from Theorem 2 with the CSL graph results in Proposition 41, it follows that RWSE embeddings are incomparable to the 1-WL color refinement algorithm.

**Proof of Theorem 2** For this proof, we first consider a pair of trees shown by Cvetković [1988]. Originally introduced as examples of trees with differing eigenvalues and graph angles, thereby distinguishable by the EA-invariant, these graphs are not distinguishable by the RWSE embedding for an arbitrary number of random walk steps. The proof is split into multiple steps, containing parts of both trees that need to be considered.

*Proof.* We separate the graph as shown in figure 7 into backbones $I, J$ and subtrees $X, Y_1, Y_2$ and the original tree $G$. $X$ stays the same throughout both graphs, whereas $Y_1, Y_2$ change their edges connecting them to the graph. In the first step, we consider the probability of returning to one of the backbone nodes. Going from either of the backbone nodes to the original tree $G$ has a probability of

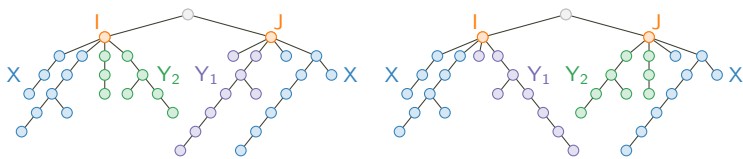

Figure 7: A pair of trees given by Cvetković [1988]. The grey node denotes an arbitrary tree concatenated to the existing tree. Each part of the tree is colored according to its respective appearance. We further label both backbone nodes in orange, denoting the left backbone node with $I$ and the right backbone node with $J$.

$\frac{1}{5}$. Furthermore, going to $X$ from either backbone node has the probability $\frac{2}{5}$. Once in either part of $X$, the return probability to the backbone node is given by $p_X$, which is equal in both graphs. Finally, the probability of going to $Y_1, Y_2$ is denoted by $p_{Y_1}, p_{Y_2}$ respectively, again equal for both graphs. Further, the return probabilities $Y_1$ and $Y_2$ are equal, as can be easily seen from the combination of $Y_1$ and $Y_2$ with the respective backbone node. This allows for a complete evaluation of the backbone nodes, assigning them probabilities $p_I$ and $p_J$.

Secondly, we consider random walks on the nodes of the subtree $X$, without returning to either backbone node. Since $X$ is equal in both graphs, the return probabilities are also the same, denoted by $\mathbf{x}$. However, they differ from the return probabilities in $Y_1$ and $Y_2$, given by $\mathbf{y_1}$ and $\mathbf{y_2}$ respectively. As a result, RWSE can distinguish $X, Y_1, Y_2$ without connecting to the backbone nodes.

Combining our previous knowledge, we now consider return probabilities for nodes in $Y_1$ and $Y_2$ without restricting ourselves to the subtrees. We know that the probability of walking towards the backbone node is given by the respective position of each node in $Y_1, Y_2$, equal in both trees. Once the random walk arrives at either backbone node, the probability to return to said backbone is given by either $p_I$ or $p_J$ and a probability of $\frac{2}{5}$ to return to the originating subgraph. As $p_I$ and $p_J$ are equal in both graphs and $p_{Y_1}, p_{Y_2}$ are equal, it follows that the return probability for nodes $y_1 \in Y_1$ is equal across both graphs, denoted by $p_{y_1 T}$. The same holds for $Y_2$ with the return probabilities denoted by $p_{y_2 T}$.

Finally, for both graphs, the nodes are assigned the RWSE probability vectors $p_I, p_J$ once, $p_X$ 17 times, and $p_{y_1 T}, p_{y_2 T}$ 8 times each. Furthermore, the two backbone nodes of both graphs cannot be distinguished using the RWSE embedding. Therefore, the graphs are equal under RWSE, however as shown by [Cai et al., 1992] every pair of trees can be distinguished using the 1-WL test, resulting in the incomparability of RWSE and the 1-WL test. □

Combining the results of Proposition 40, Lemma 43, and Theorem 2, we obtain fine-grained observations of graph structures not distinguishable by RWSE. While random walks are sufficiently robust to distinguish many common graph structures and graphs not distinguishable by 1-WL, RWSE still fails to distinguish specific trees and graphs originally proposed by Tönshoff et al. [2023].

In the following, we provide proofs supplementing the theoretical expressiveness hierarchy introduced by Zhang et al. [2024] and Black et al. [2024]. We first give an intermediate result relating RWSE and RRWP, thereby providing a lower bound for RRWP and an upper bound for RWSE. Further, we propose adapted results for LPE and SPE, comparing them to other PEs and to each other. The significant results are shown in Proposition 3 and Proposition 4.

**Proof of Proposition 3**   The proof of Proposition 3 is given by simply evaluating the corresponding embeddings for both RWSE and RRWP. Since both embeddings use the same MLP encoder layer, we restrict ourselves to evaluating random walk matrices directly. We first state an extended version of Proposition 3 and provide a proof.

**Lemma 44** (Proposition 3 in the main paper). *Let $G, H$ be two non-isomorphic graphs and $\boldsymbol{P}_k^{RW}(G), \boldsymbol{P}_k^{RW}(H)$ the generated RWSE encodings for both graphs with a random walk length $k$. Then for the generated RRWP encodings $\boldsymbol{P}_k^{RR}(G), \boldsymbol{P}_k^{RR}(H)$ it follows:*

$$\boldsymbol{P}_k^{RR}(G) = \boldsymbol{P}_k^{RR}(H) \Rightarrow \boldsymbol{P}_k^{RW}(G) = \boldsymbol{P}_k^{RW}(H).$$

*In addition, at least one pair of graphs exists that is distinguishable by RRWP but not by RWSE.*

*Proof.* Given random walk matrices $\boldsymbol{R}_G := \boldsymbol{D}_G^{-1}\boldsymbol{A}_G, \boldsymbol{R}_H := \boldsymbol{D}_H^{-1}\boldsymbol{A}_H$, its power matrices up to the power of $k$ and the corresponding RRWP embeddings $\boldsymbol{P}_k^{RR}(G), \boldsymbol{P}_k^{RR}(H)$ for two non-isomorphic graphs we can directly deduce that for each tensor in the RRWP embeddings, corresponding to the RRWP embedding for a single node in each graph, the diagonal elements of the random walk matrix are the same for each power up to $k$. Therefore, it directly follows that $\boldsymbol{P}_k^{RR}(G) = \boldsymbol{P}_k^{RR}(H)$ results in $\boldsymbol{P}_k^{RW}(G) = \boldsymbol{P}_k^{RW}(H)$, with $\boldsymbol{P}_k^{RW}(G), \boldsymbol{P}_k^{RW}(H)$ denoting the RWSE encodings obtained from the same random walk matrices. We provide a simple example of a pair of graphs that are not distinguishable by the RWSE embedding, whereas RRWP can distinguish between them. This example follows directly from Proposition 40 since it is proven there that RWSE cannot distinguish this pair of graphs. However, it directly follows from the definition of the random walk matrix. It assumes that the MLP encoder preserves identity and that the two graphs can be distinguished by their differences in skip lengths, as defined by CSL graphs. □

We now want to consider the expressive power of LPE with respect to RWSE. For this, we use results obtained by Black et al. [2024] and Zhang et al. [2024] and combine them with similar results obtained for SignNet and BasisNet [Lim et al., 2023]. Our additional observations are highlighted in Proposition 4.

**Proof of Proposition 4**   We provide an expanded version of Proposition 4 split into three parts (Lemma 45, Lemma 46, and Proposition 47), considering the expressive power of SAN [Kreuzer et al., 2021] and RWSE first, expanding the proof to LPE and SPE in Lemma 46, and finally upper bounding RRWP by using SPE. With these three parts, we are then able to derive the proposition.

**Lemma 45** (RWSE and SAN). *Let the number $k$ of eigenvalues $\lambda \in \mathbb{R}^n$ and eigenvectors $\boldsymbol{V} \in \mathbb{R}^{n \times n}$ used in the SAN encoding be equal to the number of nodes in non-isomorphic graphs $G, H$. Then given the encodings $\boldsymbol{P}_k^{SAN}(G), \boldsymbol{P}_k^{SAN}(H)$ with it follows:*

$$\boldsymbol{P}_k^{SAN}(G) = \boldsymbol{P}_k^{SAN}(H) \Rightarrow \boldsymbol{P}^{RW}(G) = \boldsymbol{P}^{RW}(H),$$

*for a pair of RWSE encodings $\boldsymbol{P}^{RW}(G), \boldsymbol{P}^{RW}(H)$ and an arbitrary number of random walk steps.*

**Lemma 46.** *(RWSE and LPE) Given Lemma 45 and the LPE embeddings $\boldsymbol{P}_k^{LPE}(G), \boldsymbol{P}_k^{LPE}(H)$ for two non-isomorphic graphs $G, H$ with $k$ nodes it follows:*

$$\boldsymbol{P}_k^{LPE}(G) = \boldsymbol{P}_k^{LPE}(H) \Rightarrow \boldsymbol{P}^{RW}(G) = \boldsymbol{P}^{RW}(H),$$

*for a pair of RWSE embeddings $\boldsymbol{P}^{RW}(G), \boldsymbol{P}^{RW}(H)$ and an arbitrary number of random walk steps. The same result follows by replacing LPE with SignNet, BasisNet, or SPE as the eigenvector-based embedding.*

**Proof of Lemma 45 and Lemma 46**   In the following, we provide the proofs for both lemmas. Since Lemma 46 proposes an extension of the previous lemma, we first show the specialized case for the SAN embedding and expand it to the more general case of eigenvector-based encodings. With proofs provided for both lemmas, we can directly derive Proposition 4 by combining them with Proposition 47.

*Proof.* The proof follows the comparison between SignNet and RWSE presented by Lim et al. [2023]. Since the RWSE embedding is determined by the random walk matrix and its powers, we first determine a corresponding relation between the random walk matrix $(\mathbf{D}^{-1}\mathbf{A})$ and eigenvalues and eigenvectors of the normalized graph Laplacian. Due to the definition of the random walk matrix, the eigenvectors of said matrix are determined by $v_i^R = D^{-1/2}v_i$, where $v_i$ denotes the corresponding eigenvector of the normalized graph Laplacian. This results in the following equation relating the random walk matrix diagonal to the eigenvectors of the graph Laplacian [Lim et al., 2023]:

$$(\text{diag}(\mathbf{D}^{-1}\mathbf{A}))^k = \text{diag}\left( \sum_{i=1}^{k} (1 - \lambda_i)^k v_i v_i^T \right). \tag{16}$$

Following Lim et al. [2023], the linear layer can approximate $\sum_{i=0}^{k}$ and the transformer encoder to approximate $(1 - \lambda_i)^k$ as both are permutation equivariant functions from vectors to vectors. Since eigenvalues and eigenvectors are directly given to the SAN embedding and the linear and transformer encoder layer being able to approximate $(1 - \lambda_i)^k$ for each $\lambda_i$ the approximation directly follows. This approximation assumes using all $k$ eigenvalues and the complete eigenvectors obtained from the decomposition.

For Lemma 46 we consider Equation (16). However, for each embedding, we must consider whether eigenvalues and eigenvectors can be recovered and $(1 - \lambda_i)^k$ can be approximated. We split the following proof for each encoding and assume we use all eigenvalues and eigenvectors.

For LPE, we can directly recover eigenvalues and eigenvectors from the input to each embedding by using the eigenvectors $\mathbf{V}_i$ passed to LPE. Using a sufficiently expressive $\phi$ and $\rho$ LPE is able to approximate $(1 - \lambda_i)^k$. This follows directly from the assumption that $\phi$ and $\rho$ are permutation-equivariant MLPs or more expressive neural network architectures, thereby able to approximate the given functions, as in the SAN case [Lim et al., 2023].

For SPE, a slightly different case has to be considered. Since SPE uses the projection matrices obtained from $\mathbf{V}\mathbf{V}^T$, the eigenvectors must be recovered.

Instead of directly recovering eigenvectors, we use the properties of the underlying projection matrices. From this, we can directly recover the eigenvectors needed from $\mathbf{V}\text{diag}(\phi_i(\lambda))\mathbf{V}^T$ for a suitable $\phi$, which can be reverted by $\rho$ to retain the eigenvectors. For the eigenvalues, we consider $\phi_i$ to be eigenvalue-preserving functions, allowing us to recover the eigenvalues from the diagonalized representation. The remaining proof follows from the observations made by Lim et al. [2023] for SignNet and BasisNet. In addition, SignNet and BasisNet Lim et al. [2023] prove that both embeddings can approximate the RWSE embedding given suitable $\phi$ and $\rho$. $\qquad\square$

**Proposition 47** (RRWP and SPE). *Given the SPE embeddings $\boldsymbol{P}_k^{SPE}(G), \boldsymbol{P}_k^{LPE}(H)$ for two non-isomorphic graphs $G, H$ with $k$ nodes it follows:*

$$\boldsymbol{P}_k^{SPE}(G) = \boldsymbol{P}_k^{SPE}(H) \Rightarrow \boldsymbol{P}^{RR}(G) = \boldsymbol{P}^{RR}(H),$$

*for a pair of RRWP embeddings $\boldsymbol{P}^{RR}(G), \boldsymbol{P}^{RR}(H)$ and an arbitrary number of random walk steps.*

With the partial hierarchy for LPE and SPE, we want to examine random walk-based PEs further. Since RWSE is upper bounded by LPE and incomparable to the 1-WL, it remains to propose an upper bound of RRWP, known to be more expressive than RWSE from Proposition 3.

**Proof of Proposition 47** Following Zhang et al. [2024] with their proof of a representation of the page rank distance using projection matrices, we show that RRWP can be represented using the page rank distance and that such distance can be approximated using information recovered from the SPE embedding. We note that a proof of SPE being more expressive than GRIT is provided by Zhang et al. [2024]. Nonetheless, we reduce the proof to involve the RRWP embedding to align with our theory framework.

First, we consider the representation of the RRWP embedding using the generalized PageRank distance. For this, we consider RRWP as a distance-based embedding of the form

$$\mathbf{P}_k^{\text{RR}}(u, v) = [\mathbf{D}^{-1}\mathbf{A}(u,v), (\mathbf{D}^{-1}\mathbf{A})^2(u,v), \ldots, (\mathbf{D}^{-1}\mathbf{A})^k(u,v)],$$

for nodes $u, v \in V(G)$. Thereby, the RRWP embedding for a selection of nodes can be represented by the multi-dimensional page rank distance $PR$ for a given weight sequence $\gamma_i$:

$$PR(u, v) = \sum_{i=0}^{\infty} \gamma_i (\mathbf{D}^{-1}\mathbf{A})^i(u, v).$$

From Zhang et al. [2024] we obtain the following equality satisfying the needed relation between page rank distance and projection matrices.

$$\sum_{k=0}^{\infty} \gamma_k (\mathbf{D}^{-1}\mathbf{A})^k = \sum_i \left( \sum_{k=0}^{\infty} \gamma_k (1 - \lambda_i)^k \right) \mathbf{P}_i(u, v)(\deg(u)^{-1/2})(\deg(v)^{-1/2})$$

with $\mathbf{P}_i(u, v)$ denoting the element at position $(u, v)$ of the $i$-th projection matrix. However, we still need to recover node-degree information from the SPE encoding and demonstrate that SPE retains the projection matrix information.

Given the property of projection matrices to recover the underlying matrix using eigenvalue decomposition [Zhang et al., 2024], we recover node degree information using the diagonal of the graph Laplacian. Since SPE uses the graph Laplacian to compute eigenvalues and eigenvectors, we can directly recover relevant degree information via the following equation.

$$\mathbf{L} = \sum_{i=1}^{n} \lambda_i \mathbf{P}_i$$

$$\text{diag}(\mathbf{L}) = \text{diag}(\sum_{i=1}^{n} \lambda_i \mathbf{P}_i) = \sum_{i=1}^{n} \lambda_i \text{diag}(\mathbf{P}_i)$$

$$\mathbf{L}_{uu} = \sum_{i=1}^{n} \lambda_i (\mathbf{P}_i(u, u))$$

Following the definition of the SPE encoding, eigenvalues and the elements of the projection matrix can be recovered using suitably expressive $\phi$ and $\rho$. Given $\phi$ to be a 2-IGN and $\rho$ to be a MLP or

1-WL expressive GNN, $\deg(u)^{-1/2}$ can be approximated by $\rho$, whereas $\sum_i \left( \sum_{k=0}^{\infty} \gamma_k (1 - \lambda_i)^k \right)$ can be approximated by a 2-IGN as shown by Maron et al. [2019a], Lim et al. [2023]. This allows for approximating the RRWP embedding via the PageRank distance using SPE as an upper bound, concluding our proof.

Combining the results of Lemma 45, Proposition 47, and Proposition 3, we obtain Proposition 4 directly, supplementing the hierarchy of PEs in their theoretical expressiveness. These results provide a comprehensive theoretical expressiveness hierarchy, showing that random walk-based embeddings are more expressive than the 1-WL test but are bounded by eigeninformation-based embeddings. We note that all embeddings are bounded by the 3-WL test as shown by Zhang et al. [2024].

**Additional results on theoretical expressiveness**  Using notation established in section Appendix B.3 we provide additional proofs to complement the framework established by Black et al. [2024] and Zhang et al. [2024] concerning theoretical expressiveness of PEs. At first, we consider the proof of Lemma 48, highlighting the connection between SAN and SignNet. Then we consider SignNet and BasisNet, expanding on the results of Lim et al. [2023] and adapting them to LPE. We provide additional results to improve the hierarchy of theoretical expressiveness in PEs and to examine LPE further, relating it to other eigenvector-based embeddings.

Throughout these proofs, we consider the respective $\phi$ and $\rho$ to be selected as MLPs or GNNs. For $\phi$, we choose, based on previous analysis by Zhang et al. [2024], a function mapping at most as expressive as a 2-IGN. Similarly, for $\rho$, we select any 1-WL expressive GNN or MLP. Note that these assumptions differ from the selections made in our empirical evaluation.

**Lemma 48.** *Given a sufficiently expressive $\phi$ and $\rho$ for SignNet, aligning with the implementation of Lim et al. [2023] and the original implementation of the SAN embedding, SignNet is at least as expressive as the SAN embedding.*

*Proof.* Let SAN and SignNet be represented by the respective color refinement algorithms shown in Definition 14. To show Lemma 48, we need to show that

$$T_{\text{GP}} \circ T_{\text{WL}}^{\infty} \circ T_{\text{SP2}} \circ T_{\phi}(\chi \text{Sign}) \preceq T_{\text{GP}} \circ T_{\text{ENC}} \circ T_L(\chi_{\text{SAN}}).$$

Since we assume a standard transformer encoder to be at most 1-WL expressive and knowing that $T_{\text{GP}}$ is order preserving concerning Definition 10, we can reduce the above equation to the following expression:

$$T_{\text{WL}}^{\infty} \circ T_{\text{SP2}} \circ T_{\phi}(\chi \text{Sign}) \preceq T_{\text{WL}}^{\infty} \circ T_L(\chi_{\text{SAN}}).$$

This expression can now be evaluated. Given two non-isomorphic graphs $G, H$ and arbitrary nodes $u, v \in V(G)$ and $x, y \in V(H)$ the following holds true:

$$T_{\text{WL}}^{\infty} \circ T_{\text{SP2}} \circ T_{\phi}(\chi \text{Sign}(u,v)) = T_{\text{WL}}^{\infty} \circ T_{\text{SP2}} \circ T_{\phi}(\chi \text{Sign}(x,y))$$
$$T_{\text{SP2}} \circ T_{\phi}(\chi \text{Sign}(u,v)) = T_{\text{SP2}} \circ T_{\phi}(\chi \text{Sign}(x,y))$$
$$\{\!\!\{ T_{\phi}(\chi \text{Sign}(u,v)) \}\!\!\} = \{\!\!\{ T_{\phi}(\chi \text{Sign}(x,y)) \}\!\!\}$$
$$\{\!\!\{ T_{\phi}(\lambda_G, \mathbf{V}^u, \mathbf{V}^v) \}\!\!\} = \{\!\!\{ T_{\phi}(\lambda_H, \mathbf{V}^x, \mathbf{V}^y) \}\!\!\}.$$

From the equivalence of the multisets, it follows directly that $\chi_{\text{SAN}}(u,v) = \chi_{\text{SAN}}(x,y)$ holds for any choice of nodes given an injective $T_{\phi}$. $\square$

With the proof of the SAN embedding concluded, we further evaluate the connections between BasisNet and SignNet and the LPE embedding. First, we show that BasisNet can approximate SignNet, an observation highlighting the differences in expressiveness noted by Lim et al. [2023]. In the second part of the proof, we conclude our comparison of eigenvector-based embeddings and LPE with BasisNet. Throughout the proof we again assume $\phi_{SN}, \phi_{\text{LPE}}$ to be at most 2-IGN expressive and $\rho_{SN}, \rho_{\text{LPE}}$ to be 1-WL expressive.

*Proof.* Let SignNet and BasisNet be represented by the color refinement algorithms from Definition 14. Then for two non-isomorphic graphs $G, H$ with nodes $u, v \in V(G)$ and $x, y \in V(H)$ we consider the color refinement algorithms for both encodings. With this it follows that we have to

show $T_{\text{GP}} \circ T_{\text{WL}} \circ T_{\text{SP1}} \circ T_{\text{BP}} \circ T_{\text{SIAM}}(\chi_{\text{Basis}}) \preceq T_{\text{GP}} \circ T_{\text{WL}} \circ T_{\text{SP2}} \circ T_\phi(\chi_{\text{Sign}})$. Since $T_{\text{GP}} \circ T_{\text{WL}}$ is order preserving we only consider the relation $T_{\text{SP1}} \circ T_{\text{BP}} \circ T_{\text{SIAM}}(\chi_{\text{Basis}}) \preceq T_{\text{SP2}} \circ T_\phi(\chi_{\text{Sign}})$. First of all, we show that $T_{\text{SIAM}}(\chi_{\text{Basis}}) \preceq T_\phi(\chi_{\text{Sign}})$:

$$T_{\text{SIAM}}(\chi_{\text{Basis}})(\lambda_G, u, v) = T_{\text{SIAM}}(\chi_{\text{Basis}})(\lambda_H, x, y)$$
$$\Rightarrow [T_{\text{IGN}}(\chi_{\text{Basis}}(\lambda_G, \cdot, \cdot))]_G(u, v) = [T_{\text{IGN}}(\chi_{\text{Basis}}(\lambda_H, \cdot, \cdot))]_H(x, y).$$

Using the definition of IGN color refinement, we can directly approximate the eigenvalues used in SignNet's initial encoding. Furthermore, a 2-IGN architecture is at least as expressive as the architectures used for $\phi$ in SignNet. Since the multisets of the projection matrices allow us to approximate the eigenvectors used by the SignNet encoding, the initial encoding of SignNet can be approximated, allowing for the approximation of $T_\phi(\chi_{\text{Sign}})$,

$$[T_{\text{IGN}}(\chi_{\text{Basis}}(\lambda_G, \cdot, \cdot))]_G(u, v) = [T_{\text{IGN}}(\chi_{\text{Basis}}(\lambda_H, \cdot, \cdot))]_H(x, y)$$
$$\Rightarrow \chi_{\text{Sign}}(\lambda_G, u, v) = \chi_{\text{Sign}}(\lambda_H, x, y) \Rightarrow T_\phi(\chi_{\text{Sign}})(\lambda_G, u, v) = T_\phi(\chi_{\text{Sign}})(\lambda_H, x, y).$$

Given that $\bar{\chi} = T_{\text{SIAM}}(\chi_{\text{Basis}})$, we now only have to show that $T_{\text{BP}}(\bar{\chi}) \preceq T_{\text{SP2}}(\bar{\chi})$. Using the same nodes as above:

$$T_{\text{BP}}(\bar{\chi})(\lambda, u) = T_{\text{BP}}(\bar{\chi})(\lambda, x)$$
$$\Rightarrow \bar{\chi}_G(\lambda, u, u) = \bar{\chi}_H(\lambda, x, x) \wedge \{\!\!\{\bar{\chi}_G(\lambda, u, v) \colon v \in V(G)\}\!\!\} = \{\!\!\{\bar{\chi}_G(\lambda, x, v) \colon v \in V(H)\}\!\!\} \wedge$$
$$\{\!\!\{\bar{\chi}_G(\lambda, v, u) \colon v \in V(G)\}\!\!\} = \{\!\!\{\bar{\chi}_H(\lambda, v, x) \colon v \in V(H)\}\!\!\} \wedge$$
$$\{\!\!\{\bar{\chi}_G(\lambda, v, v) \colon v \in V(G)\}\!\!\} = \{\!\!\{\bar{\chi}_H(\lambda, v, v) \colon v \in V(H)\}\!\!\} \wedge$$
$$\{\!\!\{\bar{\chi}_G(\lambda, v, w) \colon v, w \in V(G)\}\!\!\} = \{\!\!\{\bar{\chi}_H(\lambda, v, w) \colon v, w \in V(H)\}\!\!\} \wedge$$
$$\Rightarrow T_{\text{SP2}}(\bar{\chi})(\lambda, u, v) = T_{\text{SP2}}(\bar{\chi})(\lambda, x, y).$$

Since both parts of the relation $T_{\text{BP}} \circ T_{\text{SIAM}}(\chi_{\text{Basis}}) \preceq T_{\text{SP2}} \circ T_\phi(\chi_{\text{Sign}})$ hold and all color refinements are considered to be order-preserving and expressiveness preserving, the proof directly follows.

In case of the LPE embedding, the proof follows the same structure with $T_\phi$ being replaced with $T_\phi^{\text{LPE}}$ as given in Definition 14. Since we do not assume $T_\phi$ to be more expressive than $T_\phi^{\text{LPE}}$ and both being bounded by a 2-IGN in expressiveness, we can replace $T_\phi$, and therefore, we omit the proof. $\qquad\square$

# E    Additional technical proofs

**Multiset operations**    Let $D$ be a finite set with an arbitrary but fixed order. We denote the $i$-th element of the order on $D$ by $D_i$. Let $A$ be a finite multiset over $D$. We write $A := \{(a_i, D_i) \mid i \in [|D|]\}$ with $a_i \geq 0$, the multiplicity of element $D_i$ in $A$.

We define $|A| := \sum_i a_i$. Further, let $B := \{(b_i, D_i) \mid i \in [|D|]\}$ be another finite multiset over $D$. We define
$$A \cap B := \{(\min\{a_i, b_i\}, D_i) \mid i \in [|D|]\}$$
and
$$A \setminus B := \{(\max\{a_i - b_i, 0\}, D_i) \mid i \in [|D|]\}.$$

We note that $A \cap B$ is symmetric while $B \setminus A$ is not symmetric. Nonetheless, we prove that if $|A| = |B|$, then $|A \setminus B|$ is symmetric.

**Claim 49.**    Let $A, B$ be two multisets over a finite domain. If $|A| = |B|$, then $|A \setminus B| = |B \setminus A|$.

*Proof.*    We have that

$$|A \setminus B| = \sum_i \max\{a_i - b_i, 0\} = \sum_i a_i - \min\{a_i, b_i\} = |A| - |A \cap B|.$$

Hence, if $|A| = |B|$, then $|A \setminus B| = |A| - |A \cap B| = |B| - |A \cap B|$ and since $|A \cap B|$ is symmetric, $|A \setminus B| = |B| - |B \cap A| = |B \setminus A|$. $\qquad\square$

**Claim 50** (Proof of Claim 30). *Let $A, B \subset \mathbb{Q}$ be finite multisets with $|A| = |B|$. Then, the sum*

$$\sum_{a \in A} \exp(a) - \sum_{b \in B} \exp(b) = 0,$$

*if, and only if, $A = B$.*

*Proof.* Let $f(A, B) = \sum_{a \in A} \exp(a) - \sum_{b \in B} \exp(b)$. Note that for each element $a$ in $A$ that also appears as $b$ in $B$, we have that $\exp(a) - \exp(b) = 0$. Hence, we define $A^* := A \setminus B$ and $B^* := B \setminus A$ and have that $f(A, B) = f(A^*, B^*)$. Further, since according to the lemma statement $|A| = |B|$, we have that $|A^*| = |B^*|$; see Claim 49.

We first show that $f(A, B) = 0$ if and only if $A = B$. To this end, note that the sum is 0 if the positive and the negative summands cancel out, that is, if $A^* = B^* = \varnothing$ and hence, $A = B$. If $A \neq B$, then the above sum is a non-zero sum of exponentials with algebraic exponents, and thus, by Theorem 32, non-zero. Hence, we have $A = B \Leftrightarrow f(A, B) = 0$. $\qquad\square$

**Lemma 51** (Proof of Lemma 28). *Let $v, w \in \mathbb{Q}^{1 \times L}$ and let $X \in \{0, 1\}^{L \times d}$ be a matrix whose rows are one-hot vectors, for some $L, d \in \mathbb{N}^+$. Then, $\mathsf{softmax}(v)X = \mathsf{softmax}(w)X$, if and only if, for every $x \in \mathsf{set}(X)$,*

$$\sum_{i \in A(x)} (\alpha_i - \beta_i) = 0,$$

*where $\alpha_i := \mathsf{softmax}(v)_i$ and $\beta_i := \mathsf{softmax}(w)_i$.*

*Proof.* We have

$$\mathsf{softmax}(v)X - \mathsf{softmax}(w)X = \sum_{i=1}^{n} (\alpha_i - \beta_i) \cdot X_i = \sum_{x \in \mathsf{set}(X)} \sum_{i \in A(x)} (\alpha_i - \beta_i) \cdot x.$$

Since the rows of $X$ are one-hot vectors, $\mathsf{set}(X)$ is linearly independent we have that

$$\sum_{x \in \mathsf{set}(X)} \sum_{i \in A(x)} (\alpha_i - \beta_i) \cdot x = 0,$$

if, and only if, $\sum_{i \in A(x)} (\alpha_i - \beta_i) = 0$, for all $x \in \mathsf{set}(X)$. $\qquad\square$

**Lemma 52** (Proof of Lemma 27). *Let $v, w \in \mathbb{Q}^{1 \times L}$ and let $X \in \{0, 1\}^{L \times d}$ be a matrix whose rows are one-hot vectors, for some $L, d \in \mathbb{N}^+$. Then, $[v]_X = [w]_X$, if and only if for every $x \in \mathsf{set}(X)$,*

$$\{\!\{v_i \mid i \in [n] \wedge X_i = x\}\!\} = \{\!\{w_i \mid i \in [n] \wedge X_i = x\}\!\}.$$

*Proof.* We define, for each $x \in \mathsf{set}(X)$,

$$V(x) := \{\!\{v_i \mid i \in [n] \wedge X_i = x\}\!\}$$
$$W(x) := \{\!\{w_i \mid i \in [n] \wedge X_i = x\}\!\}.$$

For the forward implication, assume towards a contradiction that $[v]_X = [w]_X$ but there exists an $x \in \mathsf{set}(X)$ such that $V(x) \neq W(x)$. However, then there also exists a number $v \in V(x)$ that appears $x$ times in $V(x)$ but $y$ times in $W(x)$, with $x \neq y$. Without loss of generality, we assume that $x < y$. Then, the tuple $(v, x)$ appears fewer times in $[v]_X$ than in $[w]_X$, implying $[v]_X \neq [w]_X$, a contradiction.

For the backward implication, assume towards a contradiction that for all $x \in \mathsf{set}(X)$, $V(x) = W(x)$ but $[v]_X \neq [w]_X$. Then, there exists a tuple $(v, x)$ that appears $x$ times in $[v]_X$ but $y$ times in $[v]_X$, with $x \neq y$. Without loss of generality, we assume that $x < y$. But then, for the vector $x$, there exists a number $v$ that appears fewer times in $V(X)$ than in $W(X)$, implying $V(X) \neq W(X)$, a contradiction. This shows the statement. $\qquad\square$

