# OpenReview forum: "Generalizable Insights for Graph Transformers in Theory and Practice"
_NeurIPS.cc/2025/Conference — NeurIPS 2025 spotlight_

### Official Review · Reviewer_7xCc · 2025-06-29

**Clarity:** 4
**Significance:** 3
**Originality:** 3
**Rating:** 5
**Confidence:** 4

**Summary:**

This paper proposes a new graph transformer architecture called the Generalized Distance Transformer (GDTransformer). Unlike some existing graph transformer architectures that use architectural components specialized to graphs, GDTransformer uses the standard transformer architecture. Despite this, the paper proves that GDTransformer matches the expressive power of GDGraphormer. Given that the expressive power of the GDTransformer is determined by the choice of relative positional encoding, the authors prove several theoretical results comparing the expressive power of different positional encodings. The authors then empirically compare the performance of GDTransformer with different positional encoding on a mixture of real-world and algorithmic benchmarks. The authors also perform a set of experiments on the few-shot learning and size generalization capabilities of GDTransformer.

**Questions:**

## Suggestions

- I feel the term "distance" is misused in this paper, as it is applied to functions that are not metrics by the mathematical definition. For example, line 109 uses $d_G$ for the adjacency function, which is not a metric. Thus, it feels like a misnomer to call the proposed architecture GDTransformer, when it can use any relative positional encoding, not just distances.

- The name "GDTransformer" is very similar to GDGraphormer introduced in [Zhang et al, 2023]. It might be a good idea to change the name to avoid confusion.

- One of the key selling points of this paper is that GDTransformer is a simplificiation of previous graph transformer architectures while maintaining the same theoretical expressive power. However, there is little dicussion of previous graph transformer architectures, so the novelty of GDTransformer may not be obvious to readers. I would suggest adding some discussion comparing GDTransformer to the existing graph transformer architectures it improves on.

**Ethical Concerns:**

["NO or VERY MINOR ethics concerns only"]

**Final Justification:**

Thank you for your rebuttal. I think the additional BREC experiments help connect the theoretical and experimental sections. I am also satisfied with the answers to my questions. I still feel that the results of Table 3 don't entirely fit into the paper; however, I acknowledge that this experiment is insightful even if it is somewhat disconnected from the rest of the paper. I believe that the contribution of proving a standard transformer like GDTransformer matches the expressivity of GD-WL will help to simplify the field of graph transformers research, so I am updating my score to "Accept".

**Limitations:**

Yes

**Quality:**

3

**Strengths And Weaknesses:**

# Strengths

- This paper prove (Theorem 1) that the standard transformer architecture has the same expressive power as specially designed graph transformers like GDGraphormer; namely, both have the same expressive power as the GD-WL test. In contrast to GDGraphormer that used the relative PE both inside and outside of the softmax, GDTransformer proves that it is sufficient to use the relative PE inside of the softmax. Most standard transformer implementation only use relative positional encodings inside of the softmax. This results means that the one can use standard transformer implementations rather than specialized implementations. This will significantly simplify future implementations of graph transformers.
- The authors prove (Theorem 2, Propositions 3 & 4) several new results comparing the expressive power of different positional encodings, e.g. SPE >= LPE, RRWP.
- The authors provide several experiments comparing different choices of positional encodings, including experiments testing few-shot generalization and size generalization. This and the previous result can help practitioners choose among the different options of positional encodings.

# Weaknesses

- There is a disconnect between the theoretical results and the experiments. The theoretical results prove that GDTransformer matches the expressive power of GDGraphormer (Theorem 1) and prove some PEs are more expressive than others (Theorem 2, Propositions 3 & 4). The experiments in Table 1 does compare different positional encodings so validates Theorem 2 and Propositions 3 & 4; however, I am not sure how the experiments in Figure 3 connect to the theoretical results. These experiments test the generalization capabilities of GDTransformer, but I am not sure how this fits into the larger message of the paper. Is there a reason why GDTransformer would generalize better than GDGraphormer or other graph transformer or GNN models? Conversely, there is no experimental validation of Theorem 1.
To the this point, I feel like the biggest weakness of the paper is that there is no comparison between GDTransformer and existing graph transformer models. I feel like such a comparison is necessary to the message of the paper. I see these experiments as serving two purposes.
    - Theorem 1 shows that GDTransformer has the same expressive power as GDGraphormer while being simpler. Therefore, I feel like there should be an experimental comparison between GDTransformer and GDGraphormer to empirically validate this theorem. Potentially they could be compared on the BREC dataset [Wang and Zhang, 2024], which tests the expressive power of a GNN.
    - Figure 3 tests the generalization abilities of GDTransformer. However, there is no point of comparison for generalization, so it is unclear how unique GDTransfomers generalization capabilities are.
- While the authors compare some of the most prominent positional encodings, there are many others that are not included in their comparison, most notably the shortest-path or resistance distances used in GD-Graphormer, which is perhaps the most similar architecture to GDTransformer in the literature.

---

> ### Author Rebuttal · Authors · 2025-07-30
>
> We thank the reviewer for their effort and are happy to discuss their concerns below.
> First, the reviewer says
> > There is a disconnect between the theoretical results and the experiments. [...] The experiments in Table 1 does compare different positional encodings so validates Theorem 2 and Propositions 3 & 4; however, I am not sure how the experiments in Figure 3 connect to the theoretical results. These experiments test the generalization capabilities of GDTransformer, but I am not sure how this fits into the larger message of the paper. Is there a reason why GDTransformer would generalize better than GDGraphormer or other graph transformer or GNN models?
>
> We note that Figure 3 is not intended to show that the GDT is better at generalization than Graphormer-GD or other architectures. Instead, the figure shows capabilities observed with pre-trained GDT models, which might be found in other architectures as well and might be extended to additional tasks. We believe our results provide empirical motivation for the capabilities of GTs in general, with the GDT serving as an example of a GT possessing these capabilities.
>
> Next, the reviewer notes
> > Theorem 1 shows that GDTransformer has the same expressive power as GDGraphormer while being simpler. Therefore, I feel like there should be an experimental comparison between GDTransformer and GDGraphormer to empirically validate this theorem. Potentially they could be compared on the BREC dataset [Wang and Zhang, 2024], which tests the expressive power of a GNN.
>
> We agree with the reviewer that providing empirical expressiveness results on the BREC dataset provides empirical insights into Theorem 1. Hence, we provide these results for the GDT and Graphormer-GD architectures in the following table:
>
> | Model | PE | Basic | Regular | Extension | CFI | Total |
> |-|-|-|-|-|-|-|
> | Graphormer-GD | RWSE | 57 | 50 | 96 | 0 | 203 |
> | Graphormer-GD | LPE | 55 | 40 | 85 |3 | 183|
> | GDT | RWSE | 57 | 50 | 96| 0 | 203 |
> | GDT | LPE | 54 | 39 | 84 | 3  | 180 |
>
> We observe that both PEs showcase non-trivial performance on BREC, and that the differences between GDT and Graphormer-GD are negligible. We add a few remarks to these results:
>
> We choose RWSE and LPE as two representative PEs for random-walk and eigen PE, respectively. We also tried RRWP PEs but observed training instabilities for both models, which we attribute to the very small dataset size of BREC (only 64 samples per task). The small-scale dataset size seems to affect the relative PE RRWP more than the node-level PE RWSE and LPE.
> Our implementation aligns GDT and Graphormer-GD as much as possible, differing essentially only in the style of attention. As such, we expect our comparison to be fair. Moreover, we choose 4 layers, an embedding dimension of 64, 4 attention heads, and 8 random-walk and eigenvalues, respectively.
> We generally observed training instabilities caused by the multiplicative $\phi_1$ of Graphormer-GD (see [9], Equation 5). We tried different choices of element-wise functions for $\phi_1$ but observed Graphormer-GD to be unable to distinguish most graphs in BREC. To achieve the performance above, we manually initialize the weights of $\phi_1$ and $\phi_2$ such that Graphormer-GD initially performs standard attention (akin to the GDT).
>
> In general, we find BREC to be only partially suitable as an empirical evaluation of Theorem 1, as results on BREC may be skewed by hyperparameter choices and the fact that BREC contains only a few training samples. As such, results on BREC do not necessarily indicate a particular theoretical expressiveness advantage. We provide rigorous proof with Theorem 1 of an equivalent expressiveness to Graphormer-GD, while retaining an architecture that does not rely on modifications in attention.
>
>
> In addition, the reviewer says
> >Figure 3 tests the generalization abilities of GDTransformer. However, there is no point of comparison for generalization, so it is unclear how unique GDTransfomers generalization capabilities are.
>
> While we agree that such a comparison would certainly be interesting, the scope of the generalization experiments was not to provide a comprehensive evaluation of graph transformer architectures but rather to demonstrate additional capabilities of pre-trained models obtained from the GDT. We further do not claim these capabilities to be exclusive to the GDT and instead consider them to be promising insights for graph transformers at large.
>
>
> > While the authors compare some of the most prominent positional encodings, there are many others that are not included in their comparison, most notably the shortest-path or resistance distances used in GD-Graphormer, which is perhaps the most similar architecture to GDTransformer in the literature.
>
> We intentionally limit the choice of PEs to commonly used PEs in literature and to PEs providing the most powerful theoretical expressiveness: those based on eigeninformation of the graph Laplacian and those based on random-walk probabilities. For those two types, we selected RWSE and RRWP as two common choices of random-walk PEs ([2], [3], [5]). For the eigen PEs, LaplacianPEs ([5], [6]) and SPE [7] are a popular choice. In our study, we use LPE, a slight variant of LaplacianPEs, which has been shown to be expressive and empirically powerful for standard transformers [8].  As the shortest path and resistance distance are theoretically less expressive than SPE [1] we decided not to include them in the current comparison.
>  However, we agree that the evaluation of additional PEs may pose great future work.
>
>
> > I feel the term "distance" is misused in this paper, as it is applied to functions that are not metrics by the mathematical definition. For example, line 109 uses for the adjacency function, which is not a metric. Thus, it feels like a misnomer to call the proposed architecture GDTransformer, when it can use any relative positional encoding, not just distances.
>
> Aligning our architecture with the definitions by [Zhang et al., 2023], we consider a distance measure to not necessarily constitute a metric. We agree that functions are used that are not metrics by mathematical definition. However, we consider distance as an abstract measure of space between points of information. Similar to Zhang et al. (2023), we consider the adjacency function, which is referred to as a distance in their work.
>
>
> > The name "GDTransformer" is very similar to GDGraphormer introduced in [Zhang et al, 2023]. It might be a good idea to change the name to avoid confusion.
>
> The Graphormer-GD, as introduced by Zhang et al, 2023, is compared in its expressive power to the GD-WL, the same expressiveness that the GDT achieves. Since we call our model Generalized Distance Transformer (GDT) and provide a more straightforward implementation of Graphormer-GD, we believe the names are not too close and that GDT is a suitable name for our architecture.
>
>
> > One of the key selling points of this paper is that GDTransformer is a simplificiation of previous graph transformer architectures while maintaining the same theoretical expressive power. However, there is little dicussion of previous graph transformer architectures, so the novelty of GDTransformer may not be obvious to readers. I would suggest adding some discussion comparing GDTransformer to the existing graph transformer architectures it improves on.
>
> We agree that adding a discussion on the differences between the GDT and other graph transformer architectures would improve the readability of our work.
> Therefore, we add a discussion concerning popular graph transformer architectures and their differences to the GDT in the following (and will include this discussion in the paper):
> Considering the Graphormer-GD, holding the same theoretical expressiveness, the GDT simplifies the architecture involved by removing the need for additional element-wise functions \phi_1^h and \phi_2^h in the attention layer, thereby enabling the GDT to benefit from results obtained for standard biased attention.
> In addition, concerning GRIT [2], the GDT does not require changes to attention, as well as an adaptive degree scaler. Furthermore, GRIT is designed to work with the RRWP PE, which our GDT implementation includes out of the box, allowing for a variety of PEs to be used without adaptation of the architecture design.
> Compared to additional works such as Graphormer [3] and Token-GT [4], we note both improved empirical performance as well as the removal of attention or other architecture modifications in the GDT.
> GraphGPS [5] assumes the transformer architecture to be combined with a GNN to improve performance. However, this deviates significantly from transformer architectures, whereas the GDT provides a pure transformer implementation.
> Overall, the GDT provides a theoretical and empirically powerful and straightforward implementation of a graph transformer, aligning with transformer architectures found across domains.
>
> **If you are content with our provided answers, consider updating your score. Please also let us know if there are any remaining questions/concerns on your side.**
>
> ### References
> [1] Comparing Graph Transformers via Positional Encodings, Black et al, ICML 2024
>
> [2] Graph Inductive Biases in Transformers without Message Passing, Ma et al., ICML 2023
>
> [3] Do Transformers Really Perform Bad for Graph Representation?, Ying et al., NeurIPS 2021
>
> [4] Pure Transformers are Powerful Graph Learners, Kim et al.
>
> [5] Recipe for a General, Powerful, Scalable Graph Transformer, Rampášek et al., NeurIPS 2022
>
> [6] Rethinking Graph Transformers with Spectral Attention, Kreuzer et al., NeurIPS 2021
>
> [7] On the Stability of expressive Positional Encodings for Graphs, Huang et al, ICLR 2024
>
> [8] Aligning Transformers with Weisfeiler-Leman, Müller and Morris, ICML 2024
>
> [9] Rethinking the Expressive Power of GNNs via Graph Biconnectivity, Zhang et al., ICLR 2023

---

> > ### Comment · Reviewer_7xCc · 2025-07-31
> >
> > Thank you for your thoughtful reply and additional experiments. I am satisfied with the answers to all of my questions.
> >
> > > The Graphormer-GD, as introduced by Zhang et al, 2023, is compared in its expressive power to the GD-WL, the same expressiveness that the GDT achieves. Since we call our model Generalized Distance Transformer (GDT) and provide a more straightforward implementation of Graphormer-GD, we believe the names are not too close and that GDT is a suitable name for our architecture.
> >
> > I feel I didn't clearly communicate my thoughts about the name "GDTransformer", so please allow me to clarify. First, I want   to emphasize that your model name in no way affects my score. This is purely a suggestion from a colleague with the intention of helping improve your paper.
> >
> > I can see why you chose the name GDTransformer. It makes complete sense in connection with the GD-WL test. However, my concern is not that the name is inaccurate. Instead, my concern is that because the names GDTransformer and Graphormer-GD are so similar, it could be confusing for people in the graph learning community, especially given how many graph neural networks they are already have to remember. The reasoning behind the name GDTransformer is the same as reasoning behind the name Graphormer-GD. Both names describe a model that has the same expressive power as the GD-WL test, so it could be hard to remember which model is which. If somebody is reading a paper a few years from now and sees the name "GDTransformer", I don't know how they will reliably remember whether this is your model or the model from Zhang et al. A memorable model name also affects how easily someone can find your paper. I usually google "Graphormer paper" or "GAT paper" rather than searching for these papers by their title.
> >
> > One option would be a name that communicates how your model is different from Graphormer-GD. For instance, maybe the name could incorporate the fact that you are simplifying Graphormer-GD, e.g., SimpleGDTransformer, SimplifiedGDTransformer, SimpleYetPowerfulGDTransformer. Such a name would be memorable and easy to distinguish from Graphormer-GD.
> >
> > Of course, all of this is merely a suggestion, and I respect if you want to keep the original name.

---

### Official Review · Reviewer_5CNk · 2025-07-03

**Clarity:** 3
**Significance:** 4
**Originality:** 4
**Rating:** 5
**Confidence:** 3

**Summary:**

This paper starts with introducing GDT, a generalization of recent graph transformer architectures. It then proves that GDTs can be designed and parameterized to simulate the GD-WL algorithm in general. Importantly, in this proof, the paper works with full softmax attention. Using this framework, it evaluates the relative expressiveness of position embeddings used in recent graph transformers. Finally, empirical results are provided to evaluate the applicability of the developed theory.

**Questions:**

- And I correct in assuming that GDT always assumes the existence of edge features. And if they are not present in the dataset then these are assigned to 1/0 based on the adjacency?

**Ethical Concerns:**

["NO or VERY MINOR ethics concerns only"]

**Final Justification:**

The authors have resolved my concerns regarding missing RRWP performance numbers.

**Limitations:**

Yes, limitations have been properly addressed in the paper.

**Quality:**

4

**Strengths And Weaknesses:**

### Strengths

- Being able to rigorously show that a graph transforner with softmax attention is able to emulate GD-WL is a big contribution.

- Since the goal of this work is to takl about expressibility of different position embeddings, and not necessarily their “optimizability”, the choice of restricting experimental evaluation to only large graphs was a good idea.


### Weaknesses

- One real-world dataset per task-type (graph/node/edge classificaiton), seems a little low.

- While the theory predicts SPE is atleast as expressive as LPE, the results in Table 1 show that the performance of SPE is much worse than LPE. How would you explain that?

- Since all 3 real-world datasets considered have a similar number of tokens, what causes RRWP to be OOT on COCO and Code? Also, what would be the estimated training time for RRWP on COCO and Code? If getting results for RRWP on these datasets is feasible, I would suggest including the results and noting down the extra compute time needed for it in the paper.
    Excluding a method based on the inefficiency of current implementation seems a bit wrong. After all, transformers is also slow in theory, but once researchers designed fast attention kernels, using attention became feasible, even for large datasets.

---

> ### Author Rebuttal · Authors · 2025-07-30
>
> We thank the reviewer for their effort and are happy to address their concerns in the following. First, the reviewer says
> > One real-world dataset per task-type (graph/node/edge classificaiton), seems a little low.
>
> Note that while we use fewer real-world datasets than is common in most graph learning papers, we consider exclusively datasets with large scale. We motivate this in our paper by increasing the generalizability of the insights and allowing training at scale. While we only pre-train on three real-world datasets, we provide additional empirical results typically not found in graph transformer papers, such as size generalization and few-shot transfer experiments.
>
> > While the theory predicts SPE is atleast as expressive as LPE, the results in Table 1 show that the performance of SPE is much worse than LPE. How would you explain that?
>
> As observed in previous works [1],[2], theoretical expressiveness does not necessarily imply an increased performance on real-world or synthetic datasets. With SPE designed to be both stable and theoretically expressive, this design may lead to reduced performance on real-world tasks. Furthermore, most PEs are designed concerning their theoretical expressiveness or specific architectures. Therefore, empirical performance of PEs may not align with the results shown in theory. In general, theoretical expressiveness does not necessarily improve empirical performance in PEs [3][4]. As such, studying the interplay between expressiveness and generalization in GTs is a promising research direction. Our streamlined GT architecture provides an ideal basis for this.
>
> Finally, the reviewer notes
> > Since all 3 real-world datasets considered have a similar number of tokens, what causes RRWP to be OOT on COCO and Code? Also, what would be the estimated training time for RRWP on COCO and Code? If getting results for RRWP on these datasets is feasible, I would suggest including the results and noting down the extra compute time needed for it in the paper.
>
> We agree with the reviewer to include RRWP where possible for an extended empirical evaluation. To address this issue, we optimized our code to compute RRWP at runtime for both datasets, allowing us to include the results for Coco and Code. These results can be seen in Table 1, highlighting the performance of RRWP on Code. With an F1 score of 19.42 RRWP performs slightly better than RWSE, SPE, and the NoPE baseline model.  However, we note RRWP to be unstable in training on Coco, with collapses in validation performance observed for multiple learning rates and seeds. Further, RRWP performs worse than other PEs on Coco,
> Furthermore, adding to insight 3 of the paper, we provide the runtime and memory requirements of RRWP computed for both experiments in Table 2. As observed with other datasets, computing RRWP at runtime adds runtime and memory overhead, leading to increased runtime for both tasks compared to other PEs. Especially with Code, we observe double the runtime and memory requirements compared to other PEs. However, the memory requirements for Coco remain similar to those of other PEs. Nonetheless, RRWP requires additional runtime for both tasks.
>
> Test performance:
> | | CoCo (F1) | Code (F1)  |
> | -- | -- | -- |
> RRWP-32|  39.91 $\pm$ 1.07 |  19.42$\pm$ 0.1|
>
>
> Computational requirements:
>
> The computational requirements were computed by running the respective GDT with RRWP for 1000 steps. The GPU memory requirement is computed as the maximum GPU memory allocated during these steps. Runtime was measured as the average runtime for a single step, averaged across 1000 steps.
> | | CoCo | Code  |
> | -- | -- | -- |
> GPU mem. [MB] | 5223.77  |  19221.97 |
> Runtime [s]  | 0.1352 |  0.219
>
> We are happy to answer the reviewer's question:
>
> > And I correct in assuming that GDT always assumes the existence of edge features. And if they are not present in the dataset then these are assigned to 1/0 based on the adjacency?
>
> Correct. This is equivalent to learning an initial edge embedding, which is the same across all edges (but helps the model to distinguish neighbors from non-neighbors in the attention).
>
> **Please let us know if there are any remaining questions/concerns on your side.**
>
> ### References
> [1] On the Stability of expressive Positional Encodings for Graphs, Huang et al, ICLR 2024
>
> [2] Comparing Graph Transformers via Positional Encodings, Black et al, ICML 2024
>
> [3] Aligning Transformers with Weisfeiler-Leman, Müller and Morris, ICML 2024
>
> [4] Recipe for a General, Powerful, Scalable Graph Transformer, Rampášek et al., NeurIPS 2022

---

> ### Comment · Reviewer_5CNk · 2025-08-03
> **Rebuttal Response**
>
> Thank you for answering my questions. I appreciate the authors optimizing their code and providing performance numbers for RRWP that were previously marked OOT. I have a few follow up questions:
>
> 1. What were the rough total training times for the results in the paper? Just a few numbers would be sufficient. I am just curious.
> 2. As far as I am aware, using an attention bias slows down attention kernel implementations a lot. So, maybe it's worthwile to even consider graph transformers without any attention bias at all (not even 1/0 to mark the adjacency) under your framework? E.g. just having laplacian eigenvec position embeddings. Since these have a benefit of being more efficient on the current deep learning hardware/software stack.

---

> > ### Author Response · Authors · 2025-08-04
> >
> > > What were the rough total training times for the results in the paper? Just a few numbers would be sufficient. I am just curious.
> >
> > We are happy to provide some numbers on the total training times for results obtained in the paper.
> > All runtimes are given in a day, hour, minute (d,h,m) format and were obtained using a single Nvidia L40 GPU, 12 CPU cores and 120GB of RAM. Each runtime is given for a single run and seed. Note that these numbers only include a selection from all results in the paper.
> >
> > | Dataset | Model size | PE | Runtime |
> > | - | - | - | - |
> > | PCQ | 16M | RWSE | 1d 4h 5m|
> > | PCQ | 16M | LPE | 1d 5h 19m|
> > | PCQ | 16M | RRWP| 3d 4h 11m|
> > | PCQ | 90M | RWSE| 3d 10h 22m|
> > | CoCo| 16M| RWSE| 1d 7h 55m|
> > | CoCo| 16M| LPE| 1d 7h 0m|
> > | CoCo| 16M| SPE| 1d 11h 5m|
> > | Code| 16M| RWSE| 13h 52m|
> > | Code| 16M| LPE| 14h 11m|
> > | Code| 16M| SPE| 15h 8m|
> > | MST | 16M| RWSE| 1h 34m|
> > | MST | 16M| RRWP| 3h 16m|
> > |MST | 16M | LPE |1h 39m|
> > | MST | 90M| LPE| 3h 57m|
> > | MST | 160M| LPE| 5h 45m|
> > | Flow| 16M| LPE| 0h 57m|
> > | Flow| 16M| RWSE| 0h 55m|
> > | Flow| 16M| RRWP| 1h 14m|
> > | Flow| 16M| SPE| 1h 01m|
> > | Bridges| 16M| RWSE| 1h 13m|
> > | Bridges| 16M| LPE| 1h 40m|
> > | Bridges| 90M| LPE| 2h 51m|
> > | Bridges| 90M| RWSE| 2h 43m|
> >
> > > As far as I am aware, using an attention bias slows down attention kernel implementations a lot. So, maybe it's worthwile to even consider graph transformers without any attention bias at all (not even 1/0 to mark the adjacency) under your framework? E.g. just having laplacian eigenvec position embeddings. Since these have a benefit of being more efficient on the current deep learning hardware/software stack.
> >
> > We thank the reviewer for raising this point. Indeed, running attention kernels without attention bias is faster. Moreover, we can integrate a recent result in [1] into our framework to omit the attention bias for LPE and SPE. Concretely, the authors in [1] show that LPE and SPE can be parameterized such that the unnormalized attention matrix becomes the adjacency matrix; see Theorem 7 and 8 in [1]. In the context of our framework, this reconstructed adjacency matrix implicitly takes the role of the attention bias, similar to how rotary positional embeddings (RoPE) modify the attention scores in causal transformers for language merely through implicit dot-products during attention computation [2]. As such, we improve over the result in [1] in that we also achieve the 1-WL lower-bound but our framework does not additionally require embeddings of the node degrees. Finally, we would like to point the reviewer to [3], which have a similar result for LPE but consider a more non-standard transformer definition. Overall, our framework remains expressive with standard attention without the attention bias for some PEs, allowing for the most efficient attention kernels, but for general PEs requires the attention bias for expressivity. We will add a brief discussion of these connections to the paper and thank the reviewer for their insightful question.
> >
> > [1] Aligning Transformers with Weisfeiler-Leman, Müller and Morris, ICML 2024
> > [2] RoFormer: Enhanced Transformer with Rotary Position Embedding, Su et al., 2021
> > [3] Comparing Graph Transformers via Positional Encodings, Black et al., ICML 2024

---

> > > ### Comment · Reviewer_5CNk · 2025-08-09
> > > **Final response**
> > >
> > > Thanks for the additional details and references! I consider my concerns resolved

---

### Official Review · Reviewer_RXcq · 2025-07-03

**Clarity:** 3
**Significance:** 3
**Originality:** 3
**Rating:** 5
**Confidence:** 3

**Summary:**

This paper introduces Generalized-Distance Transformer (GDT), a new graph transformer (GT) architecture designed to bridge the gap between theoretical expressiveness and practical performance across diverse domains. The authors aim to provide generalizable insights for the design and application of GTs by leveraging standard attention mechanisms alongside advanced positional embeddings (PEs). Experiments over various datasets and tasks show the effectiveness of the proposed GT.

**Questions:**

- Why do you not consider other GT variants?

- How would GDT extend to dynamic graphs (e.g., temporal social networks) or heterogeneous graphs (e.g., knowledge graphs with multiple node/edge types)?

**Ethical Concerns:**

["NO or VERY MINOR ethics concerns only"]

**Final Justification:**

I keep my positive score.

**Limitations:**

yes

**Paper Formatting Concerns:**

None.

**Quality:**

3

**Strengths And Weaknesses:**

Strengths:

- I am not an expert in this field. But it seems that the GDT is shown to be provably expressive. The work advances GT theory by linking standard attention-based architectures to the GD-WL algorithm.

- Experiments span graph-, node-, and edge-level tasks, in- and out-of-distribution settings, and model scales (15M to 160M parameters). Datasets include both real-world benchmarks and synthetic algorithmic tasks. The results show the generalizability and effectiveness of the proposed GT.

Weaknesses:

- The scope of the theoretical contributions and practical applications in this paper needs to be clearly defined. In Section 2.2, the authors state: “While many variations of GTs exist, we consider the standard transformer encoder based on Vaswani et al. [2017].” However, they do not explain why other variants are excluded. Additionally, the paper focuses primarily on four types of positional embeddings (PEs). Are there other PEs available? If so, why were those not considered? This point was not sufficiently clarified.

- Some experimental findings may be difficult to interpret for readers who are not specialists in the field. For example, in Section 4.3, the performance of LPE slightly declines on the MST task with the 160M-parameter model (compared to the 90M model). The paper merely states that it “still outperforms both 15M models” but does not further analyze the cause of this fluctuation. It is recommended to include additional experiments or analyses to improve the robustness of the conclusion.

---

> ### Author Rebuttal · Authors · 2025-07-30
>
> We thank the reviewer for their effort and are happy to address their concerns in the following. The reviewer says
> > The scope of the theoretical contributions and practical applications in this paper needs to be clearly defined. In Section 2.2, the authors state: “While many variations of GTs exist, we consider the standard transformer encoder based on Vaswani et al. [2017].” However, they do not explain why other variants are excluded.
>
> An essential consideration in our paper for both theoretical and empirical results is that we use a standard attention and do not provide any alterations to the attention layer, as is typical in other works on graph transformers. As a result, we do not have to redesign the transformer specifically for graph tasks. This allows us, for example, to use highly optimized standard transformer implementations in practice. However, we agree with the reviewer that further clarification on this may be beneficial and will add additional explanation to the paper.
>
> > Additionally, the paper focuses primarily on four types of positional embeddings (PEs). Are there other PEs available? If so, why were those not considered? This point was not sufficiently clarified.
>
> While a plethora of different PEs are available, our study focuses on the most common and popular types of PEs for GTs: those based on eigeninformation of the graph Laplacian and those based on random-walk probabilities. For those two types, we selected RWSE and RRWP as two common choices of random-walk PEs ([1], [2], [3]). For the eigen PEs, LaplacianPEs ([4], [2]) and SPE [5] are a popular choice. In our study, we use LPE, a slight variant of LaplacianPEs, which is expressive and empirically powerful for standard transformers [6].
>
> Finally, the reviewer says
>
> > Some experimental findings may be difficult to interpret for readers who are not specialists in the field. For example, in Section 4.3, the performance of LPE slightly declines on the MST task with the 160M-parameter model (compared to the 90M model). The paper merely states that it “still outperforms both 15M models” but does not further analyze the cause of this fluctuation. It is recommended to include additional experiments or analyses to improve the robustness of the conclusion.
>
> We observe the performance of LPE and RWSE on the MST tasks to be rather saturated. As both models provide better test scores than the underlying 15M models, we propose beneficial results due to scaling of the model. Further, we note that the size generalization performance of the 160M model with LPE is increased over the 90M model, while showing similar performance at the graph size seen during training.
>
> We are happy to answer the questions of the reviewer below.
>
> First, the reviewer asks,
> > Why do you not consider other GT variants?
>
> We do not consider other GT variants in our work, as the scope of our paper is to provide theoretical and empirical results for our GT without modifications to attention. With previous works relying on these modifications, we aim to highlight the generality of standard transformers and leverage their existing optimized implementations while still being theoretically and empirically powerful on graph data. We will add this distinction more clearly to our work.
>
> Next, the reviewer asks,
>
> > How would GDT extend to dynamic graphs (e.g., temporal social networks) or heterogeneous graphs (e.g., knowledge graphs with multiple node/edge types)?
>
> We do not consider the temporal setting in our evaluation of the GDT, but find it interesting for future work. Heterogeneous graphs could be naturally extended to and captured by the GDT: Different node types can be encoded differently into node embeddings, and edge types can be distinguished by either edge-level tokenization or edge-based biased attention.
>
> **Please let us know if there are any remaining questions/concerns on your side.**
>
> ### References
> [1] Graph Inductive Biases in Transformers without Message Passing, Ma et al., ICML 2023
>
> [2] Recipe for a General, Powerful, Scalable Graph Transformer, Rampášek et al., NeurIPS 2022
>
> [3] Do Transformers Really Perform Bad for Graph Representation?, Ying et al., NeurIPS 2021
>
> [4] Rethinking Graph Transformers with Spectral Attention, Kreuzer et al., NeurIPS 2021
>
> [5] On the Stability of expressive Positional Encodings for Graphs, Huang et al, ICLR 2024
>
> [6] Aligning Transformers with Weisfeiler-Leman, Müller and Morris, ICML 2024

---

> > ### Comment · Reviewer_RXcq · 2025-08-06
> >
> > Thanks for your response. I have no further concerns.

---

### Official Review · Reviewer_Agzt · 2025-07-04

**Clarity:** 3
**Significance:** 2
**Originality:** 2
**Rating:** 4
**Confidence:** 4

**Summary:**

The paper introduces the Generalized-Distance Transformer (GDT), a graph transformer architecture that aims to bridge the gap between theoretical expressivity and practical performance in graph learning. Sepcifically, the authors provide comprehensive comparison among position encodings and experimental validation.

**Questions:**

See Weaknesses.

**Ethical Concerns:**

["NO or VERY MINOR ethics concerns only"]

**Final Justification:**

After I further elaborate my concerns, the authors provide a sufficiently persuasive feedback. I do not have any major concern to this work and have lifted my score.

**Limitations:**

yes

**Paper Formatting Concerns:**

None.

**Quality:**

3

**Strengths And Weaknesses:**

Strengths:

1. The paper provides strong thoeretical proof to compare each of variants of GT.

2. The experiments span 8 million graphs across diverse tasks and domains, including out-of-distribution reasoning. The GDT’s ability to learn transferable representations (e.g., few-shot COCO-to-PASCAL transfer) highlights its practical utility.

3. The work provides practical guidelines for GT design, such as the importance of PEs and attention mechanisms, which are broadly applicable beyond the proposed architecture.

Weaknesses:

1. While the authors provide detailed proof. These information mostly are not surprising, and the authors basically confirm these intuitive idea. For example, insight 1 & 2 are sort of intuitive, as it seems that the authors are simply claiming biased attention and positional encodings are useful. Also, insight 3 has been widely observed in previous work, as these models' performances do not vary too much. I believe the interesting part is in insight 4 & 5.

2. The rankings experimantal results do not align with the theoretical results, which means the Table 1 does not match the expressive power of positional embeddings.

3. There are also some other works[1] analyzes the expressive power of graph transformer. It seems that there's overlap between [1] and Section3 in this paper. Could the authors further discuss the difference from the previous work?

4. Few-shot experiments seem not sufficient. As the authors test this work on 6 benchmarks, the authors only investigate two few shot cases (Bridges -> Cycle detection & COCO -> Pascal). Could the authors conduct more experiments to provide a more convincing results?

[1]What Improves the Generalization of Graph Transformers? A Theoretical Dive into the Self-attention and Positional Encoding

---

> ### Author Rebuttal · Authors · 2025-07-30
>
> We thank the reviewer for their effort in reviewing our work. We are happy to hear that the reviewer finds insights 4 and 5 in our paper interesting. In the following, we address their concerns about insights 1-3.
> First, the reviewer says
> > While the authors provide detailed proof. These information mostly are not surprising, and the authors basically confirm these intuitive idea. For example, insight 1 & 2 are sort of intuitive, as it seems that the authors are simply claiming biased attention and positional encodings are useful.
>
> From a theoretical perspective, we consider proving these results to be important, although they may be regarded as intuitive. As our paper aims to provide rigorous theoretical proof for results obtained, we highlight insights 1 and 2 as theoretical contributions to the understanding of graph transformers. As we highlight in the paper, weighted means, as computed by standard attention, do not fit into the sum-aggregation-based proofs of expressivity common in graph learning. As such, we leverage a specific property of softmax to circumvent issues usually encountered with weighted means in terms of injectively mapping multisets.
> Since many previous works use a modified attention layer, their results cannot be directly applied to our proposed architecture. We provide a rigorous proof to allow a theoretical and empirical comparison of graph transformers using standard attention.
>
> > Also, insight 3 has been widely observed in previous work, as these models' performances do not vary too much.
>
> Previous works ([1], [2], [3]) often compare their proposed PEs favorably to other works in both theory and empirical evaluation. However, we show that these differences are not as pronounced as suggested by previous works when evaluated at scale. Further, we find that the efficiency of the PE is vital to its usage on real-world data at scale, observations not made in previous works such as [1] and [2].
>
>
>
> Next, the reviewer says
> > The rankings experimantal results do not align with the theoretical results, which means the Table 1 does not match the expressive power of positional embeddings.
>
> Many previous PE designs are motivated by their enhanced theoretical expressiveness. In our study, we find that theoretical expressiveness does not necessarily correlate with (predictive) performance at scale. Note expressiveness does not necessarily imply a small generalization gap, but purely measures a model’s approximation capacity.
> Similar results can be observed in the works of Huang et al. [1] and Black et al. [4], showcasing experimental results that do not align with previously obtained theoretical expressiveness. However, in our work, we additionally study the computational cost of each PE in practice, which reveals a far more critical axis to successfully scaling GTs.
>
>
> > There are also some other works[5] analyzes the expressive power of graph transformer. It seems that there's overlap between [5] and Section3 in this paper. Could the authors further discuss the difference from the previous work?
>
> We believe the paper noted by the reviewer to be fundamentally different in its scope than our proposed paper. In [5], the major theoretical results concern the generalization properties of graph transformers, while we investigate theoretical expressivity concerning our proposed architecture and combinations with PEs. We would kindly ask the reviewer to point us to where they believe overlap exists with our work. Nonetheless, we find the suggested work to be relevant to our work and will add it to our related work.
>
>
> > Few-shot experiments seem not sufficient. As the authors test this work on 6 benchmarks, the authors only investigate two few shot cases (Bridges -> Cycle detection & COCO -> Pascal). Could the authors conduct more experiments to provide a more convincing results?
>
> We note that finding additional few-shot tasks is not trivial, for several reasons:
>
> 1. The node and edge feature spaces of upstream and downstream task need to be aligned. This prohibits, for example, transferring from molecular tasks in OGB to other molecular tasks (e.g., QM9). The authors in [6] have recently highlighted this issue for graph foundation models in general.
> 2. Upstream and downstream tasks need to be related. The few-shot transfer we present in the paper leverages the fact that upstream and downstream tasks are semantically related. To obtain additional few-shot tasks requires to find a pair of tasks (A, B) where A is a large-scale pre-training dataset and B has the same feature domain as A, as well as is semantically related. For example, many OGB tasks share the same feature domain as PCQM4Mv2 (2D molecules), but lack plausible biological or chemical relationships. This is also due to the narrow prediction space of PCQM4Mv2, as it merely predicts the HOMO LUMO Gap.
>
> We regard it as a fundamental challenge for future to develop benchmarks which contain aligned feature spaces, semantically related tasks, and large pre-training datasets to allow for rigorous benchmarking of few-shot transfer capabilities such as those we demonstrate with the GDT. Our code base allows for an easy and comprehensive extension to new training and few-shot objectives. This enables us to include additional few-shot tasks derived from pretraining data in the future as they become available.
>
> **If you are content with our provided answers, consider updating your score. Please also let us know if there are any remaining questions/concerns on your side.**
>
> ### References
> [1] On the Stability of expressive Positional Encodings for Graphs, Huang et al, ICLR 2024
>
> [2] Graph Inductive Biases in Transformers without Message Passing, Ma et al, ICML 2023
>
> [3] Sign and Basis Invariant Networks for Spectral Graph Representation Learning, Lim et al, ICLR 2023
>
> [4] Comparing Graph Transformers via Positional Encodings, Black et al, ICML 2024
>
> [5] What Improves the Generalization of Graph Transformers? A Theoretical Dive into the Self-attention and Positional Encoding, Li et al, ICML 2024
>
> [6] Position: Graph Foundation Models are Already Here, Mao et al., ICML 2024

---

> > ### Comment · Reviewer_Agzt · 2025-08-04
> > **Response**
> >
> > Thanks for your detailed clarification.
> >
> > However, my concerns remain:
> >
> > 1. More recent graph transformers have abandoned the classic attention design as it introduces n^2 complexity, hindering the scalability of the model. Building theoretical foundation on such framework kinds of diminished the contribution of the community.
> >
> > 2. The authors claim 'However, we show that these differences are not as pronounced as suggested by previous works when evaluated at scale'. From the experimental part of previous works [4, 5], it seems that it has been enough for us to observe the differences are not pronounced.
> >
> > 3. Do the authors further investigate the reason that the performance at scale is not aligned with theoretical results? Otherwise, it seems the theoretical proof is not useful to the practitioners.
> >
> > 4. In few-shot experiments, whether the first problem can be addressed by a project head? At least for now, the results are still not that convincing.
> >
> >
> > Based on above concerns, I decide to maintain the score.
> >
> > [1]. vcr-graphormer: a mini-batch graph transformer via virtual connections
> >
> > [2]. Less is More: on the Over-Globalizing Problem in Graph Transformers
> >
> > [3]. SGFormer: Simplifying and Empowering Transformers for Large-Graph Representations
> >
> > [4]. Graph Inductive Biases in Transformers without Message Passing
> >
> > [5]. On the Stability of expressive Positional Encodings for Graphs

---

> > > ### Author Response · Authors · 2025-08-05
> > >
> > > We thank the reviewer for engaging into discussion with us. We will comment on the remaining concerns hereafter.
> > >
> > > > More recent graph transformers have abandoned the classic attention design as it introduces n^2 complexity, hindering the scalability of the model. Building theoretical foundation on such framework kinds of diminished the contribution of the community.
> > >
> > > While a growing number of papers has investigated linear-time attention variants, both specifically for graph transformers, as well as more broadly for transformers [1, 2], we would like to highlight that there are several successful real-world applications of graph transformers that use quadratic attention [3, 4, 5]. Moreover, linear attention variants can often be understood as approximations to quadratic attention [2, 6, 7], meaning that theoretical insights for quadratic attention implicitly translate to such linear attention mechanisms (though note that not necessarily all linear attention mechanism are approximations of quadratic attention).
> > > Finally, we would like to say that studying standard attention does not diminish efforts in the community for linear-time attention for graph transformers. Rather, our study provides insights for quadratic graph transformers which could be extended to linear-time graph transformers in the future.
> > >
> > > > The authors claim 'However, we show that these differences are not as pronounced as suggested by previous works when evaluated at scale'. From the experimental part of previous works [4, 5], it seems that it has been enough for us to observe the differences are not pronounced.
> > >
> > > We are happy to elaborate further on why we believe our Insight 3 to provide novel insights. The works cited by the author, GRIT [8] and SPE [9] do not at all conclude that the differences between PEs are not as pronounced. Rather, these works introduce new PEs and demonstrate their superiority over existing PEs. Moreover, GraphGPS [10] conducts many ablation studies over PEs, indicating significant task-specific differences in PE performance; see [10, Appendix B]. As such, our Insight 3 is directly relevant to practitioners seeking to choose an appropriate PE for their application.
> > > Concretely, our findings indicate that the optimal choice of PE is not as task-specific as previously thought. Rather, through careful implementation and given sufficient data for training, differences in predictive performance between PEs vanishes. What remains are concerns regarding efficiency where PEs differ significantly. The latter has not been explored and highlighted sufficiently in previous works, in particular those cited by the reviewer [8,9]. As such, we believe to provide novel and actionable insights for graph transformers at sufficient scale.
> > >
> > > [1] HyperAttention: Long-context Attention in Near-Linear Time, Han et al., ICLR 2024
> > > [2] Rethinking Attention with Performers, Choromanski et al., ICLR 2021
> > > [3] Accurate structure prediction of biomolecular interactions with AlphaFold 3, Abramson et al., Nature 2024
> > > [4] Probabilistic weather forecasting with machine learning, Price et al., Nature 2024
> > > [5] Swallowing the Bitter Pill: Simplified Scalable Conformer Generation, Wang et al., ICML 2024
> > > [6] Less is More: on the Over-Globalizing Problem in Graph Transformers, Xing et al., ICML 2024
> > > [7] NodeFormer: A Scalable Graph Structure Learning Transformer for Node Classification, Wu et al., NeurIPS 2022
> > > [8] Graph Inductive Biases in Transformers without Message Passing, Ma et al., ICML 2023
> > > [9] On the Stability of expressive Positional Encodings for Graphs, Huang et al, ICLR 2024
> > > [10] Recipe for a General, Powerful, Scalable Graph Transformer, Rampášek et al., NeurIPS 2022

---

> > > > ### Author Response · Authors · 2025-08-05
> > > >
> > > > > Do the authors further investigate the reason that the performance at scale is not aligned with theoretical results? Otherwise, it seems the theoretical proof is not useful to the practitioners.
> > > >
> > > > We respectfully disagree with the reviewer that "the performance [of PEs] at scale is not aligned with theoretical results". Our theoretical results show how different PEs are related in terms of expressivity. Increased expressivity does not and is not intended to directly translate to better empirical performance [11]. Rather, expressivity provides guarantees about the classes of graphs which a model can distinguish. This should be seen as a measure of a fundamental capability of the model architecture, not a metric necessarily correlating with increased predictive performance in practice.
> > > > The usefulness of our theoretical proofs for practitioners is summarized in Insight 2: Any choice of PE (among the four PEs we study) enhance the expressivity of the GDT. This means that if practitioners know that expressivity strictly higher than 1-WL is necessary for their concrete application, then our theory gives guidance into how different PEs are related, and in particular, that any choice of PE among RWSE, LPE, RRWP and SPE increases the expressivity of the GDT beyond 1-WL.
> > > >
> > > > > In few-shot experiments, whether the first problem can be addressed by a project head? At least for now, the results are still not that convincing.
> > > >
> > > > We would like to highlight that we regard the few-shot experiments as additional empirical evidence for the capabilities of the GDT. To this end, we identified two cases where effective few-shot transfer is possible and rigorously validated both cases, comparing multiple PEs, multiple shots, and evaluating over multiple random seeds to obtain robust results. Nonetheless, by no means is it our intention to claim that we conducted an exhaustive few-shot study on the GDT over a wide range of tasks. After discussion with the reviewer we see the need for a more nuanced discussion of our few-shot results and to highlight more clearly the difficulties in identifying meaningful few-shot tasks for evaluation to encourage future work in this direction. As such, we regard our results as a first step to studying these few-shot transfer capabilities in GTs with more evaluation needed for  comprehensive insights across application domains. We will make this more clear in future versions of our work.
> > > >
> > > > [11] WL meet VC, Morris et al., ICML 2023

---

> > > > > ### Comment · Reviewer_Agzt · 2025-08-08
> > > > > **Response**
> > > > >
> > > > > Thanks authors for their patient clarification. I currently see the value this research. I believe most of my concerns have been addressed. Hence, I would like to increase my rating. Good luck.

---

### Decision · Program_Chairs · 2025-09-17

**Decision:**

Accept (spotlight)

**Comment:**

Despite strong performance, Graph Transformer (GT) architectures lack unified theoretical and practical insights due to diverse attention mechanisms, positional embeddings, and expressivity approaches. This paper proposes the Generalized-Distance Transformer (GDT), which integrates recent GT advancements into a standard attention framework, and provide a fine-grained analysis of its representational power. All the reviewers agree that the paper provides strong theoretical insight and should be accepted.